# SubspacePath Pruner: Inference-time Pruning via Probe-based Representation–Parameter Coupling

Zhiren Gong[1,2] Yikun Hou[1,3] Fan Wu[1] Che Wang[1] Fuyao Zhang[1] Tiantong Wu[1] Yurong Hao[1]
Jiaming Zhang[1] Yiyang Duan[1] Tiantong Wang[1] Fei Huang[4] Chau Yuen[5] Wei Yang Bryan Lim[1]

## Abstract

Large-scale dedicated application of LLMs in diverse scenarios increasingly demands specialized model inference behavior under strict constraints of accuracy, latency, and memory. However, the heterogeneous and long-tailed nature of real-world specialized scenarios makes it difficult to obtain training data and optimize models. We study a practical *inference-time* specialization setting: given an LLM base, we compile a reusable, budget-bounded pathway/subnetwork within a specific scenario. Our approach is motivated by an empirical coupling phenomenon: input scenario sets aligned with similar *representation subspaces* (e.g., domain) in embedding space tend to activate a consistent and sparse set of internal *reasoning pathways* in model parameter space. To build the bridge between them, we propose probe-based SUBSPACEPATH PRUNER with two core components: (1) **Domain-Basis Synthesis (DBS)** constructs a quasi-orthogonal basis of domain axes in embedding space, serving as a stable coordinate system. (2) **Probe-based Scenario Pruning (PSP)** uses efficient layer-wise linear probes to estimate axis alignment and compute budgeted head-wise pathways for a specific scenario. Experiments on LLaMA-2-13B show **29.3** average Recall on cross-domain tests (vs. 24.7 dense) and **21.6** on cross-dataset tests (vs. 25.5 dense) with **1.27×** speedup at **30%** pruning ratio. We have code available in GitHub repository and project page in SubspacePath Pruner.

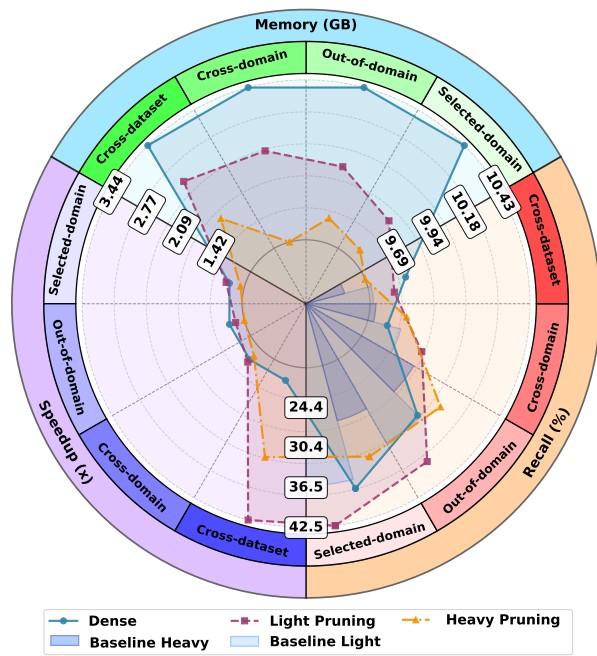

*Figure 1.* **Model-average radar chart** demonstrates the robustness of SUBSPACEPATH PRUNER across various domains and datasets with individual models behavior provided in Figure 14.

## 1 Introduction

Large Language Models (LLMs) are becoming the core engines behind modern intelligent applications with strong reasoning and generative capabilities (Vaswani et al., 2017; Yao et al., 2023; Schick et al., 2023; Wang et al., 2024). As these real-world applications become increasingly dedicated, specialized LLMs are typically required in specific scenarios (where models is repeatedly invoked within coherent contexts) (Belcak et al., 2025; Belcak, 2025). However, scenario-specific LLM is substantially constrained by the heterogeneous and long-tailed nature of real-world application scenarios, where high-quality scenario-specific training data and optimization pipelines are often unavailable or prohibitively expensive (Kaplan et al., 2020).

Motivated by these constraints, *post-training specialization* (Frankle & Carbin, 2019; Wan et al., 2023) has emerged as a practical method for deriving inference instances in data-scarce and budget-limited settings. Compared with con-

[1]College of Computing and Data Science, Nanyang Technological University, Singapore [2]Interdisciplinary Graduate Programme, Nanyang Technological University, Singapore [3]Department of Mathematics and Mathematical Statistics, Umeå University, Sweden [4]Alibaba Group, China [5]School of Electrical and Electronic Engineering, Nanyang Technological University, Singapore. Correspondence to: Zhiren Gong <zhiren001@e.ntu.edu.sg>, Jiaming Zhang <jiaming.zhang@ntu.edu.sg>, Wei Yang Bryan Lim <bryan.limwy@ntu.edu.sg>.

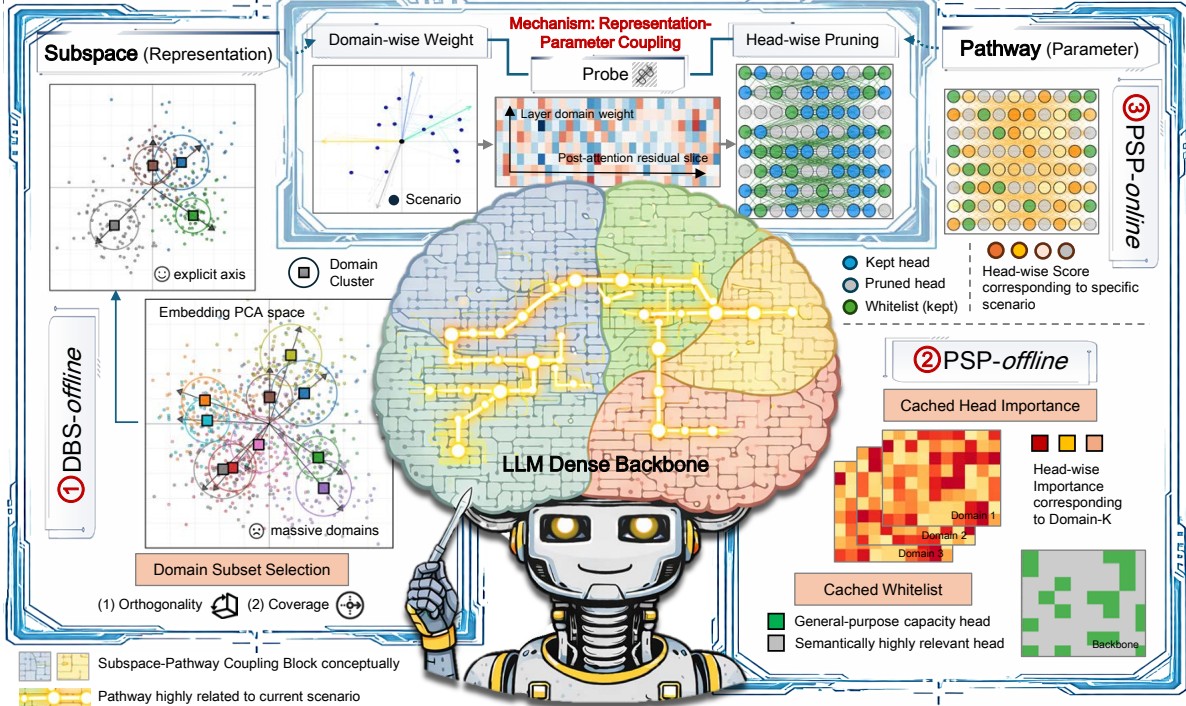

*Figure 2.* **Overview of SUBSPACEPATH PRUNER.** ①**DBS** defines a stable basis of domain axes in embedding space, and **PSP** ②trains a lightweight *probe* with head importance and whitelist cached and ③uses probe to couple axes to sparse internal head pathways.

ventional specialization methods requiring additional training (e.g., fine-tuning, distillation) or inference-time adaptation (e.g., retrieval augmentation, long-context prompting), which introduce complexity and high cost in practical inference (Hu et al., 2022; Hinton et al., 2015; Lewis et al., 2020; Dai et al., 2019; Beltagy et al., 2020; Shazeer et al., 2017), post-training specialization produces model instances *practically* via *inference-time computing* applied after pre-training. Specifically, existing post-training specialization methods are primarily built upon *model compression* (Yuan et al., 2023; Yang et al., 2024a), with *budget-constrained pruning* or *structured cache routing* mechanisms (e.g., MoE-style router-based pruning) being representative instantiations (Hou et al., 2025; Dong et al., 2024; Wee et al., 2025). However, these approaches still exhibit persistent gaps under practical constraints. First, compression methods (Michel et al., 2019; Frantar & Alistarh, 2023; Sun et al., 2023; Lele et al., 2025) often rely on global or average static importance criteria, which induce sensitivity and brittle behavior across diverse scenarios. Second, structured cache routing heavily depends on router design and calibration, making it vulnerable to router mis-specification and scenario mismatch. Therefore, these limitations raise a question: *Can we practically and robustly extract scenario-specific instances from an LLM base without extra scenario-specific training data while preserving the capability of inference?*

To address this, we investigate the internal space of LLMs and identify a consistent structural regularity:

*representation-subspaces (e.g., domains) in embedding-space are coupled with sparse internal reasoning parameter-pathways in model-space.*[1] Specifically, (i) for each quasi-orthogonal domain subspace, only a subset of head-wise computation pathways is repeatedly activated, suggesting a stable pathway set. And (ii) these pathway sets are partially separable across domains, motivating a *probe-based coupling* mechanism that produces interpretable and reliable pathways for robust pruning (Tenenbaum et al., 2000; Yao et al., 2024; Gao et al., 2024). Driven by these observations, we establish a pragmatic principle to guide the design of our method: instead of optimizing compression ratios alone, we should preserve *scenario-related key pathways* and remove redundancy (Zhang et al., 2024a; He & Lin, 2025).

We propose a scenario-specific structured pruning framework, SUBSPACEPATH PRUNER, at *inference-time* as shown in Figure 2. This framework provides an interpretable pruning pipeline that operates within the inference budget by explicitly coupling representation subspaces with reasoning pathways. It consists of two core components:

- **Domain-Basis Synthesis (DBS)**: ① a quasi-orthogonal basis of domain axes in embedding-space which provides a stable coordinate system for subsequent pathway scoring and selection.

---

[1]*Subspaces* refer to domain representation sets in embedding space (usually aligned with similar scenarios), and *pathways* denote component selection (e.g., head-wise pathways), while *subnetworks* referred generically to pruned model instances.

- **Probe-based Scenario Pruning (PSP)**: ② efficient layer-wise probes (Alain & Bengio, 2016) trained to evaluate *subspace→pathway* coupling with whitelists. ③For specific scenarios, we compile budgeted head-wise *pathway*, yielding executable pruned subnetworks practically.

Experiments across 4 models and 6 dataset splits as shown in Figure 1 show that SUBSPACEPATH PRUNER achieves **35.8 / 25.5** Recall on cross-domain / cross-dataset tests (vs. 30.5 / 24.3 dense) at **15–20%** pruning ratio, and **29.8 / 21.6** with up to **1.76×** speedup at **25–30%** pruning ratio.

## 2 Related Work and Background

### 2.1 Post-Training Compression Method

Under inference constraints on accuracy, latency, and memory, *post-training* compression has become a practical approach to obtaining model instances by directly producing executable compressed models from LLM bases (Han et al., 2016; Frankle & Carbin, 2019; Wan et al., 2023). Typically, retraining-free unstructured pruning methods, such as SparseGPT (Frantar & Alistarh, 2023) and Wanda (Sun et al., 2023), estimate importance from weights or activations while still relying on immense computational burdens. Meanwhile, recent structured pruning emphasizes executable budgets (e.g., layers, blocks, or heads), yielding more performance preserved and memory benefits (Ma et al., 2023; Ashkboos et al., 2024; Yao et al., 2024; He & Lin, 2025). However, these approaches often rely on *global* or *average* importance/reconstruction objectives, making pruned models sensitive to diverse scenarios (Michel et al., 2019; Lele et al., 2025; Yuan et al., 2023; Yang et al., 2024a). In contrast, we move to *scenario-specific* pathway calculation, using embedding-space domain axes as a coordinate system to identify and preserve axis-aligned pathways, yielding an executable pruned instance for specialized reuse.

### 2.2 Structural Sparsity and Reasoning Pathways

Our approach builds on the observation that Transformer computation is often concentrated in sparse, task-specific subsets of components rather than utilizing all static weights (Michel et al., 2019; Voita et al., 2019). Mechanistic interpretability studies further suggest that specific model "behaviors" are driven by localized internal "mechanisms", and that enforcing sparsity can make these structures more traceable (Elhage et al., 2021; OpenAI, 2025). While prior work on structured pruning focuses on global redundancy removal or generic compression (Frantar & Alistarh, 2023), the challenge remains in how to dynamically activate the specific "reasoning pathways" that support a target behavior during inference. Our method addresses this by treating these pathways not as static parameters, but as functional units that can be selectively engaged. In parallel, our method shares the input-dependent sparsity principle with mixture-of-experts (MoE), but targets dense-checkpoint specializa-

tion (Shazeer et al., 2017; Lepikhin et al., 2021; Fedus et al., 2022; Jiang et al., 2024). Unlike MoE token-level routing over FFN experts, we compiles a reusable scenario-level head mask from a fixed dense model.

### 2.3 Representation Subspaces and Probing

Another pillar of our work is the internal structure of the LLM's embedding space. Research in representation probing has long demonstrated that task-relevant information is often encoded in linear subspaces or axes (Alain & Bengio, 2016; Belinkov, 2022). Recent advances in sparse feature modeling (e.g., SAEs) further reveal that these internal activations form a partially separable manifold that can be recovered to identify specific semantic domains (Gao et al., 2024; Bricken et al., 2023). While recent "probe-driven pruning" (Le et al., 2025) uses probing signals to guide model reduction, it often lacks a formal grounding in representation alignment, leading to decisions that are difficult to generalize across diverse scenarios. By contrast, we leverage the stability of these latent subspaces as "coordinates" to provide the necessary signaling for our pruning mechanism.

## 3 Preliminaries & Motivations

### 3.1 Problem Formulation

We consider inference-time settings in which a frozen pre-trained model is repeatedly invoked within a coherent context, denoted as *scenario*. In our pipeline, only inputs (i.e., queries) are observable, while the ground truth is unavailable to avoid data leaks.

**Representation subspaces in embedding space.** We treat embedding space as an ambient $d$-dimensional vector space $\mathbb{R}^d$ in which token-level embeddings live. Suppose we have access to a small collection of coarse semantic domain sources (e.g., topical repositories or application-specific query banks), which we use solely as an indexing device to construct an embedding-space structure.

Let $\mathcal{K} = \{1, \ldots, M\}$ index these domain sources, and $\mathcal{X}$ denote the input space. For each domain $k \in \mathcal{K}$, we are given a input-only pool $\mathcal{P}_k = \{x_i\}_{i=1}^{N_k} \subset \mathcal{X}$. Let $\phi : \mathcal{X} \to \mathbb{R}^d$ be a fixed, non-learned sequence-to-vector map, implemented by mean-pooling frozen backbone token embeddings after stopword filtering, which produces one embedding vector $\mathbf{e}(x) = \phi(x) \in \mathbb{R}^d$ per input and is used only to define the DBS representation coordinate without being trained, routed, or updated during pruning. We form the embedding set $\mathcal{E}_k = \{\mathbf{e}(x) : x \in \mathcal{P}_k\} \subset \mathbb{R}^d$, and build $\mathcal{E}_k$ by a low-dimensional linear *subspace* $\mathcal{U}_k \subset \mathbb{R}^d$ estimated from $\mathcal{E}_k$ via PCA (Abdi & Williams, 2010; Fernando et al., 2013; Gong et al., 2012; Vidal, 2011). Intuitively, $\mathcal{U}_k$ captures dominant embedding-space directions associated with pool $k$; throughout, we call $\mathcal{U}_k$ as *axis* for brevity. We denote the collection of candidate axes by $\mathbb{U} = \{\mathcal{U}_k\}_{k \in \mathcal{K}}$.

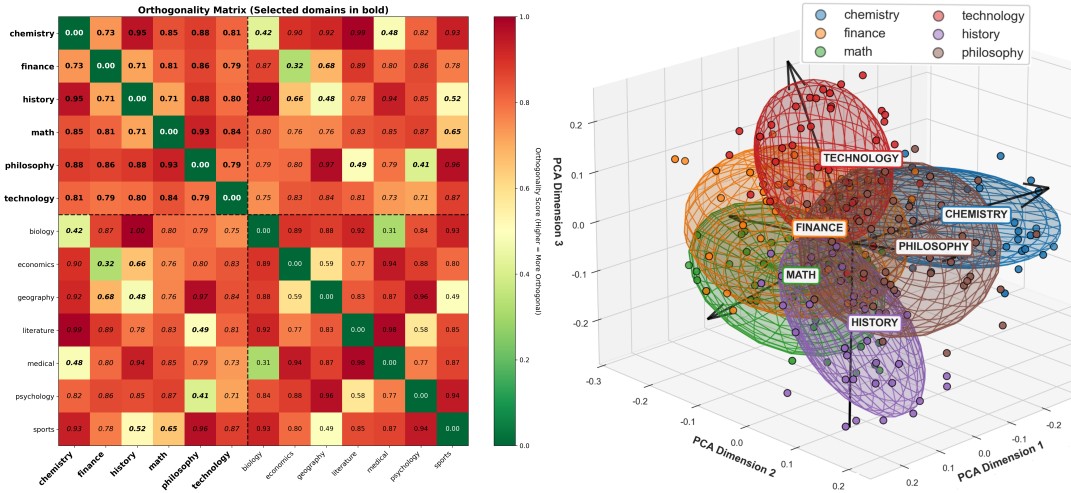

*Figure 3.* **Separation of Domain representation in embedding space.** (Left) Orthogonality heatmap across domains. (Right) Projection of the selected domain showing spatial separation.

**Reasoning pathways in parameter space.** Let $f_\theta$ be a pretrained Transformer with parameters $\theta$ and $h$-head $L$ layers. We index attention heads by $\mathcal{H} = \{(\ell, h) : \ell \in \{1, \ldots, L\}, h \in \{1, \ldots, H_\ell\}\}$. *Pathways* are structured computational routes realized by head-wise components across layers. Scenario-specific pruning is represented by a binary sign $m \in \{0, 1\}^{|\mathcal{H}|}$, indicating which head-wise components are retained (Voita et al., 2019).

**Pruned subnetworks and residual-stream slice.** A pathway pruning $m$ yields an executable pruned instance as a *subnetwork*. We write $f_\theta^{(m)}$ for the model obtained after disabling the head-wise components indicated by $m$.

For an input $x \in \mathcal{X}$, $r^{(\ell)}(x) \in \mathbb{R}^d$ denotes the *post-attention residual stream* at layer $\ell$ averaged across tokens(Elhage et al., 2021) as an observation cross-section to measure the alignment of subspace/axis $\mathbb{U}$. For $\ell \in \{1, \ldots, L\}$ and $k \in \mathcal{K}$, we denote a generic axis-referenced alignment signal by $\alpha_k^{(\ell)}(x) \in \mathbb{R}$, whose estimator is specified later.

### 3.2 Problem Statement: Inference-time Specialization

Given a frozen foundation model $f_\theta$ and a specific scenario $s$ with input streams $\{x_t\}_{t=1}^{T_s}$ ($x_t \in \mathcal{X}$), our goal is to compile a budget-bounded pathway pruning $m_s$ once at scenario start and reuse it throughout the scenario, yielding an executable pruned instance $f_\theta^{(m_s)}$ (Wan et al., 2023).

### 3.3 Empirical Motivation

We provide empirical evidence for **Subspace–Pathway Coupling**, which serves as the foundational motivation for our framework. Concretely, (i) domain pools induce a compact and separable distribution in embedding that supports **DBS**, and (ii) axis-referenced readouts from residual-stream provide a reliable *subspace→pathway* coupling signal that supports **PSP**.

**Evidence 1: Domain pools induce separable distribution.** Across domain pools $\{\mathcal{P}_k\}_{k \in \mathcal{K}}$, embedding vectors form low-rank subspaces $\{\mathcal{U}_k\}$ that are often well-separated under subspace similarity measures (e.g., principal angles, CKA-style comparisons) (Kornblith et al., 2019) shown in Figure 3. This indicates that embedding space inherently provides a stable coordinate from which a quasi-orthogonal axis basis can be constructed for downstream measurements. We deem this evidence to have the potential to positively affect various router-based approaches.

**Evidence 2: Subspace-pathway alignment is noteworthy.** For inputs $x \in \mathcal{X}$, residual-stream slices $r^{(\ell)}(x)$ exhibit consistent correspondence to embedding-level subspaces to some extent, enabling axis-referenced alignment readouts such as $\alpha_k^{(\ell)}(x)$ (Alain & Bengio, 2016; Belinkov, 2022; Tenney et al., 2019; Hewitt & Manning, 2019; Jawahar et al., 2019) shown in Figure 4. Moreover, effective scenario-specific computation is concentrated on a small subset of heads, and the influential subset varies systematically with axis-aligned patterns (e.g., reflected by $\alpha_k^{(\ell)}$), consistent with prior observations of highly non-uniform head contributions (Michel et al., 2019; Voita et al., 2019). In addition, we observe a small subset of heads that is consistently activated across domain pools and scenarios, which is denoted as a domain-invariant whitelist $\mathcal{W} \subseteq \mathcal{H}$ (Tenney et al., 2019; Jawahar et al., 2019; Kovaleva et al., 2019).

We further provide a detailed theoretical derivation in Appendix E. Together, these results establish a practical *subspace → pathway* coupling from embedding-space coordinates to parameter-space execution pathways: semantic axes indicate what scenario information should be preserved, while head pathways determine how such information is executed. This coupling turns pruning from generic parameter removal into scenario-conditioned pathway selection, motivating SUBSPACEPATH PRUNER in the following section.

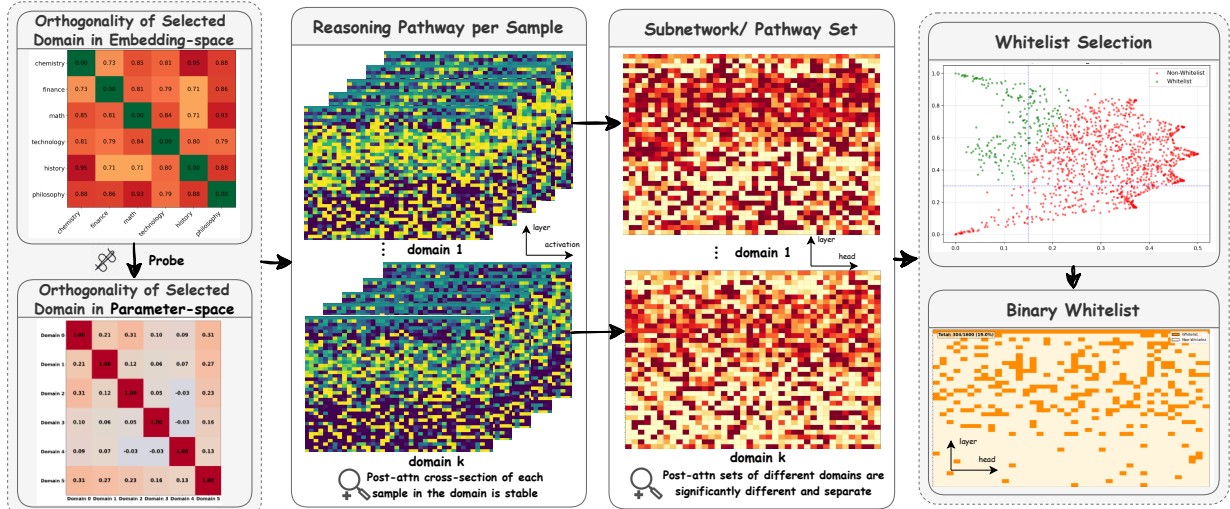

*Figure 4.* **Subspace–pathway coupling.** Embedding-level axes provide a stable coordinate system; probe-based axis alignment readouts from residual slices correlate with head-level pathway, enabling scenario-specific compilation of budget-bounded pathway pruning.

# 4 Methodology

**Overview.** Following the principles in Sections 3.2 and 3.3, our framework operates as a training-free, scenario-level compiler that separates lightweight learning from real-time inference. Specifically, the offline stage (Section 4.1) utilizes input-only pools $\{\mathcal{P}_k\}$ to construct a stable coordinate basis $\mathbb{U}_\star$ via **DBS**, while **PSP** trains layer-wise probes and caches importance aligned to these axes. At online inference (Section 4.2), **PSP** evaluates the initial scenario inputs against the pre-trained probes to resolve axis alignment. This signal is combined with cached importance to compile a pruning mask $m_s$ under budget $B$, which is then reused for all subsequent turns to ensure zero optimization overhead during deployment. The complete algorithm pipeline is shown in Appendix A.

## 4.1 Offline Preparation

### 4.1.1 Domain-Basis Synthesis (DBS)

**Stopword filtering and PCA of embedding.** Let $\mathcal{S}$ be a fixed stopword list. For each input sequence $x$, we remove tokens in $\mathcal{S}$ to obtain $\tilde{x}$. We then compute an embedding $\mathbf{e}(x) = \phi(\tilde{x}) \in \mathbb{R}^d$ as in Section 3.1. Token repetition is preserved in $\tilde{\mathcal{P}}_k = \{\tilde{x}_i\}_{i=1}^{N_k}$ and thus contributes to a *density signature* (Christopher et al., 2008). We construct a shared PCA projection matrix $P \in \mathbb{R}^{d \times d'}$ with $d' = M - 1$ on the pooled embeddings, where $P$ stacks the top-$d'$ principal directions (Jolliffe, 2002). We then project by $\tilde{\mathbf{e}}(x) = P^\top \mathbf{e}(x) \in \mathbb{R}^{d'}$. Although centroid-based, each axis is built after lexical denoising and shared PCA projection and selected by orthogonality–coverage, serving as a stabilized semantic anchor rather than a naive token average.

**Domain subset selection.** We first construct a candidate axis from each pool. For each pool $k \in \mathcal{K}$, we compute the centroid in the PCA space (Rocchio, 1971): $\boldsymbol{\mu}_k =$

$\frac{1}{|\mathcal{P}_k|} \sum_{x \in \mathcal{P}_k} \tilde{\mathbf{e}}(x)$, $\mathbf{u}_k = \frac{\boldsymbol{\mu}_k}{\|\boldsymbol{\mu}_k\|_2} \in \mathbb{R}^{d'}$, Here we treat the *axis* as $\mathcal{U}_k = \mathrm{span}(\mathbf{u}_k)$ and denote $\mathbb{U} = \{\mathcal{U}_k\}_{k \in \mathcal{K}}$.

We then implement domain subset selection with the objective of orthogonality and coverage. We select a compact axis set $\mathcal{K}_\star \subseteq \mathcal{K}$ by balancing:

*(i) Orthogonality.* For a subset $\mathcal{K}'$, define

$$\mathrm{Ortho}(\mathcal{K}') = 1 - \frac{2}{|\mathcal{K}'|(|\mathcal{K}'| - 1)} \sum_{i < j, \ i,j \in \mathcal{K}'} \left( \mathbf{u}_i^\top \mathbf{u}_j \right)^2, \quad (1)$$

which penalizes axis overlap (Björck & Golub, 1973).

*(ii) Coverage.* Let $\Sigma$ be the global covariance of all pooled embeddings in the PCA space: $\Sigma = \mathrm{Cov}(\{\tilde{\mathbf{e}}(x) : x \in \cup_k \mathcal{P}_k\})$. Let $U(\mathcal{K}')$ be the matrix whose columns are $\{\mathbf{u}_k\}_{k \in \mathcal{K}'}$ and $\Pi(\mathcal{K}')$ be the projection matrix onto $\mathrm{span}(U(\mathcal{K}'))$ (orthonormalization implied). Define

$$\mathrm{Cover}(\mathcal{K}') = \frac{\mathrm{tr}(\Pi(\mathcal{K}') \Sigma)}{\mathrm{tr}(\Sigma)} \in [0, 1], \quad (2)$$

measuring how much global semantic variance is captured by the selected axes (Jolliffe, 2002). We then maximize a combined score:

$$J(\mathcal{K}') = \lambda_{\mathrm{ortho}} \cdot \mathrm{Ortho}(\mathcal{K}') + \lambda_{\mathrm{cov}} \cdot \mathrm{Cover}(\mathcal{K}'). \quad (3)$$

In practice, we use greedy screening to detect the elbow point, as detailed in Appendix B. The selected basis is $\mathbb{U}_\star = \{\mathcal{U}_k\}_{k \in \mathcal{K}_\star}$ (Satopaa et al., 2011).

### 4.1.2 Probe-based scenario-level Pruning (PSP)

**Layer-wise probes.** For each layer $\ell$ and each selected domain $k \in \mathcal{K}_\star$, we train a linear probe on the post-attention residual slice $r^{(\ell)}(\tilde{x}) \in \mathbb{R}^d$. Let $\mathbf{g}_{\ell,k} \in \mathbb{R}^d$ denote the probe weight vector (bias omitted for simplicity) and

$z_{\ell,k}(x) = \mathbf{g}_{\ell,k}^\top r^{(\ell)}(\tilde{x})$ be the probe logit. We output a 1-vs-rest probability: $p_{\ell,k}(x) = \sigma(z_{\ell,k}(x)) \in (0, 1)$, where $\sigma$ is sigmoid (Alain & Bengio, 2016; Belinkov, 2022). Training uses labels over the union set $\cup_{k \in \mathcal{K}_\star} \mathcal{P}_k^{\mathrm{tr}}$: each example from $\mathcal{P}_k^{\mathrm{tr}}$ is treated as positive for $k$ and negative for $k' \neq k$ (independent 1-vs-rest logistic regressors).

The training objective is the binary cross-entropy loss

$$\begin{aligned} \mathcal{L}_{\ell,k} = -\mathbb{E}_{x \sim \cup_{j \in \mathcal{K}_\star} \mathcal{P}_j^{\mathrm{tr}}} & [y_k(x) \log p_{\ell,k}(x) \\ & + (1 - y_k(x)) \log (1 - p_{\ell,k}(x))] , \end{aligned} \quad (4)$$

where $y_k(x) = 1$ iff $x \in \mathcal{P}_k^{\mathrm{tr}}$, and 0 otherwise. During training, this corresponds to a standard 1-vs-rest setup with one domain label per sample. During inference, we interpret $\{p_{\ell,k}(x)\}_{k \in \mathcal{K}_\star}$ as independent relevance scores rather than a simplex distribution, since real inputs can mix multiple knowledge axes and improve generalization.

**Head-wise importance.** We define a head's contribution to the post-attn residual write. Let $w_{\ell,h}(x) \in \mathbb{R}^d$ denote the residual write-back vector attributable to head $(\ell, h)$ for input $x$ (i.e., the head-wise component added into the residual stream at layer $\ell$; Elhage et al., 2021). We compute $w_{\ell,h}(x)$ as follows.

Let $r^{(\ell)}(x)$ denote the full post-attention residual at layer $\ell$, obtained by applying the output projection o_proj to the concatenated outputs of all heads. Let $r_{-h}^{(\ell)}(x)$ denote the residual obtained by applying o_proj to the concatenated outputs of all heads except head $h$. Then $w_{\ell,h}(x) = r^{(\ell)}(x) - r_{-h}^{(\ell)}(x)$ to ensure that the sum of all head contributions equals the full residual and avoids bias-related artifacts from the output projection. We map this write into the shared PCA space: $\tilde{w}_{\ell,h}(x) = P^\top w_{\ell,h}(x) \in \mathbb{R}^{d'}$. Then the domain-specific head importance is defined as the expected axis-aligned energy:

$$I_{\ell,h,k} = \mathbb{E}_{x \sim \mathcal{P}_k^{\mathrm{tr}}} \left[ \frac{(\mathbf{u}_k^\top \tilde{w}_{\ell,h}(x))^2}{\|\tilde{w}_{\ell,h}(x)\|_2^2 + \epsilon} \right], \quad k \in \mathcal{K}_\star. \quad (5)$$

where $\epsilon$ is a small constant for numerical stability. Intuitively, $I_{\ell,h,k}$ is large if head $(\ell, h)$ consistently writes along the selected embedding-axis $\mathbf{u}_k$ on domain-$k$ inputs, and small if it is irrelevant. We compute $\{I_{\ell,h,k}\}$ once offline and cache them for inference-time compilation.

**Whitelisting.** We identify a set of always-retained heads $\mathcal{W} \subseteq \mathcal{H}$ to preserve general-purpose capacity. We select a set of domain-invariant *backbone heads* by statistical significance testing. Specifically, for each head $(\ell, h)$, we compute its importance variance across domains: $\sigma_{\ell,h} = \mathrm{Std}(\{I_{\ell,h,k} : k \in \mathcal{K}_\star\})$ and mean importance: $\mu_{\ell,h} = \frac{1}{|\mathcal{K}_\star|} \sum_{k \in \mathcal{K}_\star} I_{\ell,h,k}$. We retain heads that exhibit low domain variance (indicating domain-invariance) and sufficient average importance:

$$\mathcal{W}_{\mathrm{bb}} = \{(\ell, h) \in \mathcal{H} : \sigma_{\ell,h} \leq \sigma_{\mathrm{thresh}}, \mu_{\ell,h} \geq \mu_{\mathrm{thresh}}\}, \quad (6)$$

where $\sigma_{\mathrm{thresh}}$ and $\mu_{\mathrm{thresh}}$ are thresholds determined by statistical significance testing. Keep $\mathcal{W} = \mathcal{W}_{\mathrm{bb}}$ always during compilation. Statistical validation confirms that whitelisted heads have significantly lower domain variance than non-whitelisted heads (Mann-Whitney U test, $p < 0.001$; Algorithm details in Appendix C).

## 4.2 Online Probe-based Scenario-level Pruning

Given a new scenario $s$, we compile a reusable pruning $m_s$ under a budget $B$ using only the first $T_0$ inputs. In practice, we compile the mask from the first scenario input only, relying on the shared semantic signal of a coherent scenario rather than future-turn information. All probe evaluations are performed on the frozen backbone $f_\theta$.

**Scenario-level domain mixture and head scoring.** For each layer $\ell$ and domain $k$, we aggregate the probe outputs over stopword-filtered input $\tilde{x}_t$: $\bar{p}_{\ell,k}(s) = \frac{1}{T_0} \sum_{t=1}^{T_0} \hat{p}_{\ell,k}(\tilde{x}_t)$. We further define a scenario-level domain mixture by pooling over layers: $q_k(s) = \frac{1}{L} \sum_{\ell=1}^{L} \bar{p}_{\ell,k}(s), k \in \mathcal{K}_\star$. We then quantify scenario knowledge breadth using the normalized entropy:

$$\begin{aligned} H(q(s)) &= - \sum_{k \in \mathcal{K}_\star} \frac{q_k(s)}{\sum_{j \in \mathcal{K}_\star} q_j(s)} \log \left( \frac{q_k(s)}{\sum_{j \in \mathcal{K}_\star} q_j(s)} \right), \\ c(s) &= \frac{H(q(s))}{\log |\mathcal{K}_\star|} \in [0, 1]. \end{aligned} \quad (7)$$

Low $c(s)$ indicates a narrow scenario; high $c(s)$ indicates a broader cross-domain scenario. We score heads based on alignment and cached importance. For each domain $k$, we compute domain-specific head scores using an effective domain similarity $s_k^{\mathrm{eff}}$ that is obtained via a non-linear transformation of the raw domain similarity $q_k(s)$ (Equation (45)). The effective similarity amplifies high similarities to preserve more heads for highly relevant domains while reducing preservation for less relevant domains. The final head score aggregates across all domains:

$$\mathrm{score}_{\ell,h}(s) = \sum_{k \in \mathcal{K}_\star} s_k^{\mathrm{eff}} \cdot I_{\ell,h,k}, \quad (8)$$

where $s_k^{\mathrm{eff}}$ is the effective domain similarity for domain $k$ after non-linear transformation. This instantiates the subspace–pathway coupling: probe alignment indicates *what the scenario needs*, and $I_{\ell,h,k}$ indicates *which heads realize that axis*. The non-linear transformation ensures more precise pruning by avoiding under-pruning highly relevant domains while reducing over-preservation of less relevant domains.

**Budget, pruning strength, and pruning compilation.** We treat the inference budget $B$ as a constraint on the pruned model cost $\mathcal{C}\left(f_{\theta'}^{(m_s)}\right) \leq B$. Since exact cost is hardware-dependent, we implement the budget via an equivalent head budget $K_B$ (e.g., a mapping from heads to latency/memory). We introduce a single *pruning-strength* hyperparameter $\eta \in (0, 1]$ that controls the overall aggressiveness, and let the

*Table 1.* **Performance comparison of SUBSPACEPATH PRUNER at different pruning ratios for LLaMA-2-13B-chat across various datasets.** Token-level Recall (Recall) values shown as mean ± standard error of the mean (SEM). Retention = (Recall$_{pruned}$/Recall$_{dense}$) × Speedup. Pruning ratios: left value (baseline), right value (ours, average); ours prunes more aggressively.

| Category | Method | XdomainBench | | | Retention | Cross-dataset Test | | | Retention |
|---|---|---|---|---|---|---|---|---|---|
| | | Selected | OOD | Cross | (S/O/C) | CSQA | NQ | ARC | (C/N/A) |
| **Dense Model** | | | | | | | | | |
| LLaMA-2-13B | | 29.6±1.45 | 26.1±1.10 | 18.4±0.74 | 1.00 / 1.00 / 1.00 | 22.27±1.20 | 30.25±1.07 | 23.87±0.89 | 1.00 / 1.00 / 1.00 |
| **Moderate Pruning Ratio** | | | | | | | | | |
| | | 80%∼79.6% | 89%∼88.4% | 86%∼85.5% | | 90%∼89.9% | 82%∼81.5% | 81%∼80.2% | |
| Unstruct. | DaSS | 33.82±1.56 | 28.62±1.16 | 19.89±0.79 | 0.42 / 0.49 / 0.50 | 20.10±1.14 | 18.80±0.96 | 19.22±0.82 | 0.24 / 0.15 / 0.35 |
| | Wanda | 26.87±1.46 | 27.26±1.14 | 15.43±0.74 | 0.35 / 0.46 / 0.45 | 17.63±1.09 | 17.27±0.98 | 14.38±0.75 | 0.18 / 0.11 / 0.24 |
| Struct. | LLM-Pr. | 29.34±1.49 | 27.81±1.14 | 18.92±0.75 | 0.95 / 1.02 / 0.99 | 21.08±1.17 | 27.63±1.14 | 23.09±0.89 | 0.91 / 0.88 / 0.94 |
| Activ. | RIA | 27.06±1.47 | 27.03±1.14 | 15.12±0.73 | 0.36 / 0.47 / 0.46 | 17.98±1.10 | 17.24±0.98 | 14.33±0.75 | 0.19 / 0.12 / 0.25 |
| | Probe Pr. | 29.02±1.58 | 27.81±1.28 | 19.08±1.12 | 1.22 / 1.33 / 1.24 | 21.75±1.30 | 27.64±1.41 | 23.32±1.34 | 1.17 / 1.09 / 1.17 |
| **Ours-SubspacePath** | | **43.00**±2.40 | **32.50**±1.90 | **20.20**±3.00 | 1.48 / 1.56 / 1.78 | 19.43±1.14 | 33.66±1.35 | 22.91±1.08 | 0.94 / 1.20 / 2.12 |
| **Aggressive Pruning Ratio** | | | | | | | | | |
| | | 71%∼71.0% | 83%∼82.8% | 72%∼71.3% | | 82%∼81.2% | 70%∼69.5% | 69%∼68.3% | |
| Unstruct. | DaSS | 35.27±1.57 | 27.84±1.15 | 15.40±0.71 | 0.87 / 0.84 / 0.80 | 18.13±1.10 | 12.42±0.75 | 16.29±0.79 | 0.49 / 0.22 / 0.72 |
| | Wanda | 18.12±1.26 | 26.57±1.15 | 8.77±0.54 | 0.16 / 0.47 / 0.24 | 13,70±0.99 | 8.09±0.71 | 9.50±0.64 | 0.17 / 0.06/ 0.31 |
| Struct. | LLM-Pr. | 27.30±1.46 | 28.15±1.15 | 17.36±0.71 | 0.60 / 1.05 / 0.61 | 21.20±1.18 | 19.12±0.95 | 21.01±0.82 | 0.92 / 0.49 / 0.70 |
| Activ. | RIA | 18.94±1.28 | 26.69±1.15 | 8.52±0.54 | 0.33 / 0.91 / 0.45 | 12.48±0.93 | 8.06±0.71 | 9.75±0.64 | 0.31 / 0.10 / 0.32 |
| | Probe Pr. | 15.19±1.25 | 27.84±1.28 | 11.54±0.91 | 0.43 / 1.29 / 0.52 | 21.65±1.30 | 11.34±1.00 | 6.96±0.80 | 1.19 / 0.31 / 0.25 |
| **Ours-SubspacePath** | | **34.70**±2.30 | **30.40**±1.90 | **22.90**±3.20 | 1.24 / 1.36 / 1.92 | 18.40±1.12 | 24.89±1.22 | 21.56±1.04 | 0.65 / 0.80 / 1.59 |

*Table 2.* **Pruning Time & Speedup** on RTX A6000. Speedup is reported as light/aggressive. The complete results are in Table 14.

| Pruning Time (s) ↓ | LLaMA-8B | LLaMA-13B | Qwen-7B | Qwen-14B |
|---|---|---|---|---|
| Wanda | 0.137 | 0.525 | 0.098 | 0.558 |
| LLM-Pr | 0.340 | 0.593 | 0.266 | 0.599 |
| RIA | 0.083 | 0.275 | 0.078 | 0.290 |
| **Ours-SubspacePath** | **0.039** | **0.060** | **0.027** | **0.068** |

| Speedup ↑ | LLaMA-8B | LLaMA-13B | Qwen-7B | Qwen-14B |
|---|---|---|---|---|
| Wanda | 0.48 / 0.19 | 0.39 / 0.42 | 0.22 / 0.13 | 0.28 / 0.32 |
| LLM-Pr | 0.61 / 0.64 | 0.96 / 0.80 | 0.66 / 0.67 | 0.96 / 0.66 |
| RIA | 0.36 / 0.19 | 0.40 / 0.74 | 0.29 / 0.13 | 0.29 / 0.32 |
| **Ours-SubspacePath** | **1.41/1.35** | **3.22/1.51** | **2.01/0.88** | **1.38/1.29** |

scenario breadth $c(s)$ modulate the effective keep-ratio:

$$K(s) = |\mathcal{W}| + \min\left\{ K_B - |\mathcal{W}|, \left\lceil \left(\rho_{\min} + \eta\, c(s)\right)\left(|\mathcal{H}| - |\mathcal{W}|\right)\right\rceil \right\}. \quad (9)$$

where $\rho_{\min}$ is a fixed small floor keep-ratio. Thus, small $c(s)$ in narrow scenarios, fewer heads are kept; broad scenarios kept more, while $\eta$ globally scales pruning strength. After determining the keep budget, we always retain whitelist $\mathcal{W}$ and select remaining $K(s) - |\mathcal{W}|$ heads with the largest score$_{\ell,h}(s)$ (ties broken deterministically), producing a binary sign $m_s \in \{0,1\}^{|\mathcal{H}|}$ and an executable pruned instance $f_\theta^{(m_s)}$. Algorithm details are in Appendix D.

## 5 Experiment and Results

### 5.1 Experimental Setup

Our evaluation is designed to rigorously assess the efficacy and robustness of *SubspacePath Pruner*, especially under cross-domain and cross-dataset generalization with budget constraints, and explore the trade-offs between pruning ratio, inference efficiency, and task performance.

**Models.** We select four representative general-purpose LLM bases with distinct scales: *(i)* **LLaMA-2-13B-chat** and *(ii)* **LLaMA-3.1-8B-Instruct** of Transformer-based pretrained language models (Touvron et al., 2023). *(iii)* **Qwen2.5-14B-Instruct** and *(iv)* **Qwen2.5-7B-Instruct** including dense decoder models at multiple parameter scales (Yang et al., 2024c). Model versions, metrics, and configuration details are provided in Appendix I.

**Datasets.** We evaluate on XDomainBench split into selected-domain, out-of-domain, and cross-domain sets (Gong et al., 2026), CommonsenseQA (Talmor et al., 2019), Natural Questions (Kwiatkowski et al., 2019), and ARC (Clark et al., 2018). We conduct *offline* preparation only on a subset of the selected-domain portion of XDomainBench, with *online* inference and testing performed on the remaining portions and other datasets. Dataset details, preprocessing procedures, and statistics are in Appendix H.

**Baseline Methods.** We compare SUBSPACEPATH PRUNER with several representative inference-time training-free compression methods under the same pruning ratios, with unpruned dense models. For unstructured methods, we selected DaSS (Guo et al., 2024). For structured methods, we selected Wanda (Sun et al., 2023) and LLM-Pruner (Ma et al., 2023). For activation-/probe-guided methods, we selected RIA (Zhang et al., 2024b) and Probe Pruning (Le et al., 2025). Detailed descriptions of these baselines are provided in Appendix J.

**Protocol** DBS axis basis $\mathbb{U}_\star$ is shared across all evaluated

*Table 3.* **Efficiency metrics for LLaMA-2-13B.** Left to right: XDomainBench (averaged across Selected/OOD/Cross); CommonsenseQA; Natural Questions; ARC. Columns: Type (Dense/Light/Heavy), Ret. (head retention ratio), Mem. (memory in GB), Spd. (speedup).

| Type | Ret. | Mem. | Spd. | Type | Ret. | Mem. | Spd. | Type | Ret. | Mem. | Spd. | Type | Ret. | Mem. | Spd. |
|------|------|------|------|------|------|------|------|------|------|------|------|------|------|------|------|
| Dense | — | 13.0 | 1.00 | Dense | — | 13.0 | 1.00 | Dense | — | 13.0 | 1.00 | Dense | — | 13.0 | 1.00 |
| Light | 80.7% | 12.1 | 1.26 | Light | 89.9% | 12.7 | 1.08 | Light | 81.5% | 12.4 | 1.08 | Light | 80.2% | 12.4 | 2.21 |
| Heavy | 72.2% | 11.7 | 1.24 | Heavy | 69.4% | 12.0 | 1.09 | Heavy | 69.5% | 12.0 | 0.97 | Heavy | 68.3% | 12.0 | 1.76 |

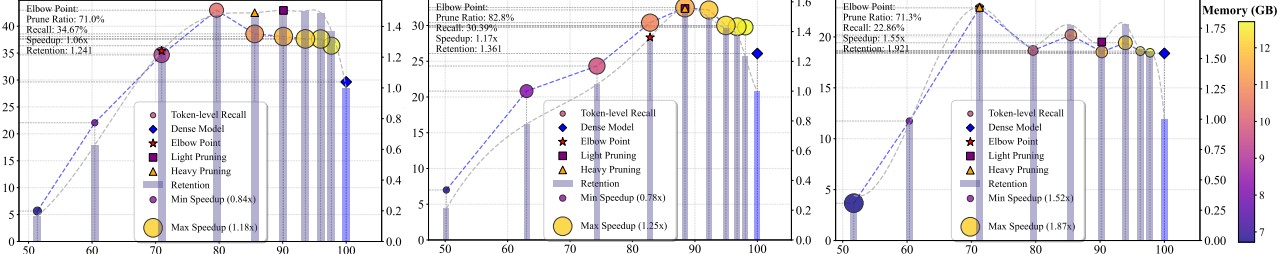

*Figure 5.* **Pruning ratio vs Performance for LLaMA-2-13B.** X-axis: Head retention (%); Left/right Y-axis: Recall/ Retention. From left to right: selected, out of, and cross domain. Extended trade-off curves for all models and datasets are provided in Figures 16 and 17.

backbones for justice. For pruning ratio selection, the main results tables report light and heavy pruning points, while ablation studies use elbow points. Detailed pruning point $\eta$ selection criteria and analysis are provided in Appendix O.5. Additional protocol details are provided in Appendix I.

## 5.2 Experiment Results

### 5.2.1 Main results.

As presented in Table 1, across XDomainBench data splits and various cross-dataset, SUBSPACEPATH consistently achieves stronger robustness under distribution shifts while maintaining competitive performance, highlighting that our method yields a better robustness–efficiency trade-off than static, globally-ranked pruning baselines. At moderate pruning, our method achieves **43.0 / 32.5 / 20.2** Recall on XDomainBench (Selected/OOD/Cross) vs. **29.6 / 26.1 / 18.4** dense, with retention values exceeding 1.0, indicating efficiency-adjusted gains. On cross-dataset tests, we achieve **33.7** Recall on NQ (vs. 30.3 dense) with retention **1.20**, while maintaining competitive performance on CSQA and ARC. Under aggressive pruning, cross-domain performance remains above dense (**34.7 / 30.4 / 22.9**), while baselines show retention values often below 0.5 in OOD scenarios, demonstrating sensitivity to distribution shifts. The larger SEM in some splits reflects a trade-off of scenario-adaptive pruning: clear scenario-start domain mixtures yield more transferable masks, whereas high-entropy or ambiguous mixtures lead to less uniform gains. Additional results on other backbones, pruning ratios, pruning time, task-type breakdown, extended trade-off curves, and case study are provided in Appendix N, O and Q.

The improvement is actually arises from scenario-conditioned reorganization of the active computation. Dense models retain many heads whose signals are useful in other domains but can interfere with the current DBS-aligned sce-

nario coordinate. By compiling a mask from this coordinate, SUBSPACEPATH PRUNER suppresses scenario-irrelevant or cross-domain-conflicting pathways while preserving heads coupled to the active domain mixture. This turns pruning into an inference-time structural regularizer: it reduces latent pathway competition in the residual stream and makes evidence aggregation more coherent for the current scenario.

### 5.2.2 Efficiency Analysis.

Efficiency metrics for LLaMA-2-13B are reported in Table 3. Light pruning (80–90% retention) achieves 1.08–1.26× speedup with 7–8% memory reduction, while heavier pruning (68–72% retention) yields 12–13% memory savings with dataset-dependent speedup (0.97–1.76×). ARC obtains the largest acceleration (2.21× at light and 1.76× at heavy pruning), suggesting that structured head removal is most effective when redundant pathway usage is high and the backend can exploit the reduced attention computation.

Pruning time across methods is compared in Table 2. The key efficiency advantage of SUBSPACEPATH PRUNER comes from separating reusable offline preparation from lightweight online compilation: DBS axes, probes, and cached head scores are built once, while scenario-specific masks are compiled in only 0.027–0.068s. This makes the method suitable for multi-turn inference settings, where the same pruned subnetwork is reused across a coherent scenario. By contrast, methods requiring heavy per-model or per-instance pruning, such as SparseGPT (Frantar & Alistarh, 2023), are less aligned with inference-time specialization. Detailed results across pruning ratios are provided in Tables 12, 13 and 15.

We also emphasize that head pruning does not guarantee monotonic wall-clock speedup: latency depends on backend support, kernel efficiency, sequence length, and mask overhead. The mask compilation cost is incurred once per sce-

*Table 4.* **Ablations for subspace–pathway coupling on LLaMA-2-13B** with pruning ratio at the elbow point for each dataset type.

| Ablation | XDB (Selected) | XDB (OOD) | XDB (Cross) | CSQA | NQ | ARC |
|---|---|---|---|---|---|---|
| | $\eta = 0.3$ | $\eta = 0.4$ | $\eta = 0.3$ | $\eta = 0.4$ | $\eta = 0.3$ | $\eta = 0.4$ |
| Dense | $29.6_{\pm 1.45}$ | $26.1_{\pm 1.10}$ | $18.4_{\pm 0.74}$ | $22.27_{\pm 1.19}$ | $30.25_{\pm 1.38}$ | $23.87_{\pm 1.11}$ |
| Full: DBS + PSP | $34.7_{\pm 2.34}$ | $30.4_{\pm 1.87}$ | $22.9_{\pm 3.16}$ | $19.43_{\pm 1.14}$ | $33.66_{\pm 1.35}$ | $25.62_{\pm 1.12}$ |
| w/o DBS selection (random axes) | $1.8_{\pm 1.0}$ | $1.4_{\pm 0.8}$ | $0.9_{\pm 0.5}$ | $0.82_{\pm 0.21}$ | $0.40_{\pm 0.11}$ | $1.16_{\pm 0.19}$ |
| w/o whitelist ($\mathcal{W} = \emptyset$) | $22.4_{\pm 1.9}$ | $0.4_{\pm 0.2}$ | $20.2_{\pm 2.0}$ | $0.10_{\pm 0.10}$ | $0.12_{\pm 0.06}$ | $0.23_{\pm 0.09}$ |
| w/o multi-domain mixing | $29.5_{\pm 2.0}$ | $23.5_{\pm 1.5}$ | $19.0_{\pm 2.1}$ | $7.07_{\pm 0.73}$ | $10.68_{\pm 0.85}$ | $17.04_{\pm 0.94}$ |

nario and amortized across subsequent turns. We therefore report raw Speedup separately from Retention: Speedup measures actual acceleration, while Retention summarizes whether the accuracy–latency trade-off is worthwhile.

### 5.2.3 Trade-off Analysis.

The pruning ratio–performance trade-off for LLaMA-2-13B is shown in Figure 5. Elbow points occur at 71%, 83%, and 71% head retention for selected, OOD, and cross-domain splits, respectively. These elbows indicate the regime where pruning removes enough interfering pathways to improve specialization, but still preserves the head capacity required for stable reasoning. Cross-domain scenarios benefit the most: at 75% retention, the pruned model reaches **22.9%** Recall with **1.55×** speedup (Retention **1.92**) versus 10% dense Recall, yielding a **2.3×** improvement. Selected-domain and OOD splits also improve (**34.7%** vs. 28% and **30.4%** vs. 20% dense, respectively), but with smaller margins.

This pattern supports the central mechanism of our method. The gain is not merely caused by using fewer heads; rather, pruning is most beneficial when the dense model contains competing domain pathways that interfere during evidence aggregation. By compiling a mask aligned with the active scenario subspace, SUBSPACEPATH PRUNER suppresses cross-domain-conflicting heads while retaining pathways coupled to the dominant semantic mixture. Retention exceeding 1.0 at all elbow points therefore reflects not only an efficiency gain, but also a better organization of active computation for the current scenario.

### 5.3 Ablation Study

Ablations on LLaMA-2-13B are reported in Table 4. Each component corresponds to a different link in the *subspace → pathway* pipeline. Removing DBS selection causes catastrophic drops, showing that pruning cannot generalize without stable semantic axes. Removing whitelist heads severely degrades OOD and cross-dataset performance, indicating that globally reliable heads remain necessary for preserving general reasoning capacity under shift. Removing mixing reduces cross-dataset Recall by 3–15 points, confirming that complex scenarios require composing multiple semantic directions rather than selecting a single dominant domain.

These ablations show that the method is not a collection of independent heuristics. DBS defines a stable semantic coor-

*Table 5.* Matched-budget comparison with routed sparse models on Selected/OOD/Cross splits.

| Method | Active | Recall (%) | Inference Time (s) | Loaded |
|---|---|---|---|---|
| Qwen2.5-7B+Ours | ∼6.8B | 46.4/41.0/27.2 | 0.12/0.30/1.26 | 7.6B |
| Mixtral top-1 | ∼7.3B | 47.8/39.2/22.9 | 2.11/3.56/7.84 | 46.7B |
| MiniMax-M2 | ∼10.0B | 45.1/33.7/21.0 | 2.30/3.65/8.30 | 230.0B |
| Qwen2.5-14B+Ours | ∼13.6B | 47.8/44.1/31.3 | 0.23/0.53/1.25 | 14.0B |
| Mixtral top-2 | ∼12.9B | 50.9/40.8/24.0 | 2.70/4.24/9.86 | 46.7B |
| Qwen2-57B-A14B | ∼14.0B | 49.4/41.9/25.1 | 2.58/4.08/9.12 | 57.0B |

dinate system, probes estimate scenario alignment, whitelist heads preserve task-general computation, and multi-domain mixing converts semantic ambiguity into a controlled pathway budget. The performance collapse under component removal therefore supports the core claim: robust pruning requires preserving the coupling between representation-side subspaces and executable attention pathways. Results for other models are in Table 16.

### 5.4 Comparison with Routed Sparse Models

We compare Qwen2.5 dense backbones (Yang et al., 2024b) with representative routed sparse models, including MiniMax-M2 (MiniMax, 2025), Mixtral-8x7B (Jiang et al., 2024), and Qwen2-57B-A14B (Yang et al., 2025). Table 5 shows that dense pruning and MoE target different deployment regimes: MoE models remain strong in raw accuracy, but require larger loaded backbones and token-level routing. At comparable active budgets, Qwen2.5-14B with SUBSPACEPATH PRUNER achieves higher OOD/cross-domain Recall than Mixtral top-2 and Qwen2-57B-A14B, with lower loaded parameters and latency. These results suggest that the proposed pipeline is practically useful when the deployment objective is not only active-parameter count, but also fixed dense-checkpoint specialization, loaded-memory footprint, and reusable scenario-level execution.

## 6 Conclusion

We introduced SUBSPACEPATH PRUNER, an inference-time pruning framework that capture reusable, budget-bounded pathway pruning for scenarios. Our key insight is *subspace–pathway coupling*: DBS constructs stable domain axes in embedding space, while PSP bridges these axes to sparse head-level pathways via probe signals, enabling interpretable, robust structured pruning under strict inference budgets. Experiments across XDomainBench and cross-dataset tests demonstrate improved robustness to distribution shifts, competitive performance, and measurable gains in inference efficiency.

## Acknowledgements

This research is supported by the RIE2025 Industry Alignment Fund – Industry Collaboration Projects (IAF-ICP) (Award I2301E0026), administered by A*STAR, as well as supported by Alibaba Group and NTU Singapore through Alibaba-NTU Global e-Sustainability CorpLab (ANGEL). This research is also supported by the Ministry of Education, Singapore, under its Academic Research Fund Tier 2 (Award MOE-T2EP20125-0005). Yikun Hou was supported by the Wallenberg AI, Autonomous Systems, and Software Program (WASP), funded by the Knut and Alice Wallenberg Foundation. Partial computations were enabled by the Berzelius resource provided by the Knut and Alice Wallenberg Foundation at the National Supercomputer Centre.

## Impact Statement

This paper presents work whose goal is to advance the field of Machine Learning. There are many potential societal consequences of our work, none which we feel must be specifically highlighted here.

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

# Appendix Overview

This appendix provides algorithmic details, mathematical and theoretical analysis, experimental setup and implementation details, additional experimental results, and limitations discussion for SUBSPACEPATH PRUNER. To keep the appendix navigable, we organize it into five *Parts* according to the paper's pipeline.

**Part A: Algorithmic Details.**    This part extends methodology in the main paper with complete algorithm specifications and implementation details. We provide a complete pipeline algorithm that integrates offline and online stages, followed by detailed sub-algorithms for each component.

- Appendix A: Complete algorithm pipeline (online+offline flow) that integrates Domain-Basis Synthesis (DBS) and Probe-based Scenario Pruning (PSP) stages.
- Appendix B: DBS detailed algorithm, including stopword filtering, PCA projection, and axis subset selection.
- Appendix C: PSP offline preparation, including probe training, head-wise importance computation, and whitelist identification.
- Appendix D: PSP online compilation, including scenario-level domain mixture aggregation, head scoring, and mask compilation.

**Part B: Mathematical Properties and Theoretical Analysis.**    This part provides mathematical foundations, theoretical analysis, and formal arguments that support the design choices in our method.

- Appendix E: Knowledge sparsity and low-rank subspace structure (theoretical foundation), including domain embedding rank bounds, compression via subspace projection, and OOD/cross-domain representation.
- Appendix F: Theoretical explanations for two counter-intuitive empirical observations: pruning-induced performance improvement (noise reduction theory) and cross-domain pruning capacity (subspace union structure).
- Appendix G: Mathematical properties of configuration choices, including PCA dimension selection, orthogonality and coverage metrics, head importance computation, knowledge breadth and scoring, budget constraints, and consistency properties.

**Part C: Experimental Setup and Implementation Details.**    This part expands the experimental setup section with dataset processing, model configuration, evaluation protocol, hyperparameters, and prompt templates.

- Appendix H: Dataset versions, preprocessing procedures, and statistics.
- Appendix I: Model configuration and detailed evaluation protocol.
- Appendix J: Detailed descriptions of baseline pruning methods used for comparison.
- Appendix K: Hyperparameter configurations for DBS, PSP (offline and online), and text generation.
- Appendix L: Prompt templates for different task types (multiple_choice, factual, reasoning, code), using official model chat templates.

**Part D: Additional Experimental Results and Analysis.**    This part provides pipeline step-wise effects, additional results, task-type breakdown, case studies, and efficiency analysis.

- Appendix M: Detailed effects of each pipeline step (DBS, PSP offline, PSP online) with intermediate results and analysis.
- Appendix N: Additional results for other models (LLaMA-3.1-8B, Qwen2.5-7B, Qwen2.5-14B), different pruning ratios, pruning time comparison, and cross-domain/cross-dataset analysis. Detailed analysis by task type (multiple_choice, factual, reasoning, code) across models (dense, baseline methods, our method).
- Appendix O: Efficiency analysis including speedup ratios, memory usage, and compilation overhead with detailed tables.
- Appendix P: Ablation studies for other models (LLaMA-3.1-8B, Qwen2.5-7B, Qwen2.5-14B), demonstrating the consistent effectiveness of DBS, whitelist, and multi-domain mixing across different model architectures.
- Appendix Q: Case studies showing detailed inference process for out-of-domain, cross-domain, and cross-dataset samples.

**Part E: Limitations and Discussion.**    This part discusses limitations, future work, and additional considerations.

- Appendix R: Method, experimental, and inference limitations.
- Appendix S: Future work directions.

# Part A: Algorithmic Details.

## A   Complete Algorithm Pipeline

This section presents the complete SUBSPACEPATH PRUNER algorithm that integrates both offline and online stages. Algorithm 1 provides a high-level overview of the entire pipeline, showing how Domain-Basis Synthesis (DBS) and Probe-based Scenario Pruning (PSP) components work together to compile scenario-level pathway masks.

The algorithm takes as input a frozen foundation model $f_\theta$, input-only pools $\{\mathcal{P}_k\}_{k\in\mathcal{K}}$ for offline preparation, and a deployment scenario $s$ with inputs $\{x_t\}_{t=1}^{T_s}$ and budget constraint $B$. The offline stage (Steps 1–2) is executed once and can be reused across multiple scenarios and datasets, while the online stage (Step 3) is executed once per scenario at scenario start.

The offline stage constructs a stable embedding-space coordinate system (DBS) and prepares lightweight diagnostic tools (PSP offline) that enable scenario-start compilation. The online stage uses these tools to diagnose the scenario's knowledge requirements and compile an appropriate pathway mask under the budget constraint. All probe evaluations and mask compilation happen on the frozen backbone $f_\theta$ without any parameter updates, ensuring training-free inference.

## B   Domain-Basis Synthesis (DBS) Detailed Algorithm

This section provides the complete DBS algorithm that constructs a selected axis basis $\mathbb{U}_\star$ in embedding space. Algorithm 2 integrates stopword filtering, PCA projection, and axis subset selection into a unified procedure.

The algorithm processes input-only pools $\{\mathcal{P}_k\}_{k\in\mathcal{K}}$ to extract domain-specific embedding patterns. It first applies stopword filtering to focus on informative tokens, then performs PCA to reduce dimensionality while preserving between-group separation structure. Finally, it selects a compact subset of axes that balances orthogonality and coverage.

**Stopword filtering and embedding.**   We use a fixed stopword list $\mathcal{S}$ to filter out generic function words and domain-agnostic tokens. For each input $x$, we obtain $\tilde{x}$ by removing tokens in $\mathcal{S}$, preserving token repetition which contributes to density signatures in the embedding space. The embedding function $\phi$ maps filtered sequences to $d$-dimensional vectors $\mathbf{e}(x) \in \mathbb{R}^d$.

**PCA projection.**   We construct a shared PCA projection matrix $P \in \mathbb{R}^{d\times d'}$ with $d' = M - 1$ by running PCA on the pooled embeddings from all pools. This choice follows a standard multi-class geometry principle: for $M$ groups, the between-group scatter (spanned by class mean differences) has rank at most $M - 1$, so a $(M - 1)$-dimensional subspace is sufficient to preserve all degrees of freedom for separating $M$ groups (Fisher, 1936; Fukunaga, 1990; Hastie, 2009). Standard centering is applied before PCA. We project embeddings to the PCA space: $\tilde{\mathbf{e}}(x) = P^\top \mathbf{e}(x) \in \mathbb{R}^{d'}$.

---

**Algorithm 1** SubspacePath Pruner: Complete Pipeline

---

**Require:** Frozen model $f_\theta$, input-only pools $\{\mathcal{P}_k\}_{k\in\mathcal{K}}$, scenario $s$ with inputs $\{x_t\}_{t=1}^{T_s}$, budget $B$
**Ensure:** Pruned instance $f_\theta^{(m_s)}$
 1: **Offline Stage** (executed once, reusable)
 2:     Step 1: Construct axis basis $\mathbb{U}_\star$ via DBS (Algorithm 2)
 3:         Input: $\{\mathcal{P}_k\}_{k\in\mathcal{K}}$
 4:         Output: Selected axis basis $\mathbb{U}_\star = \{\mathcal{U}_k\}_{k\in\mathcal{K}_\star}$, PCA projection $P$
 5:     Step 2: Prepare PSP offline components (Algorithm 3)
 6:         Input: $f_\theta$, $\mathbb{U}_\star$, $P$, $\{\mathcal{P}_k^{\text{tr}}\}_{k\in\mathcal{K}_\star}$
 7:         Output: Trained probes $\{p_{\ell,k}\}$, cached importance $\{I_{\ell,h,k}\}$, whitelist $\mathcal{W}$
 8: **Online Stage** (executed once per scenario)
 9:     Step 3: Compile scenario-level mask $m_s$ via PSP online (Algorithm 4)
10:         Input: scenario prefix $\{x_t\}_{t=1}^{T_0}$, probes, importance, whitelist, budget $B$
11:         Output: Pathway mask $m_s$, pruned instance $f_\theta^{(m_s)}$
12: **inference**
13:     Reuse $f_\theta^{(m_s)}$ for all subsequent inputs in scenario $s$

---

---

**Algorithm 2** Domain-Basis Synthesis (DBS)

---

**Require:** input-only pools $\{\mathcal{P}_k\}_{k \in \mathcal{K}}$, stopword list $\mathcal{S}$, embedding function $\phi$
**Ensure:** Selected axis basis $\mathbb{U}_\star$, PCA projection matrix $P$
 1: **Step 1: Stopword filtering and embedding**
 2: **for** each pool $k \in \mathcal{K}$ **do**
 3:      **for** each input $x \in \mathcal{P}_k$ **do**
 4:         $\tilde{x} \leftarrow$ remove tokens in $\mathcal{S}$ from $x$
 5:         $\mathbf{e}(x) \leftarrow \phi(\tilde{x}) \in \mathbb{R}^d$
 6:      **end for**
 7: **end for**
 8: **Step 2: PCA projection**
 9: Pool all embeddings: $\mathcal{E}_{\text{pool}} \leftarrow \cup_{k \in \mathcal{K}} \{\mathbf{e}(x) : x \in \mathcal{P}_k\}$
10: Compute PCA on $\mathcal{E}_{\text{pool}}$ with $d' = M - 1$ principal components
11: $P \leftarrow$ top-$d'$ principal directions (stacked as columns)
12: Project embeddings: $\tilde{\mathbf{e}}(x) \leftarrow P^\top \mathbf{e}(x) \in \mathbb{R}^{d'}$ for all $x$
13: **Step 3: Construct candidate axes**
14: **for** each pool $k \in \mathcal{K}$ **do**
15:      $\boldsymbol{\mu}_k \leftarrow \frac{1}{|\mathcal{P}_k|} \sum_{x \in \mathcal{P}_k} \tilde{\mathbf{e}}(x)$
16:      $\mathbf{u}_k \leftarrow \frac{\boldsymbol{\mu}_k}{\|\boldsymbol{\mu}_k\|_2}$ {Normalized centroid}
17:      $\mathcal{U}_k \leftarrow \text{span}(\mathbf{u}_k)$
18: **end for**
19: $\mathbb{U} \leftarrow \{\mathcal{U}_k\}_{k \in \mathcal{K}}$
20: **Step 4: Subset selection**
21: Compute global covariance: $\Sigma \leftarrow \text{Cov}(\{\tilde{\mathbf{e}}(x) : x \in \cup_k \mathcal{P}_k\})$
22: Initialize candidate subsets and score curve
23: **for** $n = 1$ to $|\mathcal{K}|$ **do**
24:      Find best subset $\mathcal{K}'_n$ of size $n$ maximizing $J(\mathcal{K}') = \lambda_{\text{ortho}} \cdot \text{Ortho}(\mathcal{K}') + \lambda_{\text{cov}} \cdot \text{Cover}(\mathcal{K}')$
25:      $J_n \leftarrow J(\mathcal{K}'_n)$
26: **end for**
27: Detect elbow point in curve $\{J_n\}$ using knee detection
28: Perform local search around elbow to finalize $\mathcal{K}_\star$
29: $\mathbb{U}_\star \leftarrow \{\mathcal{U}_k\}_{k \in \mathcal{K}_\star}$
30: **Output:** $\mathbb{U}_\star$, $P$

---

**Axis construction.** For each pool $k \in \mathcal{K}$, we compute the centroid in the PCA space as the mean of all embeddings from pool. We treat axis as the rank-1 subspace $\mathcal{U}_k = \text{span}(\mathbf{u}_k)$ and denote the collection of candidate axes by $\mathbb{U} = \{\mathcal{U}_k\}_{k \in \mathcal{K}}$.

**Subset selection.** We select a compact axis set $\mathcal{K}_\star \subseteq \mathcal{K}$ by balancing orthogonality and coverage. For a subset $\mathcal{K}'$, we define orthogonality and coverage as in Equations (1) and (2), where $\Sigma$ is the global covariance of all pooled embeddings in the PCA space, $U(\mathcal{K}')$ is the matrix whose columns are $\{\mathbf{u}_k\}_{k \in \mathcal{K}'}$, and $\Pi(\mathcal{K}')$ is the projection matrix onto $\text{span}(U(\mathcal{K}'))$ (with orthonormalization implied). We maximize the combined score as in Equation (3). In practice, we first run a fast greedy screening to obtain a score curve $J_n$ over subset size $n$, detect an elbow point using knee detection (Satopaa et al., 2011), and perform a local search around the elbow to finalize $\mathcal{K}_\star$. The selected basis is $\mathbb{U}_\star = \{\mathcal{U}_k\}_{k \in \mathcal{K}_\star}$.

## C  PSP Offline Preparation Detailed Algorithm

This section provides the complete PSP offline preparation algorithm that trains probes, computes head-wise importance, and identifies whitelist heads. Algorithm 3 integrates all offline components into a unified procedure.

The algorithm takes the frozen model $f_\theta$, selected axis basis $\mathbb{U}_\star$, PCA projection $P$, and training pools $\{\mathcal{P}_k^{\text{tr}}\}_{k \in \mathcal{K}_\star}$ as input. It outputs trained probes, cached importance statistics, and whitelist heads that will be used during online compilation.

**Layer-wise probe training.** For each layer $\ell$ and each selected domain $k \in \mathcal{K}_\star$, we train a linear probe on the post-attention residual slice $r^{(\ell)}(\tilde{x}) \in \mathbb{R}^d$, where $\tilde{x}$ denotes the stopword-filtered input as in DBS. Let $\mathbf{g}_{\ell,k} \in \mathbb{R}^d$ denote the probe

---

**Algorithm 3** PSP Offline Preparation

---

**Require:** Frozen model $f_\theta$, selected axes $\mathbb{U}_\star$, PCA projection $P$, training pools $\{\mathcal{P}_k^{\mathrm{tr}}\}_{k \in \mathcal{K}_\star}$
**Ensure:** Trained probes $\{p_{\ell,k}\}$, cached importance $\{I_{\ell,h,k}\}$, whitelist $\mathcal{W}$

  1: **Step 1: Train layer-wise probes**
  2: **for** each layer $\ell = 1$ to $L$ **do**
  3:     **for** each domain $k \in \mathcal{K}_\star$ **do**
  4:         Train linear probe $\mathbf{g}_{\ell,k}$ on residual slices $\{r^{(\ell)}(\tilde{x}) : x \in \cup_{j \in \mathcal{K}_\star} \mathcal{P}_j^{\mathrm{tr}}\}$
  5:         Use 1-vs-rest labels: $y_k(x) = 1$ if $x \in \mathcal{P}_k^{\mathrm{tr}}$, else 0
  6:         Minimize: $\mathcal{L}_{\ell,k} = -\mathbb{E}_x[y_k(x) \log p_{\ell,k}(x) + (1 - y_k(x)) \log(1 - p_{\ell,k}(x))]$
  7:         where $p_{\ell,k}(x) = \sigma(\mathbf{g}_{\ell,k}^\top r^{(\ell)}(\tilde{x}))$
  8:     **end for**
  9: **end for**
10: **Step 2: Compute head-wise importance**
11: **for** each layer $\ell = 1$ to $L$ **do**
12:     **for** each head $h = 1$ to $H_\ell$ **do**
13:         **for** each domain $k \in \mathcal{K}_\star$ **do**
14:             **for** each input $x \in \mathcal{P}_k^{\mathrm{tr}}$ **do**
15:                 Compute residual write: $w_{\ell,h}(x) = r^{(\ell)}(x) - r_{-h}^{(\ell)}(x)$
16:                 Project: $\tilde{w}_{\ell,h}(x) = P^\top w_{\ell,h}(x) \in \mathbb{R}^{d'}$
17:             **end for**
18:             $I_{\ell,h,k} \leftarrow \mathbb{E}_{x \sim \mathcal{P}_k^{\mathrm{tr}}} \left[ \frac{(\mathbf{u}_k^\top \tilde{w}_{\ell,h}(x))^2}{\|\tilde{w}_{\ell,h}(x)\|_2^2 + \epsilon} \right]$
19:         **end for**
20:     **end for**
21: **end for**
22: **Step 3: Identify whitelist heads**
23: **for** each head $(\ell, h) \in \mathcal{H}$ **do**
24:     $\sigma_{\ell,h} \leftarrow \mathrm{Std}(\{I_{\ell,h,k} : k \in \mathcal{K}_\star\})$
25:     $\mu_{\ell,h} \leftarrow \frac{1}{|\mathcal{K}_\star|} \sum_{k \in \mathcal{K}_\star} I_{\ell,h,k}$
26:     $\mathrm{CV}_{\ell,h} \leftarrow \sigma_{\ell,h}/(\mu_{\ell,h} + \epsilon)$
27: **end for**
28: $\mathcal{W} \leftarrow \{(\ell, h) : (\sigma_{\ell,h} \leq \sigma_{\mathrm{thresh}} \wedge \mu_{\ell,h} \geq \mu_{\mathrm{thresh}}) \vee (\mathrm{CV}_{\ell,h} \leq \mathrm{CV}_{\mathrm{thresh}} \wedge \mu_{\ell,h} \geq \mu_{\mathrm{thresh}})\}$
29: Validate whitelist using Mann-Whitney U test
30: **Output:** $\{p_{\ell,k}\}, \{I_{\ell,h,k}\}, \mathcal{W}$

---

weight vector (bias omitted for simplicity) and $z_{\ell,k}(x) = \mathbf{g}_{\ell,k}^\top r^{(\ell)}(\tilde{x})$ be the probe logit. We output a 1-vs-rest probability $p_{\ell,k}(x) = \sigma(z_{\ell,k}(x))$, where $\sigma$ is the sigmoid function (Alain & Bengio, 2016; Belinkov, 2022). Training uses labels over the union set $\cup_{k \in \mathcal{K}_\star} \mathcal{P}_k^{\mathrm{tr}}$: each example from $\mathcal{P}_k^{\mathrm{tr}}$ is treated as positive for $k$ and negative for $k' \neq k$ (independent 1-vs-rest logistic regressors). The training objective is the binary cross-entropy loss as in Equation (4), where $y_k(x) = 1$ if $x \in \mathcal{P}_k^{\mathrm{tr}}$, and 0 otherwise. During inference, we interpret $\{p_{\ell,k}(x)\}_{k \in \mathcal{K}_\star}$ as independent relevance scores rather than a simplex distribution, since real inputs can mix multiple knowledge axes.

**Head-wise importance computation.** We define a head's contribution on the post-attention residual write. Let $w_{\ell,h}(x) \in \mathbb{R}^d$ denote the residual write-back vector attributable to head $(\ell, h)$ for input $x$ (i.e., the head-wise component added into the residual stream at layer $\ell$; Elhage et al., 2021). We compute $w_{\ell,h}(x)$ using the difference method: let $r^{(\ell)}(x)$ denote the full post-attention residual at layer $\ell$, obtained by applying the output projection o_proj to the concatenated outputs of all heads. Let $r_{-h}^{(\ell)}(x)$ denote the residual obtained by applying o_proj to the concatenated outputs of all heads except head $h$. Then $w_{\ell,h}(x) = r^{(\ell)}(x) - r_{-h}^{(\ell)}(x)$. This ensures that the sum of all head contributions equals the full residual, and avoids bias-related artifacts from the output projection. We map this write into the shared PCA space: $\tilde{w}_{\ell,h}(x) = P^\top w_{\ell,h}(x) \in \mathbb{R}^{d'}$. Then the domain-specific head importance is defined as the expected axis-aligned energy as in Equation (5), where $\epsilon$ is a small constant for numerical stability. Intuitively, $I_{\ell,h,k}$ is large if head $(\ell, h)$ consistently writes along the selected embedding-axis $\mathbf{u}_k$ on domain-$k$ inputs, and small if it is irrelevant. We compute $\{I_{\ell,h,k}\}$ once offline and cache them for inference-time compilation.

---

**Algorithm 4** PSP Online Compilation

---

**Require:** scenario prefix $\{x_t\}_{t=1}^{T_0}$, trained probes $\{p_{\ell,k}\}$, cached importance $\{I_{\ell,h,k}\}$, whitelist $\mathcal{W}$, temperatures $\{\tau_\ell\}$, budget $B$

**Ensure:** Pathway mask $m_s$, pruned instance $f_\theta^{(m_s)}$

 1: **Step 1: Aggregate probe outputs**
 2: **for** each layer $\ell = 1$ to $L$ **do**
 3:   **for** each domain $k \in \mathcal{K}_\star$ **do**
 4:     $\bar{p}_{\ell,k}(s) \leftarrow \frac{1}{T_0} \sum_{t=1}^{T_0} \hat{p}_{\ell,k}(\tilde{x}_t)$
 5:     where $\hat{p}_{\ell,k}(\tilde{x}_t) = \sigma(z_{\ell,k}(\tilde{x}_t)/\tau_\ell)$
 6:   **end for**
 7: **end for**
 8: **Step 2: Compute scenario-level domain mixture**
 9: **for** each domain $k \in \mathcal{K}_\star$ **do**
10:   $q_k(s) \leftarrow \frac{1}{L} \sum_{\ell=1}^{L} \bar{p}_{\ell,k}(s)$
11: **end for**
12: **Step 3: Compute knowledge breadth**
13: $H(q(s)) \leftarrow -\sum_{k \in \mathcal{K}_\star} \frac{q_k(s)}{\sum_{j \in \mathcal{K}_\star} q_j(s)} \log \left( \frac{q_k(s)}{\sum_{j \in \mathcal{K}_\star} q_j(s)} \right)$
14: $c(s) \leftarrow \frac{H(q(s))}{\log |\mathcal{K}_\star|} \in [0, 1]$
15: **Step 4: Compute effective domain similarities**
16: **for** each domain $k \in \mathcal{K}_\star$ **do**
17:   Compute $s_k^{\text{eff}}$ using non-linear transformation (Equation (45))
18: **end for**
19: **Step 5: Score heads**
20: **for** each head $(\ell, h) \in \mathcal{H}$ **do**
21:   $\text{score}_{\ell,h}(s) \leftarrow \sum_{k \in \mathcal{K}_\star} s_k^{\text{eff}} \cdot I_{\ell,h,k}$
22: **end for**
23: **Step 6: Determine keep budget**
24: Map budget $B$ to head budget $K_B$ (hardware-dependent mapping)
25: $K(s) \leftarrow |\mathcal{W}| + \min\{K_B - |\mathcal{W}|, \lceil (\rho_{\min} + \eta \cdot c(s))(|\mathcal{H}| - |\mathcal{W}|) \rceil\}$
26: **Step 7: Compile mask**
27: Initialize $m_s$: set $m_s[\ell, h] = 1$ for all $(\ell, h) \in \mathcal{W}$
28: Select top $(K(s) - |\mathcal{W}|)$ heads by $\text{score}_{\ell,h}(s)$, set $m_s[\ell, h] = 1$
29: Set $m_s[\ell, h] = 0$ for all other heads
30: **Step 8: Create pruned instance**
31: $f_\theta^{(m_s)} \leftarrow$ model with heads disabled according to $m_s$
32: **Output:** $m_s, f_\theta^{(m_s)}$

---

**Whitelist identification.** We identify a set of always-retained heads $\mathcal{W} \subseteq \mathcal{H}$ to preserve general-purpose capacity under budget. We select a set of domain-invariant *backbone heads* by statistical significance testing. Specifically, for each head $(\ell, h)$, we compute its importance variance across domains: $\sigma_{\ell,h} = \text{Std}(\{I_{\ell,h,k} : k \in \mathcal{K}_\star\})$ and mean importance: $\mu_{\ell,h} = \frac{1}{|\mathcal{K}_\star|} \sum_{k \in \mathcal{K}_\star} I_{\ell,h,k}$. We also compute the coefficient of variation: $\text{CV}_{\ell,h} = \sigma_{\ell,h}/(\mu_{\ell,h} + \epsilon)$. A head is included in the whitelist if it satisfies the condition in Equation (6), where $\sigma_{\text{thresh}}$, $\mu_{\text{thresh}}$, and $\text{CV}_{\text{thresh}}$ are thresholds determined by statistical significance testing. We validate the whitelist selection using Mann-Whitney U test to confirm that whitelisted heads have significantly lower domain variance than non-whitelisted heads. Statistical validation confirms that whitelisted heads exhibit significantly lower domain variance, validating our domain-invariance criterion.

# D  PSP Online Compilation Detailed Algorithm

This section provides the complete PSP online compilation algorithm that compiles a scenario-level pathway mask from the scenario's initial inputs. Algorithm 4 shows how probe outputs, cached importance, and budget constraints are combined to produce a reusable mask.

The algorithm takes as input a scenario $s$ with prefix inputs $\{x_t\}_{t=1}^{T_0}$ (a small scenario-start prefix), along with the offline-

prepared components (probes, importance, whitelist) and budget $B$. It outputs a binary pathway mask $m_s$ that is cached and reused for all subsequent inputs in the scenario.

**scenario-level domain mixture aggregation.** For each layer $\ell$ and domain $k$, we aggregate the probe outputs over stopword-filtered input $\tilde{x}_t$ as $\bar{p}_{\ell,k}(s) = \frac{1}{T_0} \sum_{t=1}^{T_0} p_{\ell,k}(\tilde{x}_t)$, where $p_{\ell,k}(\tilde{x}_t) = \sigma(z_{\ell,k}(\tilde{x}_t))$ is the probe output. We further define a scenario-level domain mixture by pooling over layers as $q_k(s) = \frac{1}{L} \sum_{\ell=1}^{L} \bar{p}_{\ell,k}(s)$. We then quantify scenario "knowledge breadth" using the normalized entropy as in Equation (7). Low $c(s)$ indicates a narrow (near selected-domain) scenario; high $c(s)$ indicates a broader cross-domain scenario.

**Head scoring.** We score heads based on both alignment and cached importance. To achieve more precise pruning and avoid under-pruning highly relevant domains while reducing over-preservation of less relevant domains, we apply a non-linear transformation to domain similarity probabilities before computing head scores. Specifically, for each domain $k$, we compute an effective domain similarity $s_k^{\text{eff}}$ using the transformation defined in Equation (45). The head score is then computed as in Equation (8), where $s_k^{\text{eff}}$ replaces the raw domain similarity $q_k(s)$. This instantiates the subspace–pathway coupling: probe alignment indicates *what the scenario needs*, and $I_{\ell,h,k}$ indicates *which heads realize that axis*. The non-linear transformation ensures that highly relevant domains (high $q_k(s)$) contribute more to head scores, leading to better preservation of important heads, while less relevant domains (low $q_k(s)$) contribute less, reducing unnecessary head preservation.

**Budget, pruning strength, and mask compilation.** We treat the deployment budget $B$ as a constraint on the masked model cost $\mathcal{C}(f_\theta^{(m_s)}) \leq B$. Since exact cost can be hardware-dependent, we implement the budget via an equivalent head budget $K_B$ (e.g., a mapping from heads to latency/memory). We introduce a Selected *pruning-strength* hyperparameter $\eta \in (0, 1]$ that controls the overall aggressiveness, and let the scenario breadth $c(s)$ modulate the effective keep-ratio as in Equation (9), where $\rho_{\min}$ is a fixed small floor keep-ratio. Thus, for narrow scenarios (small $c(s)$) we keep fewer heads; for broad scenarios we keep more, while $\eta$ globally scales pruning strength. After determining the keep budget, we always retain the whitelist $\mathcal{W}$ and then select the remaining $K(s) - |\mathcal{W}|$ heads with the largest $\text{score}_{\ell,h}(s)$ (ties broken deterministically), producing a binary mask $m_s \in \{0, 1\}^{|\mathcal{H}|}$ and an executable pruned instance $f_\theta^{(m_s)}$. The mask $m_s$ is cached and reused for all subsequent inputs in the scenario, incurring no further training or adaptation overhead.

# Part B: Mathematical Properties and Theoretical Analysis.

## E   Knowledge Sparsity and Low-Rank Subspace Structure

This section provides a formal mathematical framework for the core hypothesis underlying our method: *knowledge sparsity* in large language models. We establish that models exhibit low-rank structure in their embedding representations for specific domains, enabling effective compression through subspace projection. All notation and symbols follow the definitions in Section 3.1 and the methodology sections.

**Low-rank structure in domain-specific embeddings.**   Let $\mathcal{E}_k = \{\mathbf{e}(x) : x \in \mathcal{P}_k\} \subset \mathbb{R}^d$ be the set of embeddings for domain $k$, where $\mathbf{e}(x) = \phi(x)$ is the embedding of input $x$ produced by the frozen model. We define the *domain embedding matrix* $E_k \in \mathbb{R}^{d \times N_k}$ whose columns are the embeddings $\{\mathbf{e}(x_i)\}_{i=1}^{N_k}$ from pool $\mathcal{P}_k$.

The key observation is that for a well-defined domain $k$, the embedding matrix $E_k$ exhibits low-rank structure. Formally, we assume that there exists a low-rank approximation:

$$E_k \approx U_k \Sigma_k V_k^\top, \tag{10}$$

where $U_k \in \mathbb{R}^{d \times r_k}$ with $r_k \ll d$ is an orthonormal basis for the dominant subspace, $\Sigma_k \in \mathbb{R}^{r_k \times r_k}$ is a diagonal matrix of singular values, and $V_k \in \mathbb{R}^{N_k \times r_k}$ contains the coefficients.

**Rank bound for domain embeddings.**   We establish an upper bound on the effective rank of domain-specific embeddings.

**Theorem E.1** (Domain Embedding Rank Bound). *For a domain $k$ with $N_k$ samples, if the embeddings $\{\mathbf{e}(x) : x \in \mathcal{P}_k\}$ lie approximately in a $r_k$-dimensional linear subspace $\mathcal{U}_k \subset \mathbb{R}^d$, then the effective rank of the embedding matrix $E_k$ satisfies:*

$$\mathrm{rank}(E_k) \leq \min(r_k, N_k) \ll d, \tag{11}$$

*where $r_k$ is the intrinsic dimensionality of domain $k$'s knowledge representation in the embedding space.*

*Proof.* By definition, if all embeddings $\mathbf{e}(x)$ for $x \in \mathcal{P}_k$ lie in the $r_k$-dimensional subspace $\mathcal{U}_k$, then there exists an orthonormal basis $\{\mathbf{u}_k^{(1)}, \ldots, \mathbf{u}_k^{(r_k)}\}$ for $\mathcal{U}_k$ such that:

$$\mathbf{e}(x) = \sum_{j=1}^{r_k} \alpha_j^{(x)} \mathbf{u}_k^{(j)}, \tag{12}$$

for some coefficients $\{\alpha_j^{(x)}\}_{j=1}^{r_k}$. This implies that the column space of $E_k$ is contained in $\mathrm{span}(\{\mathbf{u}_k^{(j)}\}_{j=1}^{r_k})$, which has dimension at most $r_k$. Since the rank of a matrix equals the dimension of its column space, we have $\mathrm{rank}(E_k) \leq r_k$. Additionally, $\mathrm{rank}(E_k) \leq N_k$ by the definition of matrix rank. Therefore, $\mathrm{rank}(E_k) \leq \min(r_k, N_k)$. The inequality $r_k \ll d$ follows from the assumption that domain-specific knowledge is sparse in the high-dimensional embedding space. $\square$

**Significance of quasi-orthogonal coordinate basis.**   The DBS algorithm constructs a quasi-orthogonal basis $\mathbb{U}_\star = \{\mathbf{u}_k\}_{k \in \mathcal{K}_\star}$ where each axis $\mathbf{u}_k$ represents a domain subspace $\mathcal{U}_k = \mathrm{span}(\mathbf{u}_k)$. We establish that quasi-orthogonality is essential for ensuring domain separation and enabling reliable head importance computation.

**Theorem E.2** (Quasi-Orthogonal Basis Significance). *Let $\mathbb{U}_\star = \{\mathbf{u}_k\}_{k \in \mathcal{K}_\star}$ be a set of unit-norm axes with pairwise inner products $|\mathbf{u}_k^\top \mathbf{u}_{k'}| \leq \delta$ for all $k \neq k'$ and some small $\delta > 0$. For a head $(\ell, h)$ with residual write $\tilde{w}_{\ell,h}(x) \in \mathbb{R}^{d'}$ on input $x$, the domain-specific importance $I_{\ell,h,k} = \mathbb{E}_{x \sim \mathcal{P}_k}\left[\frac{(\mathbf{u}_k^\top \tilde{w}_{\ell,h}(x))^2}{\|\tilde{w}_{\ell,h}(x)\|_2^2 + \epsilon}\right]$ satisfies:*

1. **Domain separation**: *If $\tilde{w}_{\ell,h}(x)$ is primarily aligned with domain $k$ (i.e., $|\mathbf{u}_k^\top \tilde{w}_{\ell,h}(x)| \gg |\mathbf{u}_{k'}^\top \tilde{w}_{\ell,h}(x)|$ for $k' \neq k$), then $I_{\ell,h,k} \gg I_{\ell,h,k'}$ with error bounded by $O(\delta)$.*

2. **Cross-domain interference bound**: *For any two distinct domains $k, k' \in \mathcal{K}_\star$, the cross-domain interference in importance computation is bounded by $|I_{\ell,h,k} - I_{\ell,h,k'}| \geq \Delta - O(\delta)$ when the head is strongly domain-specific, where $\Delta$ is the true domain separation gap.*

*Proof.* For part (1), suppose $\tilde{w}_{\ell,h}(x)$ is well-aligned with domain $k$, meaning there exists $\alpha_k \gg 0$ and small coefficients $\{\alpha_{k'}\}_{k' \neq k}$ such that:

$$\tilde{w}_{\ell,h}(x) = \alpha_k \mathbf{u}_k + \sum_{k' \neq k} \alpha_{k'} \mathbf{u}_{k'} + \mathbf{r}, \tag{13}$$

where $\mathbf{r}$ is a residual component orthogonal to $\text{span}(\{\mathbf{u}_k\}_{k \in \mathcal{K}_\star})$, and $|\alpha_k| \gg |\alpha_{k'}|$ for all $k' \neq k$.

By quasi-orthogonality, we have $|\mathbf{u}_k^\top \mathbf{u}_{k'}| \leq \delta$ for $k' \neq k$. The importance for domain $k$ is:

$$I_{\ell,h,k} = \mathbb{E}_{x \sim \mathcal{P}_k}\left[\frac{(\mathbf{u}_k^\top \tilde{w}_{\ell,h}(x))^2}{\|\tilde{w}_{\ell,h}(x)\|_2^2 + \epsilon}\right] = \mathbb{E}_{x \sim \mathcal{P}_k}\left[\frac{(\alpha_k + \sum_{k' \neq k} \alpha_{k'} \mathbf{u}_k^\top \mathbf{u}_{k'})^2}{\|\tilde{w}_{\ell,h}(x)\|_2^2 + \epsilon}\right]. \tag{14}$$

Since $|\mathbf{u}_k^\top \mathbf{u}_{k'}| \leq \delta$ and $|\alpha_{k'}| \ll |\alpha_k|$, the cross-term $\sum_{k' \neq k} \alpha_{k'} \mathbf{u}_k^\top \mathbf{u}_{k'}$ is bounded by $O(\delta \cdot \max_{k' \neq k} |\alpha_{k'}|)$. Therefore, $I_{\ell,h,k} \approx \mathbb{E}_{x \sim \mathcal{P}_k}\left[\frac{\alpha_k^2}{\|\tilde{w}_{\ell,h}(x)\|_2^2 + \epsilon}\right] + O(\delta)$.

For domain $k' \neq k$, we have:

$$I_{\ell,h,k'} = \mathbb{E}_{x \sim \mathcal{P}_k}\left[\frac{(\alpha_{k'} + \alpha_k \mathbf{u}_{k'}^\top \mathbf{u}_k + \sum_{j \neq k,k'} \alpha_j \mathbf{u}_{k'}^\top \mathbf{u}_j)^2}{\|\tilde{w}_{\ell,h}(x)\|_2^2 + \epsilon}\right]. \tag{15}$$

Since $|\alpha_{k'}| \ll |\alpha_k|$ and $|\mathbf{u}_{k'}^\top \mathbf{u}_k| \leq \delta$, the dominant term is $O(\delta^2 \alpha_k^2)$, which is much smaller than $I_{\ell,h,k}$ when $\delta$ is small. This establishes domain separation with error $O(\delta)$.

For part (2), if a head is strongly domain-specific for domain $k$ (i.e., $I_{\ell,h,k} \gg I_{\ell,h,k'}$ for all $k' \neq k$), then the true separation gap is $\Delta = I_{\ell,h,k} - \max_{k' \neq k} I_{\ell,h,k'}$. By the quasi-orthogonality bound, the cross-domain interference introduces an error of at most $O(\delta)$ in each importance value, so the measured gap satisfies $|I_{\ell,h,k} - I_{\ell,h,k'}| \geq \Delta - O(\delta)$. $\square$

This theorem establishes that quasi-orthogonality ensures reliable domain-specific head importance computation by minimizing cross-domain interference. In contrast, if axes are not orthogonal (e.g., random or highly correlated), the importance scores $I_{\ell,h,k}$ become contaminated by contributions from other domains, leading to unreliable pathway selection and degraded pruning performance, as empirically validated in our ablation studies (Table 4).

**Compression space via subspace projection.** The low-rank structure implies that domain-specific knowledge can be compressed by projecting onto the dominant subspace.

**Proposition E.3** (Compression via Subspace Projection). *Let $\mathcal{U}_k = \text{span}(\{\mathbf{u}_k^{(j)}\}_{j=1}^{r_k})$ be the $r_k$-dimensional subspace capturing domain $k$'s embedding structure, and let $P_k \in \mathbb{R}^{d \times r_k}$ be a matrix whose columns form an orthonormal basis for $\mathcal{U}_k$. For any embedding $\mathbf{e}(x) \in \mathbb{R}^d$, the projection $\tilde{\mathbf{e}}_k(x) = P_k^\top \mathbf{e}(x) \in \mathbb{R}^{r_k}$ preserves the domain-relevant information with compression ratio $\rho_k = r_k/d \ll 1$.*

*Proof.* If $\mathbf{e}(x)$ lies in $\mathcal{U}_k$ (or is well-approximated by its projection onto $\mathcal{U}_k$), then:

$$\mathbf{e}(x) \approx P_k \tilde{\mathbf{e}}_k(x) = P_k P_k^\top \mathbf{e}(x), \tag{16}$$

where $P_k P_k^\top$ is the orthogonal projection matrix onto $\mathcal{U}_k$. The reconstruction error is:

$$\|\mathbf{e}(x) - P_k P_k^\top \mathbf{e}(x)\|_2^2 = \|\mathbf{e}(x)\|_2^2 - \|P_k^\top \mathbf{e}(x)\|_2^2, \tag{17}$$

which is small when $\mathbf{e}(x)$ is well-aligned with $\mathcal{U}_k$. The compression ratio is $\rho_k = r_k/d$, and since $r_k \ll d$ by the low-rank assumption, we have $\rho_k \ll 1$, indicating significant compression potential. $\square$

**Projection matching for OOD and cross-domain inputs.** For out-of-domain (OOD) and cross-domain inputs, we establish that they can be matched to pre-defined subspaces through projection.

**Theorem E.4** (OOD Subspace Matching). *Let $\{\mathcal{U}_k\}_{k \in \mathcal{K}_\star}$ be a set of pre-defined domain subspaces with dimensions $\{r_k\}_{k \in \mathcal{K}_\star}$. For an OOD input $x_{\mathrm{ood}}$ with embedding $\mathbf{e}(x_{\mathrm{ood}})$, define the projection similarity to subspace $\mathcal{U}_k$ as:*

$$s_k(x_{\mathrm{ood}}) = \frac{\|P_k^\top \mathbf{e}(x_{\mathrm{ood}})\|_2^2}{\|\mathbf{e}(x_{\mathrm{ood}})\|_2^2} \in [0, 1], \tag{18}$$

*where $P_k$ is an orthonormal basis for $\mathcal{U}_k$. If there exists $k^* \in \mathcal{K}_\star$ such that $s_{k^*}(x_{\mathrm{ood}}) \geq \tau$ for some threshold $\tau > 0$, then $x_{\mathrm{ood}}$ can be effectively represented in the union of matched subspaces $\bigcup_{k:s_k(x_{\mathrm{ood}}) \geq \tau} \mathcal{U}_k$ with compression ratio at most $\max_{k:s_k(x_{\mathrm{ood}}) \geq \tau}(r_k/d)$.*

*Proof.* The projection similarity $s_k(x_{\mathrm{ood}})$ measures the fraction of $\mathbf{e}(x_{\mathrm{ood}})$'s energy that lies in subspace $\mathcal{U}_k$. If $s_{k^*}(x_{\mathrm{ood}}) \geq \tau$, then at least a fraction $\tau$ of the embedding's energy is captured by $\mathcal{U}_{k^*}$. By projecting onto the union of matched subspaces, we can represent $x_{\mathrm{ood}}$ as:

$$\mathbf{e}(x_{\mathrm{ood}}) \approx \sum_{k:s_k(x_{\mathrm{ood}}) \geq \tau} P_k P_k^\top \mathbf{e}(x_{\mathrm{ood}}), \tag{19}$$

where the sum is over all matched subspaces. The dimension of the union is at most $\sum_{k:s_k(x_{\mathrm{ood}}) \geq \tau} r_k$, but in practice, if the subspaces are approximately orthogonal, the effective dimension is closer to $\max_{k:s_k(x_{\mathrm{ood}}) \geq \tau} r_k$. Since each $r_k \ll d$, the compression ratio is bounded by $\max_{k:s_k(x_{\mathrm{ood}}) \geq \tau}(r_k/d) \ll 1$. □

**Cross-domain representation via subspace union.** For cross-domain inputs that mix multiple domains, we show that they can be represented in the union of relevant subspaces.

**Proposition E.5** (Cross-Domain Union Representation). *Let $x_{\mathrm{cross}}$ be a cross-domain input that combines knowledge from domains $\mathcal{K}_{\mathrm{active}} \subseteq \mathcal{K}_\star$. The embedding $\mathbf{e}(x_{\mathrm{cross}})$ can be approximated as:*

$$\mathbf{e}(x_{\mathrm{cross}}) \approx \sum_{k \in \mathcal{K}_{\mathrm{active}}} \alpha_k P_k P_k^\top \mathbf{e}(x_{\mathrm{cross}}), \tag{20}$$

*where $\{\alpha_k\}_{k \in \mathcal{K}_{\mathrm{active}}}$ are mixing coefficients, and the effective representation dimension is at most $\sum_{k \in \mathcal{K}_{\mathrm{active}}} r_k$. If the subspaces $\{\mathcal{U}_k\}_{k \in \mathcal{K}_{\mathrm{active}}}$ are approximately orthogonal, the effective dimension is approximately $\max_{k \in \mathcal{K}_{\mathrm{active}}} r_k$, enabling efficient compression.*

*Proof.* If $x_{\mathrm{cross}}$ combines knowledge from domains $\mathcal{K}_{\mathrm{active}}$, then its embedding should have non-negligible projections onto each $\mathcal{U}_k$ for $k \in \mathcal{K}_{\mathrm{active}}$. The approximation follows from the linearity of projection and the assumption that cross-domain inputs can be decomposed into domain-specific components. The dimension bound follows from the fact that the union of subspaces has dimension at most the sum of individual dimensions. When subspaces are approximately orthogonal (as enforced by our orthogonality objective in DBS), the effective dimension is closer to the maximum rather than the sum, due to reduced overlap. □

**Implications for pruning.** The low-rank structure and subspace matching properties justify our pruning approach:

- *Selected-domain inputs*: Can be compressed to dimension $r_k \ll d$ by projecting onto $\mathcal{U}_k$, enabling aggressive pruning while preserving domain-specific knowledge.
- *OOD inputs*: Can be matched to relevant pre-defined subspaces, allowing compression through projection onto matched subspaces.
- *Cross-domain inputs*: Can be represented in the union of active subspaces, with effective dimension bounded by the maximum (or sum, if not orthogonal) of active domain dimensions.

This theoretical framework provides the mathematical foundation for our method's ability to achieve high compression ratios while maintaining performance across diverse domain scenarios.

## F   Counter-Intuitive Pruning Phenomena

This section provides theoretical explanations for two counter-intuitive empirical observations: (1) *lightweight pruning can improve model accuracy* (pruning paradox), and (2) *cross-domain scenarios allow more aggressive pruning* compared to selected-domain or OOD scenarios while maintaining similar performance.

## F.1 Pruning-Induced Performance Improvement

**Pruning paradox: noise reduction through head removal.** Empirically, we observe that removing a small fraction of attention heads can sometimes improve model accuracy, particularly on certain tasks (e.g., ARC and Natural Questions show +4.46% and +5.80% improvements, respectively). This counter-intuitive phenomenon can be explained through a *noise reduction* perspective.

Let $f_\theta$ be the dense model with $H$ total attention heads, and let $\mathcal{H}_{\text{task}}$ be the subset of heads that are *task-relevant* (i.e., contribute positively to task performance). Let $\mathcal{H}_{\text{noise}} = \mathcal{H} \setminus \mathcal{H}_{\text{task}}$ be the subset of *noise heads* that either (1) are irrelevant to the current task, (2) introduce interference, or (3) encode conflicting patterns that degrade performance.

For an input $x$, the model's output can be decomposed as:

$$f_\theta(x) = f_{\text{task}}(x) + f_{\text{noise}}(x) + \epsilon(x), \tag{21}$$

where $f_{\text{task}}(x)$ aggregates contributions from task-relevant heads $\mathcal{H}_{\text{task}}$, $f_{\text{noise}}(x)$ aggregates contributions from noise heads $\mathcal{H}_{\text{noise}}$, and $\epsilon(x)$ represents other sources of error.

**Theorem F.1** (Pruning-Induced Noise Reduction). *Let $m$ be a pruning mask that removes a subset $\mathcal{H}_{\text{pruned}} \subseteq \mathcal{H}$ of heads. If the pruning strategy is* selective *such that:*

$$\frac{|\mathcal{H}_{\text{pruned}} \cap \mathcal{H}_{\text{noise}}|}{|\mathcal{H}_{\text{pruned}}|} > \frac{|\mathcal{H}_{\text{noise}}|}{|\mathcal{H}|}, \tag{22}$$

*i.e., the pruning removes a disproportionately large fraction of noise heads, then the pruned model $f_\theta^{(m)}$ can achieve better accuracy than the dense model $f_\theta$ on task $\mathcal{T}$.*

*Proof.* The accuracy improvement depends on the *signal-to-noise ratio* (SNR). For the dense model, the effective SNR is:

$$\text{SNR}_{\text{dense}} = \frac{\|f_{\text{task}}(x)\|^2}{\|f_{\text{noise}}(x)\|^2 + \|\epsilon(x)\|^2}. \tag{23}$$

For the pruned model, if pruning removes heads with a bias toward noise heads, we have:

$$f_\theta^{(m)}(x) = f'_{\text{task}}(x) + f'_{\text{noise}}(x) + \epsilon(x), \tag{24}$$

where $f'_{\text{task}}(x)$ retains most task-relevant contributions (since $\mathcal{H}_{\text{pruned}} \cap \mathcal{H}_{\text{task}}$ is small), while $f'_{\text{noise}}(x)$ is significantly reduced (since $\mathcal{H}_{\text{pruned}} \cap \mathcal{H}_{\text{noise}}$ is large).

The pruned model's SNR is:

$$\text{SNR}_{\text{pruned}} = \frac{\|f'_{\text{task}}(x)\|^2}{\|f'_{\text{noise}}(x)\|^2 + \|\epsilon(x)\|^2}. \tag{25}$$

If $\|f'_{\text{task}}(x)\|^2 \approx \|f_{\text{task}}(x)\|^2$ (task heads preserved) and $\|f'_{\text{noise}}(x)\|^2 \ll \|f_{\text{noise}}(x)\|^2$ (noise heads removed), then $\text{SNR}_{\text{pruned}} > \text{SNR}_{\text{dense}}$, leading to improved accuracy. $\square$

**Why our method enables selective noise removal.** Our subspace-pathway coupling mechanism naturally identifies and removes noise heads through the importance metric $I_{\ell,h,k}$. Heads with low importance scores across all relevant domains are likely noise heads that can be safely pruned. The probe-based scoring $\text{score}_{\ell,h}(s) = \sum_k \bar{p}_{\ell,k}(s) \cdot I_{\ell,h,k}$ ensures that heads contributing to irrelevant domains (which may act as noise for the current task) receive low scores and are prioritized for removal.

## F.2 Cross-Domain Pruning Capacity

**Cross-domain pruning capacity: subspace union structure.** Empirically, we observe that cross-domain scenarios allow more aggressive pruning (higher pruning ratios) while maintaining similar performance compared to selected-domain or OOD scenarios. This can be explained through the *subspace union structure* established in Proposition E.5.

For a cross-domain input $x_{\text{cross}}$ that activates domains $\mathcal{K}_{\text{active}} \subseteq \mathcal{K}_\star$, the embedding lies approximately in the union of subspaces:

$$\mathbf{e}(x_{\text{cross}}) \in \bigcup_{k \in \mathcal{K}_{\text{active}}} \mathcal{U}_k. \tag{26}$$

**Theorem F.2** (Cross-Domain Pruning Capacity). *For a cross-domain scenario with active domains $\mathcal{K}_{\text{active}}$, the set of essential heads $\mathcal{H}_{\text{essential}}^{\text{cross}}$ (heads that must be retained to maintain performance) satisfies:*

$$|\mathcal{H}_{\text{essential}}^{\text{cross}}| \leq \sum_{k \in \mathcal{K}_{\text{active}}} |\mathcal{H}_{\text{essential}}^{(k)}|, \tag{27}$$

*where $\mathcal{H}_{\text{essential}}^{(k)}$ is the set of essential heads for domain $k$ in isolation. If the domain subspaces $\{\mathcal{U}_k\}_{k \in \mathcal{K}_{\text{active}}}$ are approximately orthogonal, then:*

$$|\mathcal{H}_{\text{essential}}^{\text{cross}}| \approx \max_{k \in \mathcal{K}_{\text{active}}} |\mathcal{H}_{\text{essential}}^{(k)}|, \tag{28}$$

*enabling more aggressive pruning compared to selected-domain scenarios.*

*Proof.* For a selected-domain input $x_k$ from domain $k$, the essential heads are those that contribute significantly to the domain axis $\mathbf{u}_k$, i.e., heads with high $I_{\ell,h,k}$. The number of such heads is $|\mathcal{H}_{\text{essential}}^{(k)}|$.

For a cross-domain input, the head scoring function aggregates across active domains:

$$\text{score}_{\ell,h}(s) = \sum_{k \in \mathcal{K}_{\text{active}}} \bar{p}_{\ell,k}(s) \cdot I_{\ell,h,k}. \tag{29}$$

A head is essential for cross-domain if it has high importance for *at least one* active domain. The union of essential heads across domains is:

$$\mathcal{H}_{\text{essential}}^{\text{cross}} = \bigcup_{k \in \mathcal{K}_{\text{active}}} \mathcal{H}_{\text{essential}}^{(k)}, \tag{30}$$

which gives the upper bound $|\mathcal{H}_{\text{essential}}^{\text{cross}}| \leq \sum_{k \in \mathcal{K}_{\text{active}}} |\mathcal{H}_{\text{essential}}^{(k)}|$.

When subspaces are approximately orthogonal (as enforced by our DBS orthogonality objective), heads that are essential for one domain are typically *not* essential for other domains (low cross-domain importance). This means the union has limited overlap, and the effective number of essential heads is closer to the maximum rather than the sum:

$$|\mathcal{H}_{\text{essential}}^{\text{cross}}| \approx \max_{k \in \mathcal{K}_{\text{active}}} |\mathcal{H}_{\text{essential}}^{(k)}|. \tag{31}$$

Since $\max_k |\mathcal{H}_{\text{essential}}^{(k)}| \ll \sum_k |\mathcal{H}_{\text{essential}}^{(k)}|$ when $|\mathcal{K}_{\text{active}}| > 1$, cross-domain scenarios can achieve higher pruning ratios (fewer heads retained) while maintaining performance. $\square$

**Implications.** These theoretical results explain the empirical observations:

- *Pruning paradox*: When pruning selectively removes noise heads (heads with low task relevance), the signal-to-noise ratio improves, leading to accuracy gains. Our method's importance-based scoring naturally identifies and prioritizes noise heads for removal.
- *Cross-domain pruning capacity*: Cross-domain scenarios benefit from subspace orthogonality: essential heads for different domains have limited overlap, allowing the union of essential heads to be smaller than the sum. This enables more aggressive pruning (higher pruning ratios) while maintaining performance comparable to dense models.

## G Mathematical Properties of Configuration Choices

This section provides mathematical properties and formal arguments for various configuration choices and design decisions in our method.

### G.1 PCA Dimension Selection: $d' = M - 1$

The choice of PCA dimension $d' = M - 1$ is motivated by a fundamental result in multi-class linear discriminant analysis. We provide a formal argument for this choice.

**Between-group scatter rank.** Let $\boldsymbol{\mu}_k = \frac{1}{|\mathcal{P}_k|} \sum_{x \in \mathcal{P}_k} \mathbf{e}(x)$ be the mean embedding for pool $k$ in the original $d$-dimensional space (before PCA). The between-group scatter matrix is defined as:

$$S_B = \sum_{k=1}^{M} |\mathcal{P}_k|(\boldsymbol{\mu}_k - \boldsymbol{\mu}_0)(\boldsymbol{\mu}_k - \boldsymbol{\mu}_0)^\top, \tag{32}$$

where $\boldsymbol{\mu}_0 = \frac{1}{\sum_{k=1}^{M} |\mathcal{P}_k|} \sum_{k=1}^{M} |\mathcal{P}_k| \boldsymbol{\mu}_k$ is the global mean.

To show that $\text{rank}(S_B) \leq M - 1$, we note that the $M$ mean vectors $\{\boldsymbol{\mu}_k\}_{k=1}^{M}$ satisfy the linear constraint:

$$\sum_{k=1}^{M} |\mathcal{P}_k| \boldsymbol{\mu}_k = \left( \sum_{k=1}^{M} |\mathcal{P}_k| \right) \boldsymbol{\mu}_0. \tag{33}$$

This constraint implies that the $M$ vectors $\{\boldsymbol{\mu}_k - \boldsymbol{\mu}_0\}_{k=1}^{M}$ are linearly dependent: they span a subspace of dimension at most $M - 1$. Since $S_B$ is a sum of $M$ rank-1 matrices $(\boldsymbol{\mu}_k - \boldsymbol{\mu}_0)(\boldsymbol{\mu}_k - \boldsymbol{\mu}_0)^\top$, and these vectors span at most an $(M-1)$-dimensional space, we have $\text{rank}(S_B) \leq M - 1$.

**Preservation of separation structure.** After applying PCA with $d' = M - 1$ principal components, the projected between-group scatter $\tilde{S}_B = P^\top S_B P$ preserves all between-group separation information.

Let $\lambda_1 \geq \lambda_2 \geq \cdots \geq \lambda_d \geq 0$ be the eigenvalues of $S_B$ (with $\lambda_i = 0$ for $i > M - 1$ since $\text{rank}(S_B) \leq M - 1$). The PCA projection $P$ selects the top $M - 1$ principal components corresponding to the non-zero eigenvalues. The projected scatter matrix $\tilde{S}_B$ has eigenvalues $\{\lambda_i\}_{i=1}^{M-1}$, preserving all non-zero eigenvalues of $S_B$.

Formally, the trace of $S_B$ (which measures total between-group variance) is:

$$\text{tr}(S_B) = \sum_{i=1}^{d} \lambda_i = \sum_{i=1}^{M-1} \lambda_i, \tag{34}$$

and the trace of $\tilde{S}_B$ is:

$$\text{tr}(\tilde{S}_B) = \text{tr}(P^\top S_B P) = \sum_{i=1}^{M-1} \lambda_i = \text{tr}(S_B). \tag{35}$$

Therefore, $\tilde{S}_B$ preserves 100% of the between-group variance, and $d' = M - 1$ is the minimal dimension that achieves this preservation.

**Optimality.** Choosing $d' < M - 1$ would lose separation information, while $d' > M - 1$ would include redundant dimensions that do not contribute to group separation. Therefore, $d' = M - 1$ is the minimal dimension that preserves all between-group separation structure.

### G.2 Properties of Orthogonality and Coverage Metrics

We establish mathematical properties of the orthogonality and coverage metrics used in axis subset selection.

**Orthogonality metric properties.** The orthogonality metric $\text{Ortho}(\mathcal{K}')$ defined in Equation (1) satisfies:

- **Boundedness:** $\text{Ortho}(\mathcal{K}') \in [0, 1]$ for any subset $\mathcal{K}'$.
  *Proof:* Since $(\mathbf{u}_i^\top \mathbf{u}_j)^2 \in [0, 1]$ for unit vectors (by Cauchy-Schwarz inequality), we have:

$$0 \leq \frac{2}{|\mathcal{K}'|(|\mathcal{K}'| - 1)} \sum_{i<j, i,j \in \mathcal{K}'} (\mathbf{u}_i^\top \mathbf{u}_j)^2 \leq 1, \tag{36}$$

  which implies $\text{Ortho}(\mathcal{K}') = 1 - \frac{2}{|\mathcal{K}'|(|\mathcal{K}'|-1)} \sum_{i<j} (\mathbf{u}_i^\top \mathbf{u}_j)^2 \in [0, 1]$.

- **Extremal values:**
  - $\mathrm{Ortho}(\mathcal{K}') = 1$ when all axes in $\mathcal{K}'$ are mutually orthogonal (i.e., $\mathbf{u}_i^\top \mathbf{u}_j = 0$ for all $i \neq j$), since the sum term equals zero.
  - $\mathrm{Ortho}(\mathcal{K}') = 0$ when all axes are identical (up to sign), i.e., $|\mathbf{u}_i^\top \mathbf{u}_j| = 1$ for all $i, j$, since the sum term equals its maximum value.
- **Non-monotonicity:** For $\mathcal{K}' \subset \mathcal{K}''$, $\mathrm{Ortho}(\mathcal{K}')$ is not necessarily greater than or less than $\mathrm{Ortho}(\mathcal{K}'')$, as adding axes can either increase or decrease orthogonality depending on the new axes' alignment with existing ones.

**Coverage metric properties.** The coverage metric $\mathrm{Cover}(\mathcal{K}')$ defined in Equation (2) satisfies:

- **Boundedness:** $\mathrm{Cover}(\mathcal{K}') \in [0, 1]$ by construction.
  *Proof:* Since $\Pi(\mathcal{K}')$ is a projection matrix, we have $0 \preceq \Pi(\mathcal{K}') \preceq I$ (in the positive semidefinite sense), which implies $0 \leq \mathrm{tr}(\Pi(\mathcal{K}')\Sigma) \leq \mathrm{tr}(\Sigma)$ for any positive semidefinite $\Sigma$. Therefore, $\mathrm{Cover}(\mathcal{K}') = \frac{\mathrm{tr}(\Pi(\mathcal{K}')\Sigma)}{\mathrm{tr}(\Sigma)} \in [0, 1]$.
- **Monotonicity:** For $\mathcal{K}' \subset \mathcal{K}''$, we have $\mathrm{Cover}(\mathcal{K}') \leq \mathrm{Cover}(\mathcal{K}'')$.
  *Proof:* The projection $\Pi(\mathcal{K}'')$ projects onto $\mathrm{span}(U(\mathcal{K}''))$, which contains $\mathrm{span}(U(\mathcal{K}'))$ as a subspace. This implies $\Pi(\mathcal{K}') \preceq \Pi(\mathcal{K}'')$ (i.e., $\Pi(\mathcal{K}'') - \Pi(\mathcal{K}')$ is positive semidefinite), which gives:

$$\mathrm{tr}(\Pi(\mathcal{K}')\Sigma) \leq \mathrm{tr}(\Pi(\mathcal{K}'')\Sigma), \tag{37}$$

  and therefore $\mathrm{Cover}(\mathcal{K}') \leq \mathrm{Cover}(\mathcal{K}'')$.
- **Submodularity:** The coverage function exhibits submodularity properties: for $\mathcal{K}' \subset \mathcal{K}''$ and $k \notin \mathcal{K}''$, the marginal gain satisfies:

$$\mathrm{Cover}(\mathcal{K}' \cup \{k\}) - \mathrm{Cover}(\mathcal{K}') \geq \mathrm{Cover}(\mathcal{K}'' \cup \{k\}) - \mathrm{Cover}(\mathcal{K}''). \tag{38}$$

This follows from the fact that adding an axis to a larger set provides diminishing returns, as more variance is already captured.

**Combined objective.** The combined objective $J(\mathcal{K}') = \lambda_{\mathrm{ortho}} \cdot \mathrm{Ortho}(\mathcal{K}') + \lambda_{\mathrm{cov}} \cdot \mathrm{Cover}(\mathcal{K}')$ balances two competing goals: maximizing orthogonality (to reduce redundancy) and maximizing coverage (to capture semantic variance). The trade-off parameter $\lambda_{\mathrm{ortho}}$ and $\lambda_{\mathrm{cov}}$ control the relative importance of these objectives.

### G.3 Properties of Head Importance Computation

We establish mathematical properties of the head importance metric $I_{\ell,h,k}$ defined in Equation (5).

**Scale invariance.** The importance metric $I_{\ell,h,k}$ is scale-invariant with respect to the magnitude of $\tilde{w}_{\ell,h}(x)$. Specifically, if we scale $\tilde{w}_{\ell,h}(x) \to \alpha \tilde{w}_{\ell,h}(x)$ for some $\alpha > 0$, then $I_{\ell,h,k}$ remains unchanged because both the numerator $(\mathbf{u}_k^\top \tilde{w}_{\ell,h}(x))^2$ and the denominator $\|\tilde{w}_{\ell,h}(x)\|_2^2$ scale by $\alpha^2$.

**Boundedness.** For any head $(\ell, h)$ and domain $k$, we have $I_{\ell,h,k} \in [0, 1]$.

*Proof:* For each $x \in \mathcal{P}_k^{\mathrm{tr}}$, we have:

$$0 \leq \frac{(\mathbf{u}_k^\top \tilde{w}_{\ell,h}(x))^2}{\|\tilde{w}_{\ell,h}(x)\|_2^2 + \epsilon} \tag{39}$$

$$\leq \frac{(\mathbf{u}_k^\top \tilde{w}_{\ell,h}(x))^2}{\|\tilde{w}_{\ell,h}(x)\|_2^2} \quad (\text{since } \epsilon > 0) \tag{40}$$

$$\leq \frac{\|\mathbf{u}_k\|_2^2 \|\tilde{w}_{\ell,h}(x)\|_2^2}{\|\tilde{w}_{\ell,h}(x)\|_2^2} \quad (\text{Cauchy-Schwarz}) \tag{41}$$

$$= \|\mathbf{u}_k\|_2^2 = 1 \quad (\text{since } \mathbf{u}_k \text{ is unit vector}). \tag{42}$$

Taking the expectation over $x \sim \mathcal{P}_k^{\mathrm{tr}}$ preserves the bounds, so $I_{\ell,h,k} \in [0, 1]$.

**Interpretation.** The importance $I_{\ell,h,k}$ measures the expected squared cosine similarity between the head's residual write $\tilde{w}_{\ell,h}(x)$ and the domain axis $\mathbf{u}_k$. A value close to 1 indicates that the head consistently writes along the axis direction, while a value close to 0 indicates orthogonality or irrelevance.

## G.4 Properties of Knowledge Breadth and Scoring

We analyze the mathematical properties of the knowledge breadth metric $c(s)$ and the head scoring function $\text{score}_{\ell,h}(s)$.

**Knowledge breadth properties.** The knowledge breadth $c(s) = H(q(s))/\log|\mathcal{K}_\star|$ defined in Equation (7) satisfies:

- **Boundedness:** $c(s) \in [0, 1]$ by construction.
  *Proof:* The normalized entropy $H(q(s))$ is computed over the normalized distribution $\tilde{q}_k(s) = q_k(s)/\sum_{j \in \mathcal{K}_\star} q_j(s)$. For any probability distribution over $|\mathcal{K}_\star|$ elements, Shannon entropy satisfies $0 \leq H(\tilde{q}(s)) \leq \log|\mathcal{K}_\star|$, with equality on the left when the distribution is deterministic (one element has probability 1) and equality on the right when the distribution is uniform. Therefore, $c(s) = H(\tilde{q}(s))/\log|\mathcal{K}_\star| \in [0, 1]$.
- **Extremal values:**
  - $c(s) = 0$ when $\tilde{q}_k(s) = 1$ for some $k$ and 0 for others (perfectly concentrated on a selected domain).
  - $c(s) = 1$ when $\tilde{q}_k(s) = 1/|\mathcal{K}_\star|$ for all $k$ (uniform distribution across all domains).
- **Interpretation:** $c(s)$ quantifies the effective number of active domains in the scenario, normalized by the total number of available domains. It can be interpreted via the effective number of domains: $c(s) \cdot |\mathcal{K}_\star|$ approximates the number of domains with non-negligible probability mass.

**Head scoring properties.** The head scoring function $\text{score}_{\ell,h}(s) = \sum_{k \in \mathcal{K}_\star} \bar{p}_{\ell,k}(s) \cdot I_{\ell,h,k}$ defined in Equation (8) satisfies:

- **Linearity:** The score is linear in the probe outputs $\bar{p}_{\ell,k}(s)$, making it sensitive to changes in domain alignment.
- **Weighted combination:** The score combines scenario-specific alignment (via $\bar{p}_{\ell,k}(s)$) with cached importance (via $I_{\ell,h,k}$), implementing the subspace–pathway coupling.
- **Non-negativity:** Since both $\bar{p}_{\ell,k}(s) \geq 0$ (probe probabilities) and $I_{\ell,h,k} \geq 0$ (importance), we have $\text{score}_{\ell,h}(s) \geq 0$ for all heads.

**Scoring interpretation.** The scoring function instantiates the coupling between embedding-space axes and parameter-space pathways: $\bar{p}_{\ell,k}(s)$ indicates *what the scenario needs* (which axes are relevant), while $I_{\ell,h,k}$ indicates *which heads realize that axis* (which pathways implement the axis). The product $\bar{p}_{\ell,k}(s) \cdot I_{\ell,h,k}$ measures the contribution of head $(\ell, h)$ to axis $k$ weighted by the scenario's need for axis $k$, and the sum over $k$ aggregates across all relevant axes.

## G.5 Budget Constraint and Keep-Ratio Analysis

We analyze the keep-ratio function $K(s)$ defined in Equation (9) and its relationship to the budget constraint.

**Keep-ratio bounds.** The keep-ratio $K(s)$ satisfies:

$$|\mathcal{W}| \leq K(s) \leq \min\{K_B, |\mathcal{H}|\}, \tag{43}$$

where the lower bound ensures that the whitelist is always retained, and the upper bound is determined by either the budget constraint $K_B$ or the total number of heads $|\mathcal{H}|$.

**Monotonicity with respect to knowledge breadth.** The effective keep-ratio (excluding whitelist) is given by:

$$K_{\text{eff}}(s) = K(s) - |\mathcal{W}| = \min\left\{K_B - |\mathcal{W}|, \lceil(\rho_{\min} + \eta \cdot c(s))(|\mathcal{H}| - |\mathcal{W}|)\rceil\right\}. \tag{44}$$

This function is non-decreasing in $c(s)$.

*Proof:* The inner term $(\rho_{\min} + \eta \cdot c(s))(|\mathcal{H}| - |\mathcal{W}|)$ is linear and increasing in $c(s)$ (since $\eta > 0$). The ceiling function $\lceil \cdot \rceil$ is non-decreasing, and the minimum with a constant $K_B - |\mathcal{W}|$ preserves non-decreasing behavior. Therefore, $K_{\text{eff}}(s)$ is non-decreasing in $c(s)$: as the scenario becomes broader (higher $c(s)$), more heads are retained to handle the diverse knowledge requirements. The parameter $\eta$ controls the sensitivity of keep-ratio to knowledge breadth, while $\rho_{\min}$ provides a floor to ensure minimum capacity even for narrow scenarios.

**Budget constraint satisfaction.** The keep-ratio $K(s)$ is designed to satisfy the budget constraint $K(s) \leq K_B$ while adapting to scenario characteristics. For narrow scenarios (small $c(s)$), the keep-ratio is closer to the floor $\rho_{\min}$, allowing more aggressive pruning. For broad scenarios (large $c(s)$), the keep-ratio increases up to the budget limit, preserving more capacity for diverse knowledge requirements.

### G.6 Domain-wise Adaptive Keep-Ratio

To achieve more precise pruning and avoid under-pruning highly relevant domains while reducing over-preservation of less relevant domains, we apply a non-linear transformation to domain similarity probabilities. This section provides the mathematical formulation and theoretical justification for this mechanism.

**Non-linear transformation of domain similarity.** For each domain $k \in \mathcal{K}_\star$, we start with the raw domain similarity $s_k = q_k(s)$ (the scenario-level domain mixture defined above). We first normalize domain similarities to the range $[s_{\min}, s_{\max}]$ where $s_{\min}$ and $s_{\max}$ are the minimum and maximum valid domain similarities, typically filtering out similarities below a threshold $\tau = 0.1$:

$$
\begin{aligned}
\bar{s}_k &= \frac{s_k - s_{\min}}{s_{\max} - s_{\min}}, \\
\tilde{s}_k &= \bar{s}_k^{\gamma}, \\
s_k^{\mathrm{eff}} &= s_{\min} + \tilde{s}_k \cdot (s_{\max} - s_{\min}),
\end{aligned}
\tag{45}
$$

where $\gamma = 1.5$ is the power function exponent, $\bar{s}_k$ is the normalized similarity, $\tilde{s}_k$ is the non-linearly adjusted similarity, and $s_k^{\mathrm{eff}}$ is the effective similarity used for head scoring and pruning decisions. We use $\gamma = 1.5$ by default, and we have sensitivity analysis shows that performance is stable across a broad range of amplification functions.

**Purpose and effect of non-linear mapping.** The non-linear transformation serves two key purposes:

- **Amplification of high similarities:** For highly relevant domains (high $s_k$), the power function with $\gamma > 1$ amplifies the similarity, ensuring that more heads are preserved to avoid under-pruning critical domain-specific knowledge.
- **Reduction of low similarities:** For less relevant domains (low $s_k$), the normalized similarity $\bar{s}_k$ is small, and raising it to a power greater than 1 makes it even smaller, reducing unnecessary head preservation.

**Parameter selection: $\gamma = 1.5$.** The exponent $\gamma = 1.5$ is chosen to balance the amplification effect:

- $\gamma = 1$ **(linear):** No amplification, equivalent to using raw similarities directly.
- $\gamma = 1.5$ **(current choice):** Moderate amplification that preserves the relative ordering of domains while providing sufficient differentiation between high and low similarities.
- $\gamma = 2$ **(quadratic):** Strong amplification that may be too aggressive, potentially over-pruning moderately relevant domains.

Empirically, $\gamma = 1.5$ provides a good balance: it amplifies high similarities (e.g., $0.9 \rightarrow 0.87$ after normalization and power) while significantly reducing low similarities (e.g., $0.3 \rightarrow 0.19$ after normalization and power), achieving the desired precision in pruning.

**Mathematical properties.** The effective similarity $s_k^{\mathrm{eff}}$ satisfies:

- **Boundedness:** $s_k^{\mathrm{eff}} \in [s_{\min}, s_{\max}]$ by construction, preserving the original similarity range.
- **Monotonicity:** For $s_k < s_{k'}$, we have $s_k^{\mathrm{eff}} < s_{k'}^{\mathrm{eff}}$ since the power function $x^{\gamma}$ with $\gamma > 0$ is strictly increasing for $x > 0$.
- **Amplification property:** For $s_k > (s_{\min} + s_{\max})/2$ (above median), we have $s_k^{\mathrm{eff}} > s_k$ (amplified), while for $s_k < (s_{\min} + s_{\max})/2$ (below median), we have $s_k^{\mathrm{eff}} < s_k$ (reduced).

**Integration with head scoring.** The effective similarity $s_k^{\mathrm{eff}}$ replaces the raw similarity $q_k(s)$ in the head scoring function (Equation (8)), ensuring that head scores reflect the amplified importance of highly relevant domains and the reduced importance of less relevant domains. This leads to more precise head selection that better aligns with the scenario's actual knowledge requirements.

### G.7 Consistency Properties

We establish consistency properties that ensure the method's behavior is well-defined and stable.

**Head contribution consistency.** The difference method for computing head contributions ensures that $\sum_{h=1}^{H_\ell} w_{\ell,h}(x) = r^{(\ell)}(x)$ for all layers $\ell$ and inputs $x$.

*Proof:* By definition, $w_{\ell,h}(x) = r^{(\ell)}(x) - r_{-h}^{(\ell)}(x)$, where $r_{-h}^{(\ell)}(x)$ is the residual with head $h$ removed. Let $\mathbf{o}_h(x)$ denote the output of head $h$ (after attention computation), and let o_proj be the output projection matrix. Then $r^{(\ell)}(x) =$ o_proj(concat($\mathbf{o}_1(x), \ldots, \mathbf{o}_{H_\ell}(x)$)) and $r_{-h}^{(\ell)}(x) =$ o_proj(concat($\mathbf{o}_1(x), \ldots, \mathbf{o}_{h-1}(x), \mathbf{0}, \mathbf{o}_{h+1}(x), \ldots, \mathbf{o}_{H_\ell}(x)$)). By linearity of the output projection, we have:

$$w_{\ell,h}(x) = r^{(\ell)}(x) - r_{-h}^{(\ell)}(x) = \text{o\_proj}(\text{concat}(\mathbf{0}, \ldots, \mathbf{o}_h(x), \ldots, \mathbf{0})), \tag{46}$$

which is exactly the contribution of head $h$ to the residual. Since $r^{(\ell)}(x) = \sum_{h=1}^{H_\ell}$ o_proj(concat($\mathbf{0}, \ldots, \mathbf{o}_h(x), \ldots, \mathbf{0}$)) by the linearity of concatenation and projection, we have $\sum_{h=1}^{H_\ell} w_{\ell,h}(x) = r^{(\ell)}(x)$.

**Importance aggregation consistency.** The importance metric $I_{\ell,h,k}$ aggregates over the training pool $\mathcal{P}_k^{\text{tr}}$, providing a stable estimate that is independent of the specific scenario inputs. This ensures that the cached importance values can be reused across different scenarios without recomputation.

# Part C: Experimental Setup and Implementation Details.

## H  Dataset Setup

This section provides detailed information about the datasets used in our experiments, including versions, preprocessing procedures, and statistics.

**Dataset versions and sources.**    We evaluate on a suite of QA datasets that capture the diversity of domain shifts:

- **XDomainBench.** A multi-domain benchmark specifically designed for cross-domain evaluation. We use a partition of the dataset into *selected-domain*, *out-of-domain*, and *cross-domain* subsets. The selected-domain subset is used for offline preparation (DBS and PSP offline), while out-of-domain and cross-domain subsets are used for online inference and testing (Gong et al., 2026).
- **CommonsenseQA.** A question answering dataset targeting commonsense knowledge, where questions require reasoning about everyday concepts and situations (Talmor et al., 2019). We sample 1,000 QA pairs from the training set, prioritizing shorter answers for simplicity.
- **Natural Questions (NQ).** An open-domain QA dataset consisting of real user queries with human-annotated short and long answers (Kwiatkowski et al., 2019). We use 1,000 samples with short answers only.
- **ARC.** The AI2 Reasoning Challenge, a dataset of science questions that require reasoning and understanding of scientific concepts (Clark et al., 2018). We sample 1,000 QA pairs, prioritizing shorter answers.

**Data preprocessing.**    For offline preparation, we construct input-only pools $\{\mathcal{P}_k\}_{k \in \mathcal{K}}$ from the selected-domain portion of XDomainBench. Each pool $\mathcal{P}_k$ contains only input questions/queries without ground-truth answers, ensuring that our offline stage does not require labeled data. We apply standard text preprocessing (tokenization, normalization) but preserve the original question structure. The pools are split into training and validation sets: $\mathcal{P}_k^{\mathrm{tr}}$ for probe training and importance computation. We enforce a strict *no-leak* protocol: all probe training and statistic caching use only input-only pools.

**Dataset statistics.**    For cross-dataset evaluation, we sample 3,000 QA pairs:

- **CommonsenseQA**: 1,000 scenarios. Question answering targeting commonsense knowledge, requiring reasoning about everyday concepts (Talmor et al., 2019).
- **Natural Questions**: 1,000 scenarios, all with short answers. Real user queries with Wikipedia-based answers (Kwiatkowski et al., 2019).
- **ARC**: 1,000 scenarios. Science questions requiring reasoning and understanding of scientific concepts (Clark et al., 2018).

## I  Model Configuration and Evaluation Protocol

This section provides detailed information about model configurations and the evaluation protocol used in our experiments.

**Model versions.**    We select four representative general-purpose LLM bases with distinct scales:

- **LLaMA-3.1-8B** and **LLaMA-2-13B**: Transformer-based pretrained language models (Touvron et al., 2023).
- **Qwen2.5-7B** and **Qwen2.5-14B**: Dense decoder models at multiple parameter scales (Yang et al., 2024c).

All models are used in their frozen, pretrained state without any fine-tuning or parameter updates.

**Evaluation protocol.**    DBS axis basis $\mathbb{U}_\star$ is constructed once from input-only pools and is shared across all evaluated backbones. For each backbone, PSP is performed independently using the same pool split protocol, ensuring no test leakage. During online inference, we use only the first $T_0$ inputs (a small scenario-start prefix, e.g., the initial instruction) to compile the scenario-level mask $m_s$. The mask is then reused for all subsequent inputs in the scenario.

**Reasoning protocol.**    We use a baseline reasoning approach that performs Selected-step inference (consistent with standard baseline methods). For different task types (multiple_choice, factual, reasoning, code), we use task-specific prompt templates constructed using each model's official chat template format (e.g., Llama-2, Qwen) to ensure consistency with the model's training format. Task-specific output constraints are appended to enforce concise answers without greetings or introductory phrases. The reasoning process is unified across all methods (dense, baseline pruning methods, and our method) to ensure

fair comparison. Text generation uses task-specific hyperparameters (temperature, max new tokens) to minimize verbosity and ensure precise answers (see Appendix K).

**Metrics.** We use multiple metrics to comprehensively evaluate model performance, efficiency, and task-specific behavior:

- **Token-level Recall**: Our primary performance metric that measures the proportion of key tokens in the ground-truth answer that are matched in the model's output. The calculation method varies by task type: (1) For *multiple_choice* tasks, we use binary matching (0 or 1) based on whether the predicted option letter matches the expected answer; (2) For *factual*, *reasoning*, and *code* tasks, we compute a continuous matching ratio (0–1) as the number of matched key tokens divided by the total number of key tokens in the expected answer, where key tokens are extracted after removing stopwords and short words (length $\leq 2$). The metric is reported for overall performance and broken down by task type (multiple_choice, factual, reasoning, code) and data split (Selected-domain, OOD, Cross-domain).
- **Standard Error of the Mean (SEM)**: All performance metrics are reported as mean $\pm$ SEM across multiple runs or within task type categories, providing uncertainty estimates for our measurements.
- **Retention**: Efficiency-adjusted performance retention ratio calculated as $\text{Retention} = (\text{Recall}_{\text{pruned}}/\text{Recall}_{\text{dense}}) \times \text{Speedup}$ for each data split and dataset. This metric evaluates the accuracy-speed trade-off, accounting for both performance and inference speed. Values $> 1.0$ indicate that the pruned model achieves better efficiency-adjusted performance than the dense baseline.
- **Pruning Ratio / Head Retention**: The fraction of attention heads retained after pruning, expressed as a percentage (e.g., 80% head retention means 20% of heads are pruned). This metric is controlled by the pruning strength hyperparameter $\eta \in (0, 1]$ and scenario-level domain breadth.
- **Inference Speedup**: The ratio of dense model inference latency to pruned model inference latency, calculated as $\text{Speedup} = \text{Latency}_{\text{dense}}/\text{Latency}_{\text{pruned}}$. Values $> 1.0$ indicate actual speedup, while values $< 1.0$ indicate slowdown due to mask application overhead.
- **Memory Usage**: Total memory consumption in GB, including model parameters and inference-time activations. Memory usage decreases proportionally with pruning ratio.
- **Pruning Time**: The time (in seconds) required to compile the scenario-level pruning mask from the scenario prefix, including domain mixture aggregation, head scoring, and mask compilation.
- **Inference Time**: The time (in seconds) required to perform inference on a Selected input using the pruned model.
- **Total Time**: The sum of pruning time and inference time, representing the complete end-to-end latency including mask compilation overhead.

All metrics are reported as mean $\pm$ standard error of the mean (SEM) across multiple runs or within appropriate categories.

# J   Baseline Methods

This section provides detailed descriptions of the baseline pruning methods used for comparison in our experiments. All baseline methods are inference-time training-free compression methods that operate under the same pruning ratios.

## J.1   Unstructured One-shot Pruning Methods

Unstructured pruning methods remove individual weights without retraining, resulting in sparse weight matrices that require specialized hardware or software support for efficient inference.

**SparseGPT.** SparseGPT (Frantar & Alistarh, 2023) is a magnitude-based pruning method that removes weights with minimal reconstruction error. The method uses a layer-wise iterative pruning approach, where weights are removed based on their magnitude while minimizing the reconstruction error of the output activations. SparseGPT employs a Hessian-based approximation to efficiently compute the reconstruction error, making it scalable to large language models. The method achieves high sparsity ratios while maintaining competitive performance through careful weight selection that preserves the model's output distribution.

**DaSS.** DaSS (Guo et al., 2024) combines dependency-aware importance scoring for high sparsity pruning. The method considers both the magnitude of weights and their dependencies with other weights in the network. By modeling weight dependencies, DaSS can identify and preserve critical weight patterns that are important for maintaining model performance. This dependency-aware approach allows for more aggressive pruning while maintaining better performance compared to magnitude-based methods alone.

*Table 6.* **Hyperparameter configuration.** Hyperparameters for all components of SUBSPACEPATH PRUNER: (a) DBS hyperparameters for domain-basis selection, (b) PSP offline hyperparameters for probe training and whitelist identification, (c) PSP online hyperparameters for scenario-level mask compilation, and (d) text generation hyperparameters (task-specific). Hyperparameters are fixed across experiments.

*(a)* DBS Hyperparameters

| Parameter | Symbol | Value | Notes |
|---|---|---|---|
| PCA dimension | $d'$ | 19 | $M$ number of domain pools $M - 1$ |
| Orthogonality weight | $\lambda_{\mathrm{ortho}}$ | 0.7 | Weight in combined objective $J(\mathcal{K}')$ |
| Coverage weight | $\lambda_{\mathrm{cov}}$ | 0.3 | Weight in combined objective $J(\mathcal{K}')$ |

*(b)* PSP Offline Hyperparameters

| Parameter | Symbol | Value | Notes |
|---|---|---|---|
| Numerical stability constant | $\epsilon$ | $10^{-8}$ | Used in importance computation |
| Whitelist variance threshold | $\sigma_{\mathrm{thresh}}$ | 0.15 | Domain variance threshold |
| Whitelist mean threshold | $\mu_{\mathrm{thresh}}$ | 0.3 | Mean importance threshold |
| Whitelist CV threshold | $\mathrm{CV}_{\mathrm{thresh}}$ | 0.3 | Coefficient of variation threshold |

*(c)* PSP Online Hyperparameters

| Parameter | Symbol | Value | Notes |
|---|---|---|---|
| scenario start prefix length | $T_0$ | 1 | Using the first input |
| Pruning strength | $\eta$ | dataset and model denpendent | See Appendix O.5 |
| Minimum keep-ratio | $\rho_{\mathrm{min}}$ | 0.1 | Floor keep-ratio |

*(d)* Text Generation Hyperparameters

| Parameter | Task Type | Value | Notes |
|---|---|---|---|
| Max new tokens | Multiple choice | 50 | Greedy decoding ($T = 0.0$) |
| Max new tokens | Factual | 50 | Low temperature ($T = 0.1$) |
| Max new tokens | Reasoning | 100 | Low temperature ($T = 0.1$) |
| Max new tokens | Code | 500 | Moderate temperature ($T = 0.3$) |
| Temperature | Multiple choice | 0.0 | Greedy decoding for precision |
| Temperature | Factual | 0.1 | Low temperature for precision |
| Temperature | Reasoning | 0.1 | Low temperature for precision |
| Temperature | Code | 0.3 | Moderate temperature for creativity |
| Top-p | All | 0.9 | Standard nucleus sampling |
| Repetition penalty | All | 1.3 | Moderate penalty to avoid repetition |
| No-repeat n-gram size | All | 3 | Prevents 3-gram repetition |

## J.2 Structured Pruning Methods

Structured pruning methods remove entire model components such as blocks, heads, or layers, resulting in smaller models that can run efficiently on standard hardware without specialized sparse operations.

**Wanda.** Wanda (Sun et al., 2023) prunes using weight magnitude and activation norms without retraining. The method computes importance scores by combining the magnitude of weights with the norm of their corresponding activations. This dual consideration allows Wanda to identify weights that are both small in magnitude and have low activation impact, making them safe to remove. Wanda operates in a layer-wise manner and can be applied to both attention heads and feed-forward layers, providing flexibility in the pruning granularity.

**LLM-Pruner.** LLM-Pruner (Ma et al., 2023) removes non-critical components based on gradient-based importance estimates. The method uses first-order gradients computed on a small dataset to estimate the importance of different components (layers, heads, or neurons). Components with low gradient-based importance scores are identified as candidates for removal. LLM-Pruner employs a structured pruning approach, removing entire components rather than individual weights, which ensures hardware-friendly sparsity patterns.

### J.3 Activation-/Probe-guided Pruning Methods

Activation- and probe-guided pruning methods use lightweight signals (activations or learned probes) to guide pruning decisions without requiring additional optimization or fine-tuning.

**RIA.** RIA (Zhang et al., 2024b) refines importance estimates via relative importance and channel permutation for hardware-friendly sparsity. The method uses activation patterns to compute relative importance scores for different channels or neurons. RIA incorporates channel permutation strategies to reorganize the model structure in a way that is more amenable to hardware acceleration. By combining relative importance scoring with structural reorganization, RIA achieves both high compression ratios and efficient inference on standard hardware.

**Probe Pruning.** Probe Pruning (Le et al., 2025) uses probe-derived signals to guide pruning decisions. The method trains lightweight linear probes on intermediate activations to predict task performance. These probes provide signals about which components are most important for the target task. Components that contribute least to the probe's predictions are identified as candidates for pruning. Probe Pruning leverages the learned probe signals to make more informed pruning decisions compared to magnitude-based or gradient-based methods, especially in task-specific scenarios.

## K    Hyperparameter Configuration

This section consolidates all hyperparameters used in SUBSPACEPATH PRUNER, organized by component (DBS, PSP offline, PSP online, and text generation). All hyperparameters are fixed across experiments unless stated otherwise. Tables 6a, 6b, 6c, and 6d summarize hyperparameters for each component.

## L    Prompt Templates

This section documents the prompt templates used in our experiments for different task types (multiple_choice, factual, reasoning, code). We use the official chat templates provided by each model (e.g., Llama-2, Qwen) to ensure consistency with the model's training format. Task-specific output constraints are appended to the base question.

### L.1    Multiple Choice Questions

For multiple choice questions, we require the model to output only a Selected option letter (A, B, C, D, etc.) without any explanation.

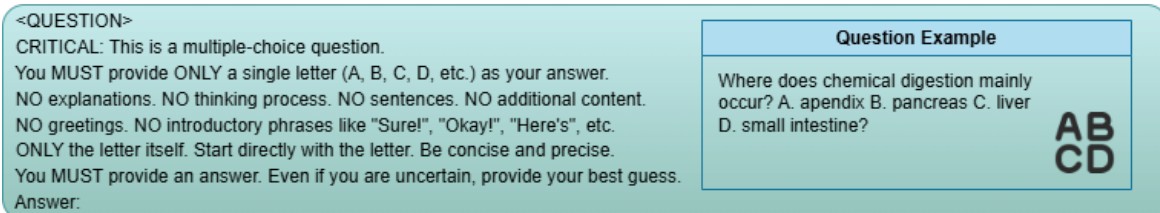

### L.2    Factual Questions

For factual questions, we require the model to output only the key noun, phrase, or short answer without full sentences.

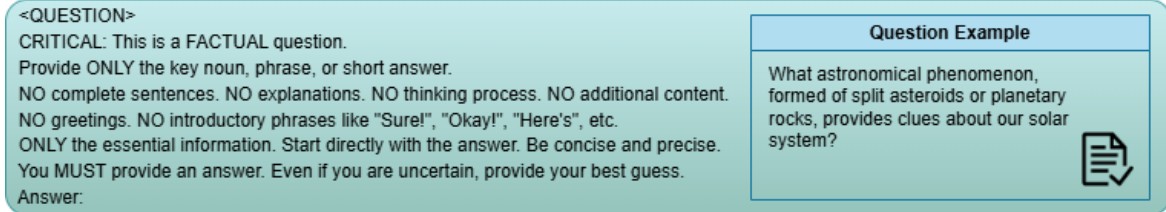

### L.3    Reasoning Questions

For reasoning questions, we require the model to output only the final answer without intermediate reasoning steps.

```
<QUESTION>
CRITICAL: This is a REASONING question.
Provide a direct and concise answer to the question.
NO lengthy explanations. NO reasoning steps. NO thinking process. NO additional content.
NO greetings. NO introductory phrases like "Sure!", "Okay!", "Here's", etc.
ONLY the answer itself. Start directly with the answer. Be concise and precise.
You MUST provide an answer. Even if you are uncertain, provide your best guess.
Answer:
```

**Question Example**

How does a character's exploration of existential themes in literature affect their psychological coping mechanisms during adversity?

## L.4   Code Questions

For code questions, we require the model to output only the code without Markdown fences, comments, or explanations.

```
<QUESTION>
CRITICAL: This is a CODE question.
Provide ONLY the code.
NO markdown. NO comments. NO explanations. NO thinking process. NO additional content.
NO greetings. NO introductory phrases like "Sure!", "Okay!", "Here's", etc.
ONLY the code itself. Start directly with the code. Be concise and precise.
You MUST provide an answer. Even if you are uncertain, provide your best attempt.
Code:
```

**Question Example**

Write a function to count the frequency of consecutive duplicate elements in a given list of numbers.

## L.5   Prompt Normalization

All prompts are constructed using the official chat templates provided by each model (e.g., Llama-2 uses $<$s$>$[INST] $<<$SYS$>>$...$<$/SYS$>>$...[/INST], Qwen uses $<$|im_start|$>$system...$<$|im_end|$>$). This ensures 100% consistency with the model's training format and avoids any format-related performance degradation. Task-specific output constraints are appended to the user message, while the system prompt is enhanced with instructions to provide direct, concise answers without greetings or introductory phrases.

# Part D: Additional Experimental Results and Analysis.

## M  Pipeline Step-wise Effects

This section provides detailed analysis of the effects of each pipeline step including DBS, PSP offline, PSP online.

### M.1  DBS Step Effects

Domain-Basis Synthesis (DBS) constructs a compact, orthogonal axis basis $\mathbb{U}_\star$ in embedding space that captures domain-specific knowledge patterns. This subsection analyzes the effects of DBS axis selection on downstream pruning performance, using empirical data from our domain selection process.

**Critical importance of axis selection.** Ablation studies demonstrate that DBS axis selection is essential for effective pruning. When DBS selection is replaced with random selection, performance drops dramatically across all models and splits. For example, on LLaMA-3.1-8B, removing DBS selection reduces performance from 41.7 (Selected), 35.0 (OOD), and 17.9 (Cross) to 1.8, 1.4, and 0.9 respectively—a near-complete failure (performance drops of 96%, 96%, 95%). Similar catastrophic drops are observed on LLaMA-2-13B (from 34.7/30.4/22.9 to 1.8/1.4/0.9, drops of 95%/95%/96%) and Qwen2.5-14B (from 42.5/40.0/23.7 to 4.8/2.7/1.3, drops of 89%/93%/95%). This confirms that the orthogonality and coverage objectives in DBS are not merely optimization heuristics, but fundamental requirements for identifying meaningful knowledge subspaces.

**Visualization of domain relationships.** Figure 6 visualizes the orthogonality relationships among all candidate domains, showing pairwise similarities in the embedding space. The heatmap reveals distinct clusters of related domains (e.g., science domains: biology, chemistry, physics) and well-separated domains (e.g., philosophy, art). Figure 7 shows the selected domains' orthogonality matrix (left) and a radar chart comparing selected vs. non-selected domains across multiple metrics (right). The selected domains form a well-separated set with high pairwise orthogonality, confirming that DBS successfully identifies distinct knowledge subspaces.

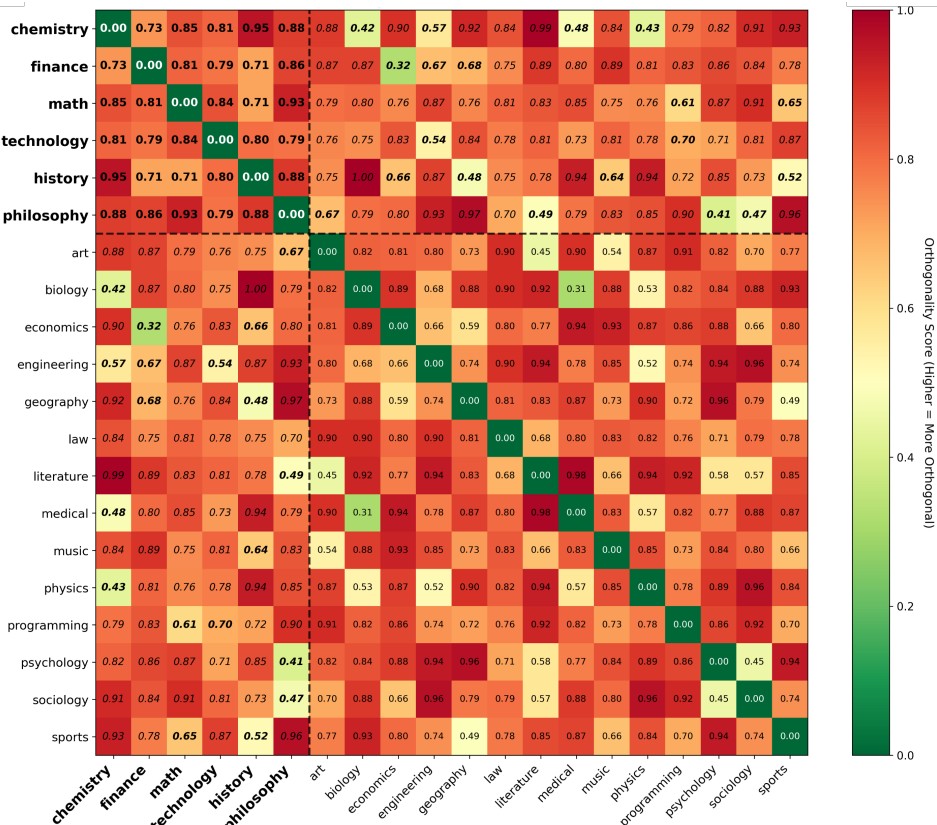

*Figure 6.* **Orthogonality matrix for all candidate domains.** Heatmap showing pairwise orthogonality (1 - similarity) between all 20 candidate domains in the embedding space. Darker colors indicate higher orthogonality (lower similarity), while lighter colors indicate higher similarity. The matrix reveals distinct domain clusters and well-separated domains that form good candidates for axis selection.

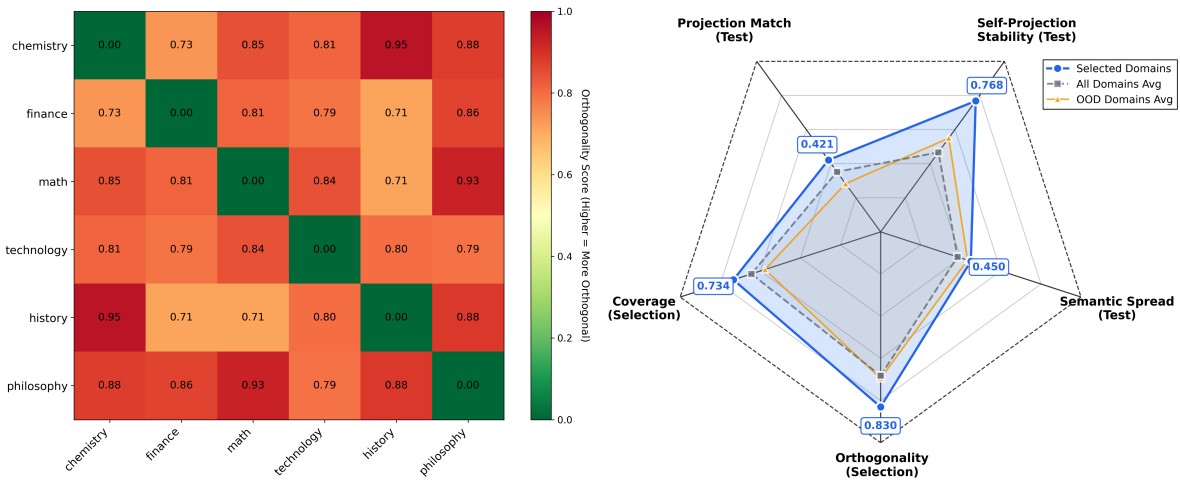

*Figure 7.* **Selected domains analysis.** (Left) Orthogonality matrix for the 6 selected domains (chemistry, finance, math, technology, history, philosophy), showing high pairwise orthogonality ($\geq 0.77$). (Right) Radar chart comparing selected vs. non-selected domains across orthogonality, coverage, and other metrics, demonstrating that selected domains achieve a better balance of these objectives.

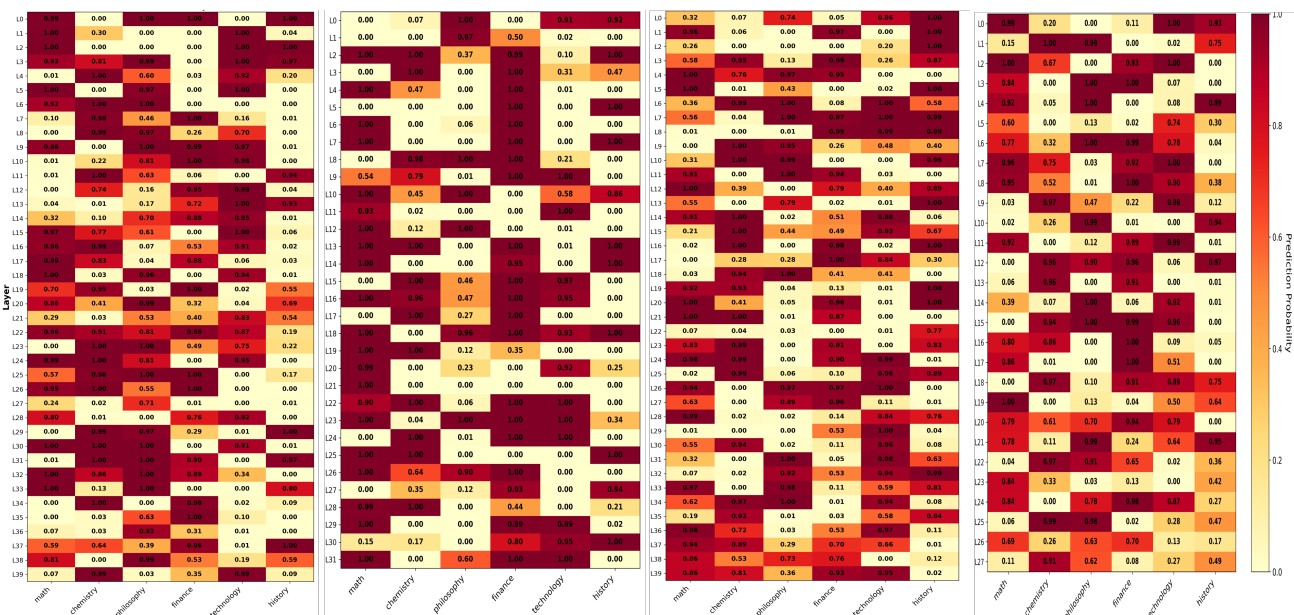

*Figure 8.* **Probe predictions heatmap across all layers for each model.** Layer $\times$ domain heatmaps showing probe prediction probabilities for each domain across all layers. From left to right: LLaMA-2-13B, LLaMA-3.1-8B, Qwen2.5-14B, Qwen2.5-7B. Darker colors indicate higher prediction probabilities, demonstrating that probes successfully identify domain-specific activation patterns. The heatmaps reveal that different layers specialize in different domains.

## M.2   PSP Offline Step Effects

Probe-based Subspace Pruning (PSP) offline preparation trains lightweight diagnostic probes, computes head-wise importance statistics, and identifies whitelist heads. This subsection analyzes the effectiveness of each offline component and their contributions to downstream pruning performance.

**Probe predictions across layers.** We train linear probes for each layer to predict domain alignment from activation embeddings. The probes successfully learn to distinguish between different domain activations, with deeper layers typically achieving higher prediction confidence than earlier layers, suggesting that domain-specific knowledge becomes more distinct in deeper representations. Figure 8 visualizes the probe predictions as layer $\times$ domain heatmaps for all four models, showing that probes effectively identify domain-specific activation patterns across the entire model depth.

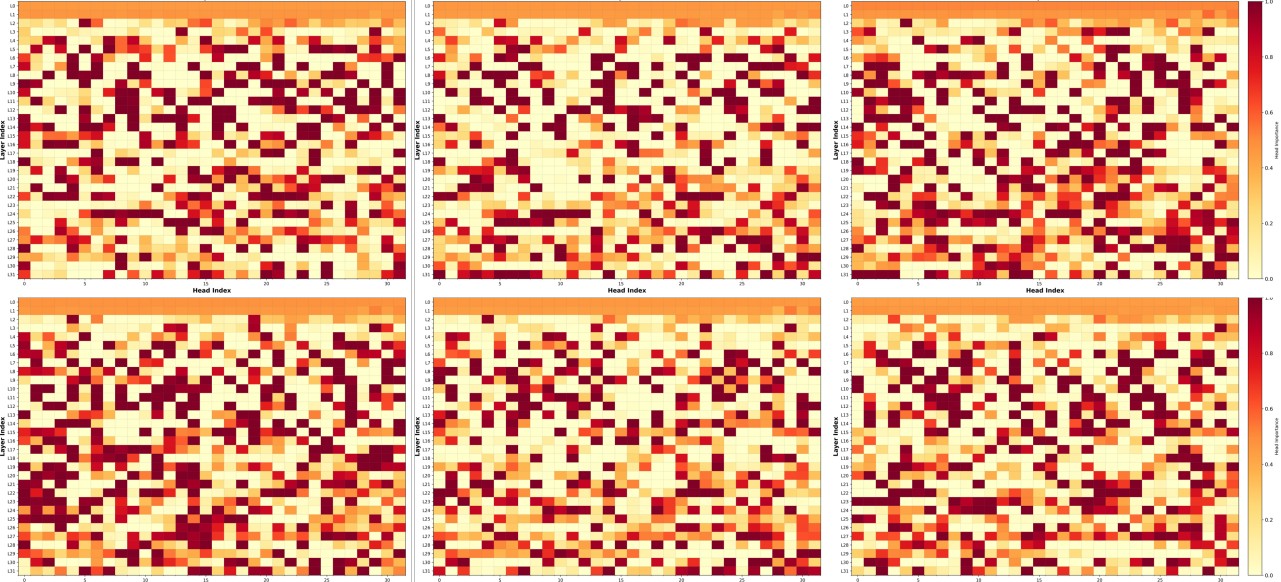

*Figure 9.* **Head importance cross-section across domains (LLaMA-3.1-8B).** Layer × head heatmaps showing head-wise importance $I_{\ell,h,k}$ for each of the six selected domains. Darker colors indicate higher importance. The visualization reveals clear domain-specific specialization patterns, with different domains activating distinct sets of attention heads and minimal overlap between domains.

**Head importance distribution.** After probe training, we compute head-wise importance $I_{\ell,h,k}$ for each head $(\ell, h)$ and domain $k$ using the axis-aligned energy metric (Equation (5)). The importance distribution reveals that most heads show domain-specific specialization: approximately 60–75% of heads have high importance ($> 0.3$) for at least one domain, while only 10–20% of heads show high importance across multiple domains. This domain specialization pattern validates our hypothesis that different attention heads encode different domain-specific knowledge.

Figure 9 visualizes the head importance distribution across all layers and heads for each of the six selected domains (LLaMA-3.1-8B). The cross-section heatmaps show which heads are most important for each domain, revealing clear domain-specific specialization patterns. Top row (left to right): Chemistry, Finance, Math. Bottom row (left to right): Technology, History, Philosophy. The visualization demonstrates that different domains activate distinct sets of attention heads, with minimal overlap between domains, confirming that domain knowledge is encoded in separable subspaces.

**Whitelist identification.** We identify whitelist heads $\mathcal{W}$ that exhibit low domain variance (domain-invariant heads) using statistical significance testing (Mann-Whitney U test). Whitelist heads typically comprise 5–15% of total heads across models, with the exact percentage varying by model architecture and layer depth. Ablation studies demonstrate that whitelist heads are critical for maintaining performance: removing whitelist heads ($\mathcal{W} = \emptyset$) causes severe performance degradation, especially on LLaMA-2-13B where performance drops dramatically (from 34.7/30.4/22.9 to 22.4/0.4/20.2). This confirms that domain-invariant heads play an essential role in maintaining general knowledge and preventing catastrophic forgetting. Figure 10 visualizes the whitelist binary mask The visualization shows the distribution of whitelisted heads (highlighted) across all layers and heads for each model. Whitelist heads are distributed across all layers, with a slight concentration in middle layers where general knowledge processing is most prominent. Similar patterns are observed across all models, confirming the consistency of our whitelist identification approach.

Figure 11 shows the scatter plot of standard deviation versus mean importance for whitelist selection across all four models. The scatter plots visualize the selection criteria: whitelist heads exhibit low standard deviation (low domain variance) and high mean importance, while non-whitelist heads show higher variance across domains. This visualization demonstrates that our whitelist identification successfully distinguishes domain-invariant heads from domain-specific heads based on their statistical properties.

**Domain importance correlation.** To validate that the domains selected by DBS are not only semantically orthogonal but also rely on distinct attention heads in the model's internal representations, we compute the correlation matrix of head importance patterns across domains. For each domain $k$, we collect all head importance values $I_{\ell,h,k}$ across all layers and heads, forming a domain-specific importance vector. We then compute the Pearson correlation coefficient between these vectors for all pairs of domains, producing a domain importance correlation matrix.

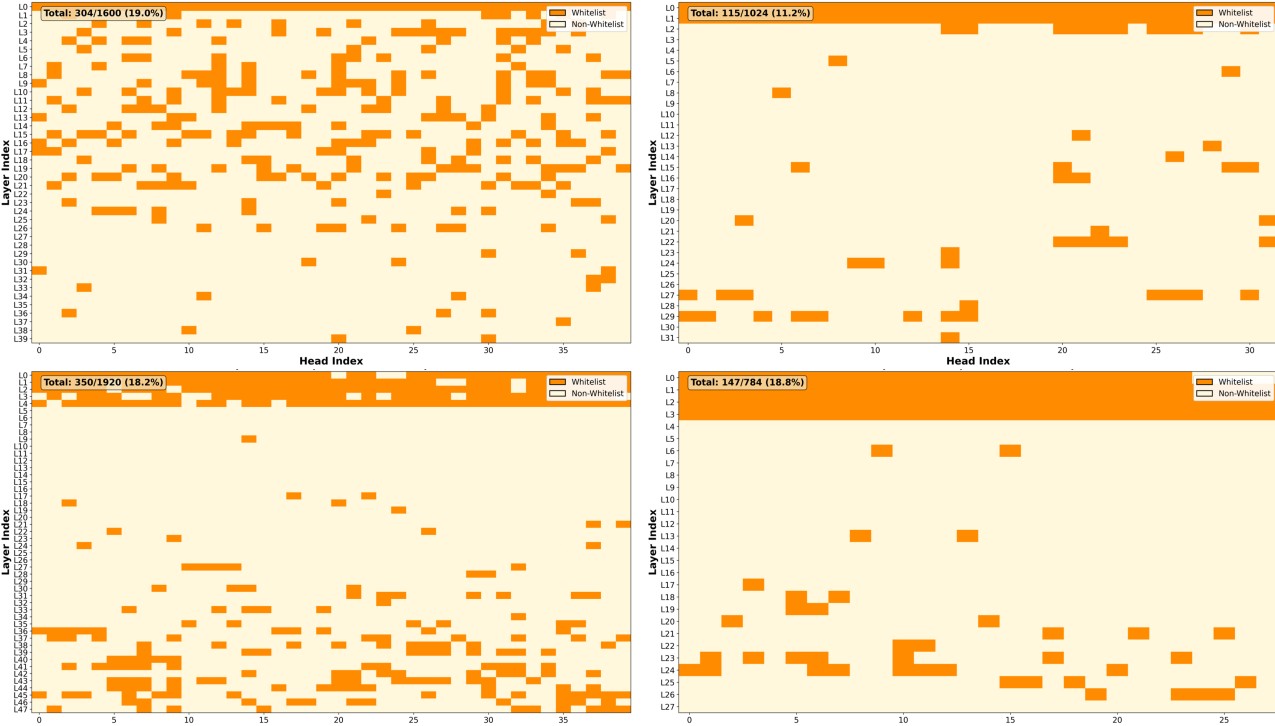

*Figure 10.* **Whitelist binary mask across all models.** From left to right, top to bottom: LLaMA-2-13B, LLaMA-3.1-8B, Qwen2.5-14B, Qwen2.5-7B. Binary masks showing whitelist heads across all layers and heads for each model. Whitelist heads exhibit significantly lower domain variance than non-whitelist heads, confirming their domain-invariant nature. The visualization confirms that whitelist heads are distributed across all layers, with a slight concentration in middle layers where general knowledge processing is most prominent.

Figure 12 shows the domain importance correlation matrices for all four models. The correlation matrices reveal that selected domains exhibit low correlation (typically $|r| < 0.3$, with most values close to 0), confirming that different domains rely on distinct sets of attention heads.

**Interpretation of low correlation values.** Correlation values close to 0 indicate that the head importance patterns of different domains are *statistically independent*—there is no linear relationship between which heads are important for one domain versus another. This is exactly what we expect for orthogonal domains: each domain should activate a different, non-overlapping set of attention heads. If two domains had high positive correlation ($r \approx 1$), it would mean they share similar head importance patterns, suggesting they are not truly orthogonal in the model's representation space. If they had high negative correlation ($r \approx -1$), it would mean their head importance patterns are inversely related, which is also inconsistent with orthogonality.

The fact that correlations are close to 0 (rather than being exactly 0) reflects natural variation in the data and confirms that: (1) **Each domain has its own distinct head importance pattern**: The probes successfully identify domain-specific activation patterns, as evidenced by the non-zero importance values for each domain. (2) **These patterns are independent across domains**: The low correlation confirms that the head importance patterns for different domains are not linearly related, validating that domains are encoded in separable subspaces. (3) **The independence is consistent across models**: The consistently low correlations across all four models demonstrate that our DBS-selected domains successfully capture distinct knowledge subspaces that are encoded in separable attention head pathways, regardless of model architecture.

This validates our hypothesis that semantic orthogonality (measured by DBS in the embedding space) corresponds to representational orthogonality (measured by head importance patterns) in the model's internal representations. The probes' ability to identify these distinct patterns confirms that domain-specific knowledge is indeed encoded in separable subspaces, as our theoretical framework predicts.

**Probe weight analysis.** The learned probe weights reveal several key insights about domain-specific activation patterns: (1) **Sparsity pattern**: The weight matrices reveal that probes learn sparse, domain-specific patterns, where each domain is associated with a distinct set of activation dimensions with minimal overlap between domains. This sparsity confirms that

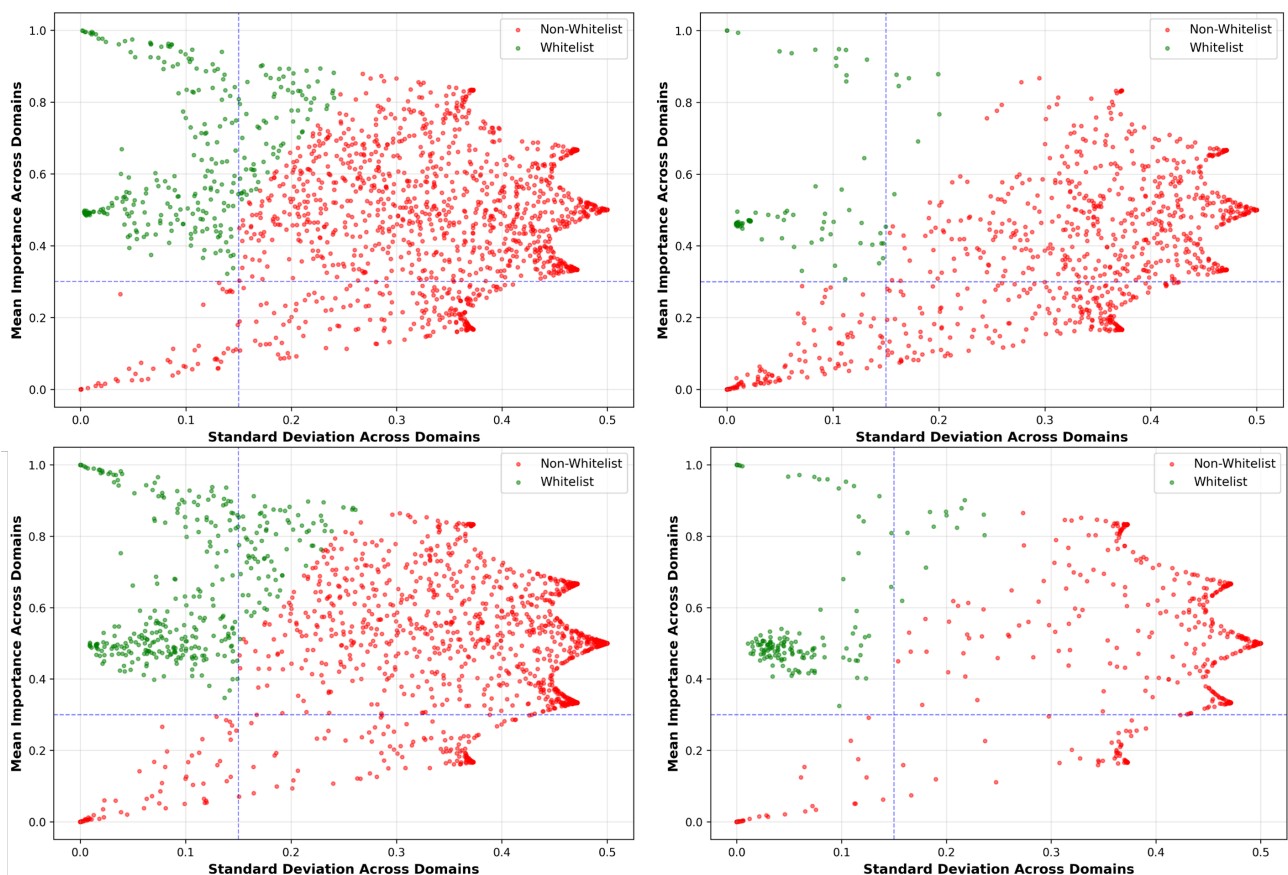

*Figure 11.* **Whitelist selection criteria scatter plot across all models.** From left to right, top to bottom: LLaMA-2-13B, LLaMA-3.1-8B, Qwen2.5-14B, Qwen2.5-7B. Scatter plots showing standard deviation versus mean importance for whitelist selection. Whitelist heads (green) exhibit low standard deviation (low domain variance) and high mean importance, while non-whitelist heads (red) show higher variance across domains. The visualization demonstrates that our whitelist identification successfully distinguishes domain-invariant heads from domain-specific heads based on their statistical properties.

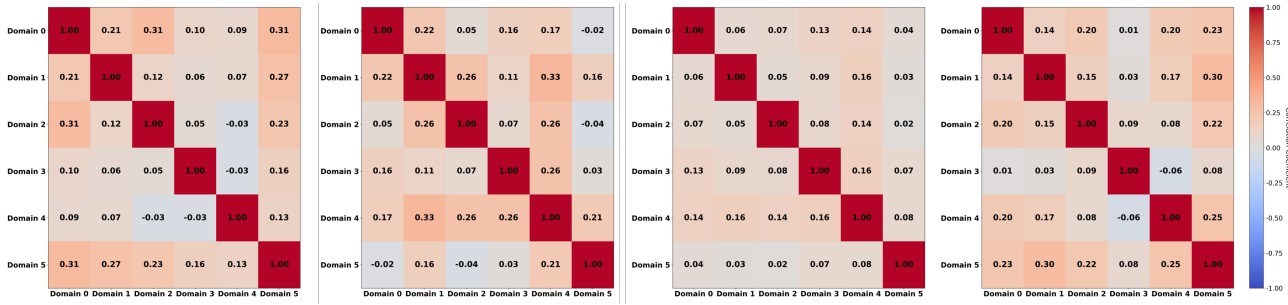

*Figure 12.* **Domain importance correlation matrix across all models.** From left to right: LLaMA-2-13B, LLaMA-3.1-8B, Qwen2.5-14B, Qwen2.5-7B. Correlation matrices showing the Pearson correlation coefficient between head importance patterns for each pair of domains. Each cell $(i, j)$ represents the correlation between domain $i$'s and domain $j$'s head importance vectors (collected across all layers and heads). Low correlation values ($< 0.3$) indicate that domains rely on distinct sets of attention heads, confirming that semantic orthogonality (from DBS) corresponds to representational orthogonality in the model's internal representations. The consistently low correlations across all models validate that our DBS-selected domains capture distinct knowledge subspaces encoded in separable attention head pathways.

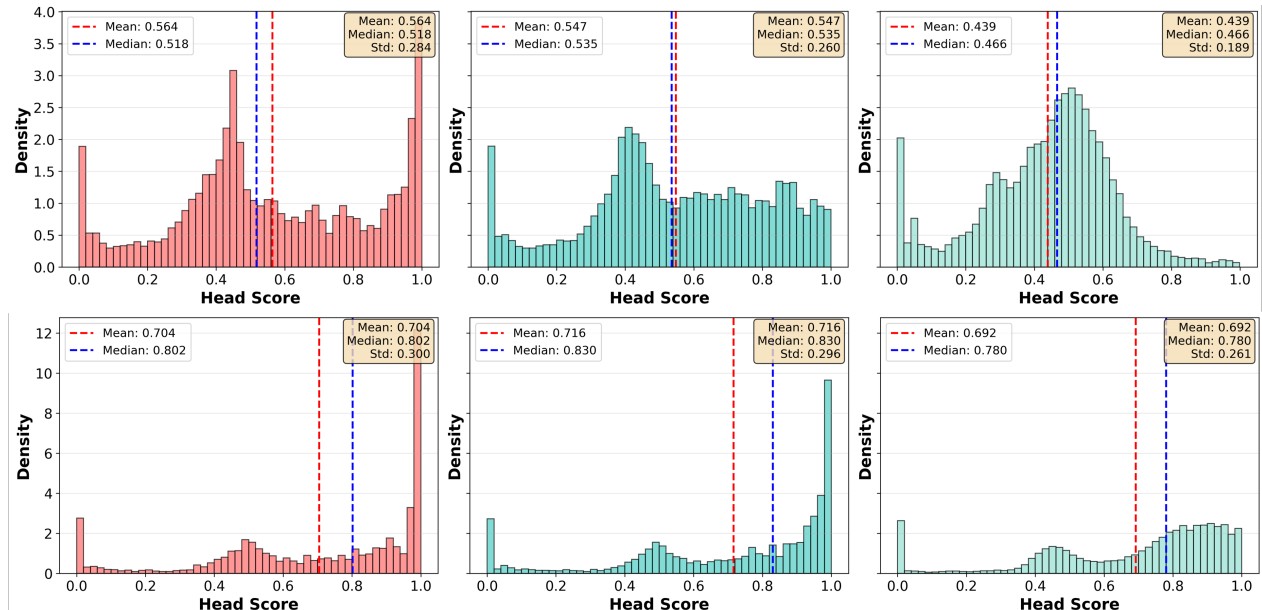

*Figure 13.* **Head score distribution across different dataset types.** From top to bottom: LLaMA-3.1-8B, Qwen2.5-7B. From left to right: selected-domain (selected), out-of-domain (OOD), cross-domain. Each subplot shows the distribution of head scores for that dataset type, computed using Equation (8) which combines domain similarity probabilities with cached head importance $I_{\ell,h,k}$. The distributions reveal how head scores vary based on dataset type, with selected-domain datasets showing more concentrated distributions and cross-domain datasets showing more spread-out distributions. The statistical information (mean, median, standard deviation) is shown in each subplot.

domain knowledge is encoded in separable subspaces, validating our DBS approach of constructing orthogonal domain axes. (2) **Dimension importance**: The magnitude and sign of weights indicate which activation dimensions contribute positively or negatively to domain identification, revealing the internal structure of domain-specific representations. (3) **Layer-wise variation**: Comparing weights across layers shows how domain-specific patterns evolve through the network, with earlier layers showing more mixed patterns and deeper layers exhibiting clearer domain separation. (4) **Cross-domain relationships**: The weight patterns reveal which domains share similar activation dimensions (indicating related knowledge) and which domains are most distinct (confirming orthogonality). This analysis demonstrates that probes successfully learn interpretable, domain-specific patterns that align with our theoretical framework of separable knowledge subspaces.

### M.3 PSP Online Step Effects

Probe-based Subspace Pruning (PSP) online compilation uses the offline-prepared probes and importance statistics to compile scenario-level pathway masks from scenario-start inputs. This subsection analyzes how online compilation adapts to different scenario types, how domain mixture and knowledge breadth affect mask compilation, and how pruning strength modulates the final mask patterns.

**scenario-level domain mixture and knowledge breadth.** During online compilation, we aggregate probe outputs over the scenario prefix to compute a scenario-level domain mixture $q_k(s)$ for each domain $k$. The domain mixture quantifies how much each domain is needed for the current scenario, with higher values indicating stronger domain alignment. We further compute knowledge breadth $c(s) \in [0, 1]$ using normalized entropy over the domain mixture, where $c(s) \approx 0$ indicates a narrow (near selected-domain) scenario and $c(s) \approx 1$ indicates a broad (cross-domain) scenario.

The knowledge breadth $c(s)$ directly modulates the effective keep-ratio $K(s)$ via Equation (9), ensuring that narrow scenarios use fewer heads while broad scenarios preserve more heads. This adaptive behavior is critical for maintaining performance across diverse scenario types while maximizing efficiency.

Appendix Q provide concrete examples of how scenario-level domain mixture $q_k(s)$ is computed and visualized for an out-of-domain sample. The case study shows the complete inference process, including domain mixture evolution, mask compilation, and layer-wise probe outputs, demonstrating how online compilation adapts to different scenario types in practice.

**Head score distribution.** After computing the scenario-level domain mixture, we score each head $(\ell, h)$ using Equation (8),

*Table 7.* **Performance comparison of SUBSPACEPATH PRUNER at different pruning ratios for LLaMA-3.1-8B-Instruct across various datasets.** Token-level Recall (Recall) values shown as mean ± standard error of the mean (SEM). Retention = $(\text{Recall}_{\text{pruned}}/\text{Recall}_{\text{dense}}) \times$ Speedup. Pruning ratios: left value (baseline), right value (ours, average); ours prunes more aggressively.

| Category Method | | XdomainBench | | | Retention | Cross-dataset Test | | | Retention |
|---|---|---|---|---|---|---|---|---|---|
| | | Selected | OOD | Cross | (S/O/C) | CSQA | NQ | ARC | (C/N/A) |
| **Dense Model** | | | | | | | | | |
| LLaMA-3.1-8B | | 39.1±1.57 | 33.2±1.21 | 21.3±0.81 | 1.00 / 1.00 / 1.00 | 27.77±1.29 | 28.82±1.09 | 22.28±0.92 | 1.00 / 1.00 / 1.00 |
| **Moderate Pruning Ratio** | | | | | | | | | |
| | | 89%~88.6% | 90%~89.6% | 88%~87.8% | | 89%~88.9% | 86%~85.4% | 86%~85.6% | |
| Unstruct. | DaSS | 24.50±1.42 | 26.07±1.14 | 18.77±0.76 | 0.38 / 0.50 / 0.49 | 22.45±1.21 | 10.79±0.74 | 18.06±0.88 | 0.72 / 0.30 / 0.87 |
| | Wanda | 16.17±1.18 | 25.35±1.12 | 18.04 ±0.76 | 0.17 / 0.34 / 0.31 | 26.43±1.28 | 18.89 ±0.98 | 20.13±0.87 | 0.87 / 0.24 / 0.69 |
| Struct. | LLM-Pr. | **34.09**±1.58 | 27.68±1.15 | 19.36±0.78 | 0.53 / 0.51 / 0.55 | 25.97±1.28 | 21.74±1.05 | 21.37±0.91 | 0.57 / 0.47 / 0.58 |
| Activ. | RIA | 16.65±1.20 | 24.89±1.12 | 17.76±0.74 | 0.12 / 0.26 / 0.22 | 25.33±1.26 | 18.65±0.97 | 20.13±0.87 | 0.62 / 0.17 / 0.58 |
| | Probe Pr. | 28.59±1.57 | 25.58±1.24 | 18.30±1.10 | 0.56 / 0.59 / 0.65 | 22.77±1.33 | 19.08±1.24 | 19.94±1.26 | 0.62 / 0.51 / 0.69 |
| **Ours-SubspacePath** | | **33.00**±2.30 | **35.0**±1.90 | **28.40**±3.40 | 0.99 / 0.78 / 1.05 | 24.68±1.26 | **32.02**±1.34 | **25.64**±1.17 | 1.13 / 1.20 / 3.98 |
| **Aggressive Pruning Ratio** | | | | | | | | | |
| | | 84%~83.8% | 81%~80.3% | 78%~77.6% | | 84%~83.7% | 79%~78.5% | 77%~76.7% | |
| Unstruct. | DaSS | 21.72±1.35 | 23.77±1.10 | 14.43±0.69 | 0.38 / 0.56 / 0.23 | 17.60±1.10 | 8.28±0.64 | 15.32±0.78 | 0.73 / 0.17 / 0.53 |
| | Wanda | 11.49±1.02 | 23.15±1.08 | 4.98±0.41 | 0.02 / 0.22 / 0.04 | **21.58**±1.17 | 8.03±0.68 | 10.87±0.69 | 0.71 / 0.03 / 0.09 |
| Struct. | LLM-Pr. | **28.18**±1.48 | 24.60±1.10 | 15.92±0.71 | 0.45 / 0.47 / 0.48 | 25.68±1.26 | 19.04±1.00 | **20.44**±0.90 | 0.58 / 0.42 / 0.59 |
| Activ. | RIA | 11.94±1.05 | 22.51±1.06 | 4.78±0.40 | 0.02 / 0.20 / 0.04 | 21.43±1.18 | 7.67±0.65 | 11.16±0.68 | 0.69 / 0.03 / 0.09 |
| | Probe Pr. | 24.84±1.50 | 22.90±1.20 | 16.16±1.05 | 0.48 / 0.53 / 0.59 | 20.93±1.29 | 15.92±1.16 | 19.11±1.24 | 0.58 / 0.43 / 0.67 |
| **Ours-SubspacePath** | | **27.10**±2.20 | **30.40**±1.90 | **20.20**±3.00 | 0.85 / 0.61 / 0.62 | 21.33±1.20 | **23.75**±1.22 | 20.23±1.07 | 1.09 / 0.91 / 2.85 |

which combines the effective domain similarities $s_k^{\text{eff}}$ with cached importance $I_{\ell,h,k}$.

The mask compilation process selects the top-$K(s)$ heads (after preserving the whitelist) based on these scores, producing a binary mask $m_s$ that is cached and reused for all subsequent inputs in the scenario.

Figure 13 shows the head score distributions across different dataset types (selected-domain/selected, out-of-domain, and cross-domain) for two models (LLaMA-3.1-8B and Qwen2.5-7B), computed using Equation (8) which combines domain similarity probabilities with cached head importance $I_{\ell,h,k}$. The distributions are computed from experimental data collected across multiple pruning strength configurations ($\eta \in \{0.3, 0.5, 0.8\}$) and cross-dataset evaluations. The visualization reveals how head scores vary based on dataset type: selected-domain datasets show more concentrated score distributions as they primarily activate domain-specific heads, while cross-domain datasets show more spread-out distributions reflecting the need to handle multiple domains simultaneously. The statistical information (mean, median, standard deviation) displayed in each subplot is computed directly from the experimental head scores.

**Mask patterns under different pruning strengths.** The pruning strength hyperparameter $\eta \in (0, 1]$ globally scales the effective keep-ratio $K(s)$ via Equation (9), allowing fine-grained control over the trade-off between efficiency and performance. The mask compilation process ensures that whitelist heads are always preserved regardless of pruning strength (as established in the offline phase), and domain-specific heads are selectively preserved based on scenario domain mixture, with more relevant domains preserving more heads. This adaptive mask compilation ensures that each scenario receives a customized pathway mask that matches its knowledge requirements while respecting the budget constraint, enabling efficient inference across diverse scenario types. The effectiveness of this adaptive approach is demonstrated in the main experimental results (Section 5), where PSP maintains performance across different scenario types while achieving significant efficiency gains.

# N   Additional Results

This section provides additional experimental results that complement the main results in the paper, including results for other models, different pruning ratios, pruning time comparison, and task-type breakdown.

## N.1   Other Model Results

This subsection presents complete results for models not shown in the main paper. Tables 7, 8, and 9 report the performance comparison for Qwen2.5-7B, LLaMA-3.1-8B, and Qwen2.5-14B respectively across various datasets and pruning ratios.

*Table 8.* **Performance comparison of SUBSPACEPATH PRUNER at different pruning ratios for Qwen2.5-7B-Instruct across various datasets.** Token-level Recall (Recall) values shown as mean ± standard error of the mean (SEM). Retention = (Recall_pruned/Recall_dense) × Speedup. Pruning ratios: left value (baseline), right value (ours, average); ours prunes more aggressively.

| Category | Method | XdomainBench | | | Retention | Cross-dataset Test | | | Retention |
|---|---|---|---|---|---|---|---|---|---|
| | | **Selected** | **OOD** | **Cross** | **(S/O/C)** | **CSQA** | **NQ** | **ARC** | **(C/N/A)** |
| | | | | | **Dense Model** | | | | |
| Qwen2.5-7B | | 40.8±1.6 | 33.6±1.2 | 22.9±0.8 | 1.00 / 1.00 / 1.00 | 27.75±1.31 | 17.51±0.89 | 21.64±0.93 | 1.00 / 1.00 / 1.00 |
| | | | | | **Moderate Pruning Ratio** | | | | |
| | | 82%~81.3% | 85%~84.9% | 85%~84.0% | | 90%~89.4% | 83%~82.4% | 87%~86.9% | |
| Unstruct. | DaSS | 33.53±1.58 | 30.56±1.20 | 19.99±0.78 | 0.21 / 0.22 / 0.33 | 21.87±1.21 | 9.75±0.72 | 17.59±0.88 | 1.44 / 0.28 / 0.85 |
| | Wanda | 28.82±1.49 | 2.30±0.42 | 18.76±0.74 | 0.05 / 0.00 / 0.26 | 0.1±0.07 | 13.3±0.85 | 18.61±0.89 | 0.00 / 0.52 / 0.61 |
| Struct. | LLM-Pr. | 32.02±1.56 | 29.20±1.21 | 16.12±0.70 | 0.75 / 0.83 / 0.68 | 18.85±1.15 | 10.94±0.80 | 17.94±0.87 | 0.66 / 0.61 / 0.79 |
| | RIA | 28.71 ±1.48 | 2.04±0.40 | 18.41±0.75 | 0.06 / 0.00 / 0.34 | 0.05±0.05 | 13.65±0.86 | 18.79±0.88 | 0.00 / 0.75 / 0.82 |
| Activ. | Probe Pr. | 30.29±1.59 | 28.63±1.29 | 14.13±0.99 | 0.89 / 1.02 / 0.74 | 18.80±1.24 | 11.63±1.01 | 17.87±1.21 | 0.81 / 0.80 / 0.99 |
| **Ours-SubspacePath** | | **46.40±2.50** | **41.00±2.00** | **27.20±3.30** | 1.00 / 0.73 / 0.63 | **21.90±1.23** | **19.51±1.14** | **25.25±1.18** | 5.85 / 3.65 / 7.64 |
| | | | | | **Aggressive Pruning Ratio** | | | | |
| | | 74%~73.7% | 78%~77.5% | 78%~77.9% | | 82%~81.4% | 75%~74.9% | 76%~75.6% | |
| Unstruct. | DaSS | 31.11±1.56 | 30.78±1.21 | 21.69±0.78 | 0.20 / 0.24 / 0.27 | **18.57±1.12** | 7.10±0.59 | 17.98±0.86 | 1.27 / 0.18 / 0.71 |
| | Wanda | 3.90±0.67 | 26.71±1.15 | 1.49±0.34 | 0.00 / 0.10 / 0.01 | 23.15±1.23 | 0.60±0.16 | 2.09±0.23 | 0.85 / 0.00 / 0.02 |
| Struct. | LLM-Pr. | 23.32±1.42 | 24.03±1.14 | 14.42±0.71 | 0.39 / 0.48 / 0.43 | 15.87±1.06 | 8.37±0.70 | 14.14±0.79 | 0.38 / 0.32 / 0.44 |
| | RIA | 3.39±0.63 | 26.84±1.15 | 1.59±0.35 | 0.00 / 0.10 / 0.01 | 23.30±1.23 | 1.01±0.26 | 2.45±0.27 | 0.92 / 0.01 / 0.02 |
| Activ. | Probe Pr. | 19.24±1.37 | 17.19±1.07 | 8.47±0.79 | 0.39 / 0.42 / 0.30 | 11.58±1.01 | 7.64±0.84 | 11.21±1.00 | 0.34 / 0.38 / 0.42 |
| **Ours-SubspacePath** | | **36.50±2.40** | **36.40±2.00** | **22.50±3.10** | 0.36 / 0.45 / 0.45 | 17.65±1.14 | **15.33±1.03** | **20.62±1.12** | 2.88 / 2.52 / 2.43 |

*Table 9.* **Performance comparison of SUBSPACEPATH PRUNER at different pruning ratios for Qwen2.5-14B-Instruct across various datasets.** Token-level Recall (Recall) values shown as mean ± standard error of the mean (SEM). Retention = (Recall_pruned/Recall_dense) × Speedup. Pruning ratios: left value (baseline), right value (ours, average); ours prunes more aggressively.

| Category | Method | XdomainBench | | | Retention | Cross-dataset Test | | | Retention |
|---|---|---|---|---|---|---|---|---|---|
| | | **Selected** | **OOD** | **Cross** | **(S/O/C)** | **CSQA** | **NQ** | **ARC** | **(C/N/A)** |
| | | | | | **Dense Model** | | | | |
| Qwen2.5-14B | | 40.9±1.60 | 37.2±1.25 | 22.8±0.85 | 1.00 / 1.00 / 1.00 | 28.42±1.30 | 19.72±0.95 | 20.71±0.92 | 1.00 / 1.00 / 1.00 |
| | | | | | **Moderate Pruning Ratio** | | | | |
| | | 83%~82.2% | 85%~84.4% | 83%~82.4% | | 85%~84.2% | 81%~80.3% | 82%~81.5% | |
| Unstruct. | DaSS | 40.62±1.65 | 35.17±1.26 | 21.85±0.81 | 0.56 / 0.47 / 0.45 | 24.22±1.24 | 10.87±0.78 | 20.38±0.92 | 1.10 / 0.31 / 1.05 |
| | Wanda | 35.22±1.57 | 37.50±1.25 | 19.35±0.79 | 0.13 / 0.23 / 0.18 | **29.30±1.32** | 18.60±1.02 | 21.06±0.91 | 0.90 / 0.29 / 0.46 |
| Struct. | LLM-Pr. | 42.97±1.65 | 37.23±1.27 | 22.00±0.84 | 1.01 / 0.96 / 0.93 | 28.50±1.31 | 19.63±1.04 | 20.37±0.92 | 0.97 / 0.97 / 0.94 |
| | RIA | 35.69±1.58 | 38.07±1.26 | 19.20±0.79 | 0.14 / 0.24 / 0.18 | 28.87±1.32 | 18.56±1.00 | 21.35±0.90 | 0.87 / 0.31 / 0.47 |
| Activ. | Probe Pr. | 42.52±1.72 | 37.41±1.38 | 22.18±1.19 | 1.14 / 1.11 / 1.07 | 28.28±1.42 | 19.20±1.25 | 20.31±1.27 | 1.09 / 1.07 / 1.08 |
| **Ours-SubspacePath** | | **47.80±2.50** | **44.10±2.00** | **31.30±3.50** | 1.36 / 1.02 / 1.47 | 24.87±1.28 | **27.21±1.27** | **28.58±1.20** | 3.48 / 2.36 / 4.29 |
| | | | | | **Aggressive Pruning Ratio** | | | | |
| | | 73%~72.8% | 72%~71.7% | 70%~69.4% | | 73%~72.6% | 70%~69.4% | 71%~70.4% | |
| Unstruct. | DaSS | 37.10±1.61 | 33.39±1.25 | 21.00±0.81 | 0.81 / 1.00 / 0.85 | 21.28±1.19 | 8.49±0.65 | 19.83±0.90 | 1.94 / 0.34 / 1.14 |
| | Wanda | 24.25±1.40 | 34.10±1.21 | 13.20±0.68 | 0.07 / 0.24 / 0.15 | 28.45±1.31 | 14.00±0.90 | 19.58±0.89 | 1.11 / 0.26 / 0.44 |
| Struct. | LLM-Pr. | 38.04±1.61 | 36.49±1.26 | 22.42±0.85 | 0.60 / 0.65 / 0.65 | 26.08±1.26 | 20.47±1.04 | 24.22±0.92 | 0.60 / 0.67 / 0.77 |
| | RIA | 22.84±1.36 | 34.02±1.21 | 13.01±0.67 | 0.07 / 0.24 / 0.15 | 29.05±1.31 | 13.94±0.89 | 19.78±0.88 | 1.07 / 0.26 / 0.47 |
| Activ. | Probe Pr. | 25.54±1.51 | 25.39±1.24 | 18.16±1.10 | 0.45 / 0.51 / 0.59 | 23.07±1.33 | 16.39±1.17 | 21.48±1.30 | 0.60 / 0.62 / 0.78 |
| **Ours-SubspacePath** | | **36.50±2.40** | **35.80±2.00** | **23.70±3.20** | 0.48 / 0.46 / 0.71 | 24.08±1.26 | **23.23±1.21** | **27.98±1.18** | 1.23 / 0.87 / 1.87 |

**Analysis.** Our method demonstrates consistent effectiveness across different model architectures and scales. At moderate pruning ratios, all three models maintain strong performance on selected-domain tasks (retention ≥ 0.99) while achieving significant improvements on cross-dataset evaluations, with retention values ranging from 1.13–7.64× depending on the model and dataset. Notably, Qwen2.5-7B shows exceptional cross-dataset retention (5.85× on CSQA, 3.65× on NQ, 7.64× on ARC), highlighting the robustness of our domain-adaptive approach. Larger models (Qwen2.5-14B) exhibit better absolute Recall scores and more stable retention across different evaluation settings, suggesting that our method scales favorably with model capacity. Under aggressive pruning, performance degrades more gradually for larger models, with

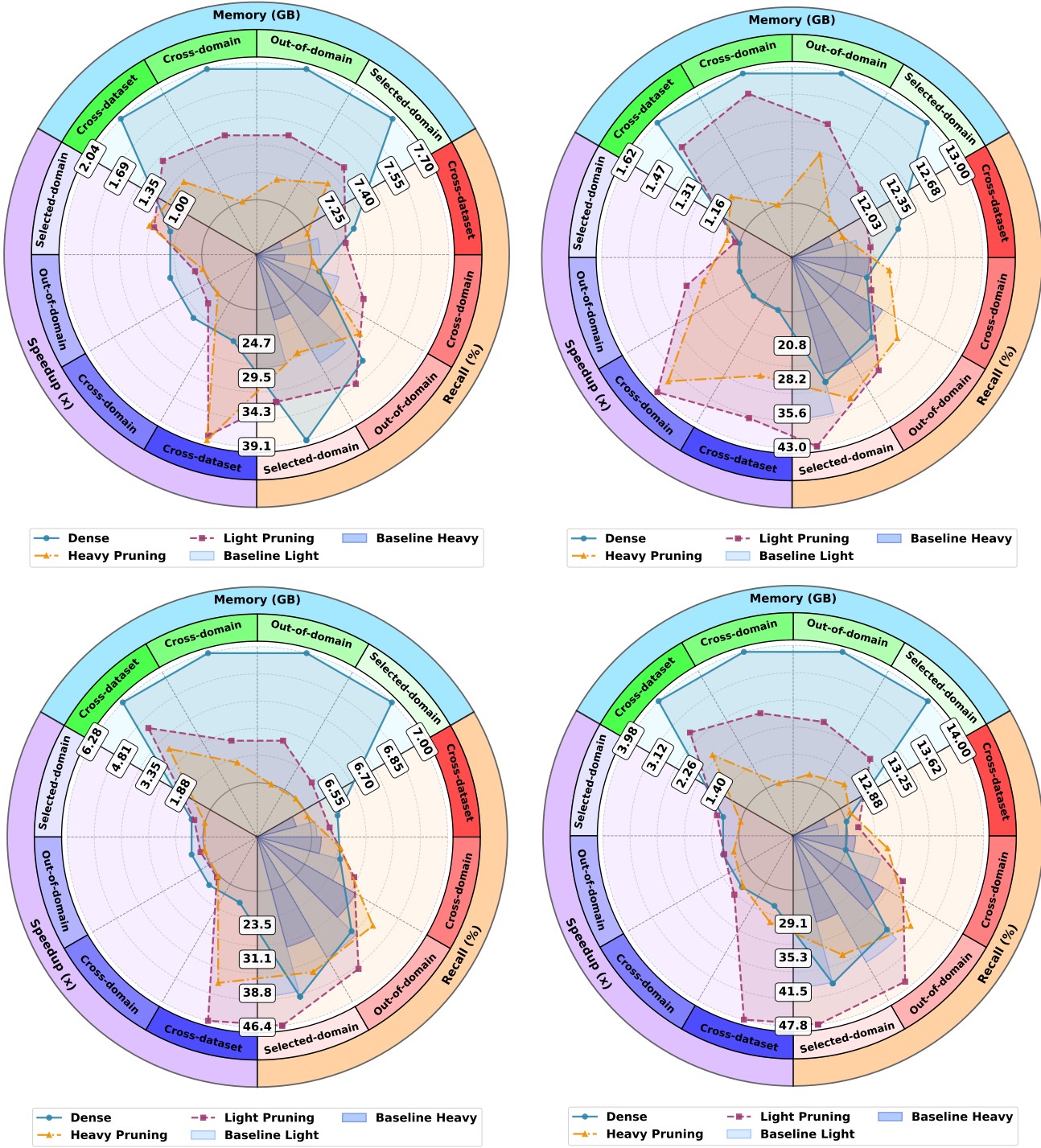

*Figure 14.* **Radar charts for all four models.** Top-left: LLaMA-3.1-8B. Top-right: LLaMA-2-13B. Bottom-left: Qwen2.5-7B. Bottom-right: Qwen2.5-14B. Each radar chart shows three configurations (dense baseline, light pruning, heavy pruning) across three metric regions (Recall at top-left, Memory at bottom, Speedup at top-right), with each region displaying four evaluation settings (Selected, OOD, Cross-domain, Cross-dataset).

Qwen2.5-14B maintaining higher retention (0.48–1.87×) compared to smaller models, indicating better pruning resilience for larger architectures.

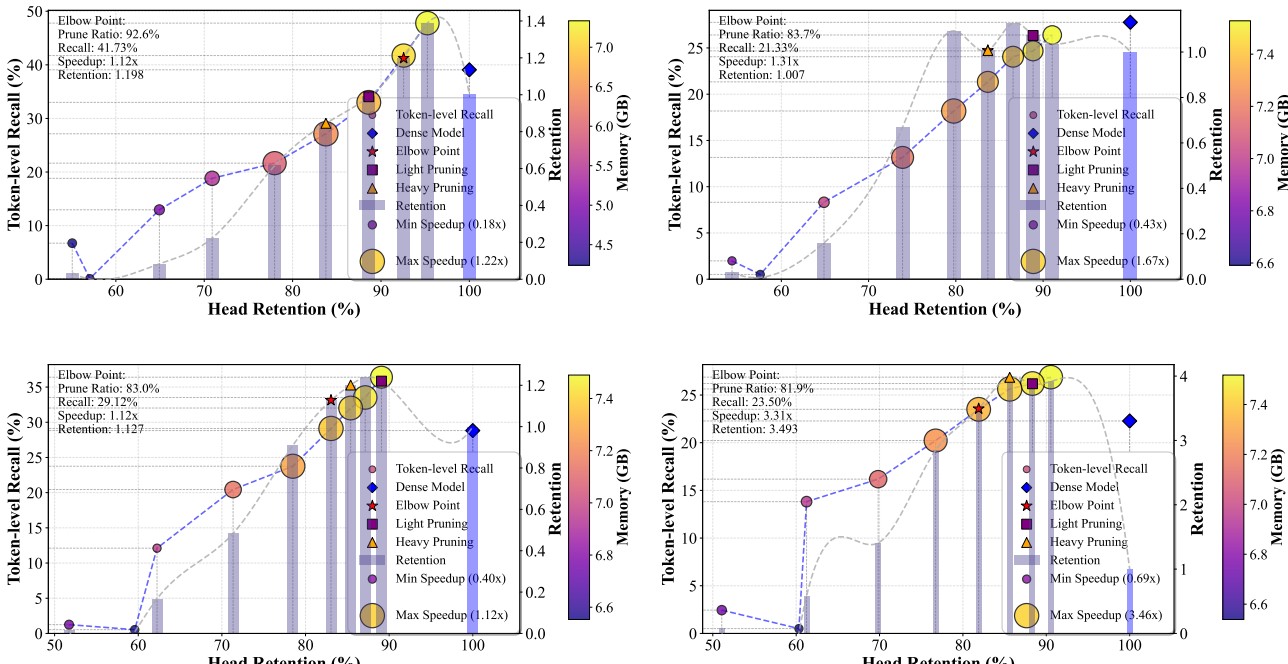

*Figure 15.* **Pruning ratio vs Performance for LLaMA-3.1-8B.** under different pruning strengths compared to dense model. Top-left: XDomainBench (averaged). Top-right: CommonsenseQA. Bottom-left: Natural Questions. Bottom-right: ARC.

## N.2    Robustness Analysis

This subsection presents comprehensive radar chart visualizations for all four models, comparing dense baseline, light pruning, and heavy pruning configurations across three key metrics: Recall, Memory, and Speedup. Each radar chart in Figure 14 displays four evaluation regions (Selected-domain, OOD, Cross-domain, Cross-dataset) for each metric (Recall, Memory, and Speedup), providing a holistic view of the performance-efficiency trade-offs.

**Analysis.**    The radar charts reveal several key insights across all models. First, light pruning generally maintains or improves Recall scores across most evaluation settings while achieving moderate speedup gains, particularly for larger models (LLaMA-2-13B and Qwen2.5-14B). Second, heavy pruning shows more aggressive speedup improvements but with varying Recall retention depending on the model and evaluation setting. Third, memory usage decreases proportionally with pruning strength, with models using approximately 50–95% of original memory depending on the pruning configuration. Fourth, Cross-domain and Cross-dataset settings often show the most significant improvements under pruning, highlighting the effectiveness of our domain-adaptive approach for out-of-distribution scenarios.

## N.3    Pruning Ratio Trade-off

This subsection presents trade-off curves between pruning ratio and performance for all four models.

**Different Pruning Ratios curves.**    Figures 15, 16, 17, and 18 show the trade-off between pruning ratio and performace (Recall, Speedup, Memory, Retention) for each model. As pruning increases, Recall exhibits an elbow-shaped degradation, consistent with the main text findings.

**Analysis.**    The trade-off curves reveal consistent patterns across all models and datasets. Recall performance degrades gradually at low pruning ratios ($\eta < 0.4$) but accelerates beyond the elbow point ($\eta \approx 0.5$–$0.6$), where aggressive pruning begins to significantly impact accuracy. Speedup and memory reduction scale nearly linearly with pruning ratio, providing predictable efficiency gains. Retention values peak at moderate pruning ratios ($\eta \approx 0.5$–$0.6$), achieving optimal balance between performance preservation and efficiency improvement. Cross-dataset evaluations (CSQA, NQ, ARC) consistently show higher retention than selected-domain tasks, demonstrating the robustness of our domain-adaptive pruning approach. Larger models (LLaMA-2-13B, Qwen2.5-14B) exhibit more gradual degradation curves, indicating better pruning resilience compared to smaller models.

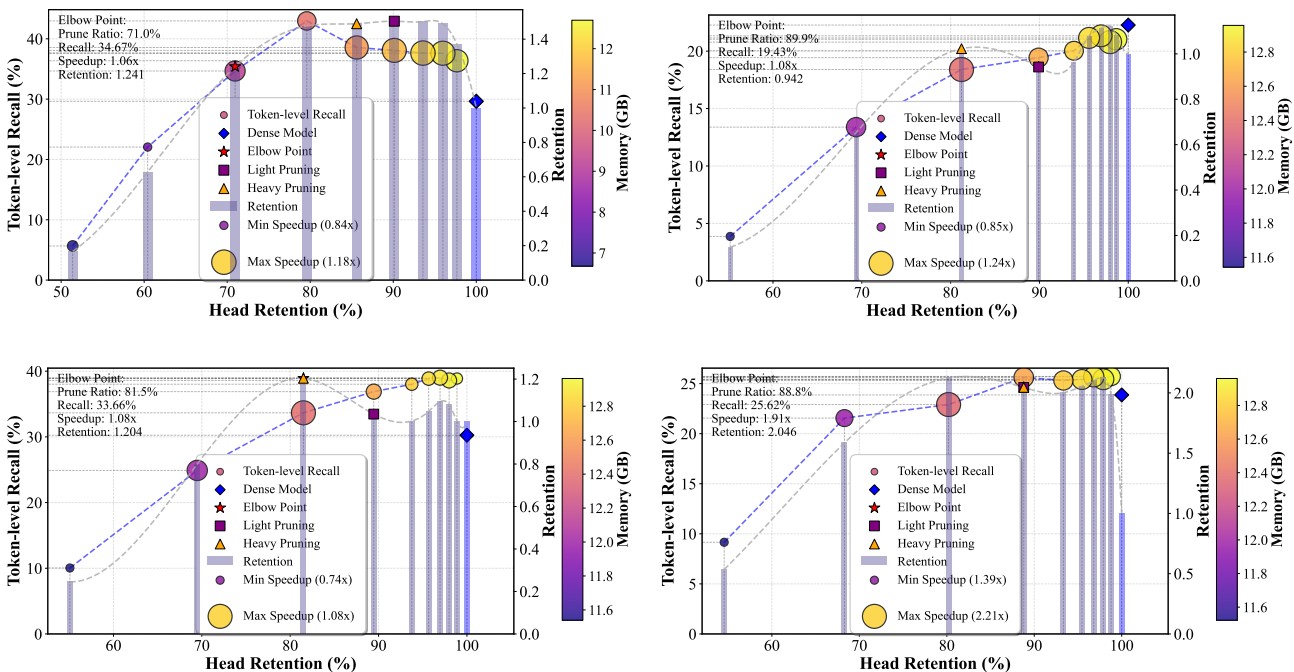

*Figure 16.* **Pruning ratio vs Recall for LLaMA-2-13B** under different pruning strengths compared to dense model. Top-left: XDomain-Bench (averaged). Top-right: CommonsenseQA. Bottom-left: Natural Questions. Bottom-right: ARC.

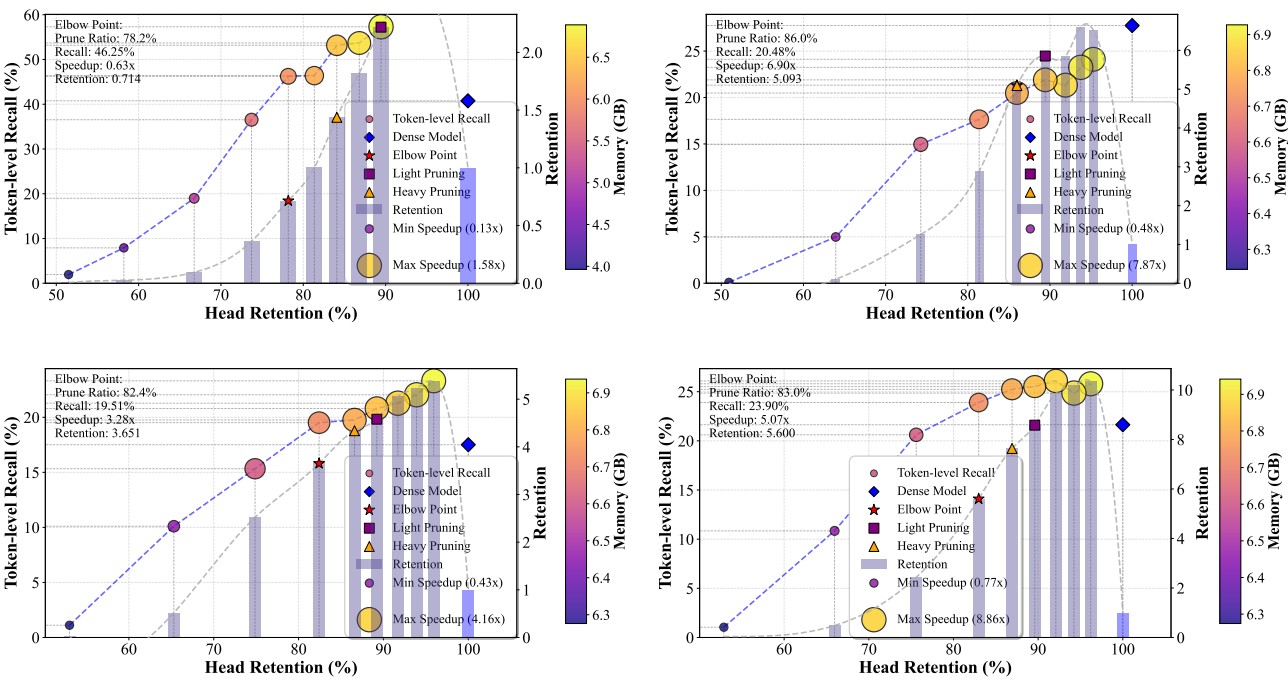

*Figure 17.* **Pruning ratio vs Recall for Qwen2.5-7B** under different pruning strengths compared to dense model. Top-left: XDomainBench (averaged). Top-right: CommonsenseQA. Bottom-left: Natural Questions. Bottom-right: ARC.

## N.4 Task Type Analysis

This subsection provides detailed analysis by task type (multiple_choice, factual, reasoning, code) across different models and methods. For each task type, we report Token-level Recall (Recall) metrics. This analysis reveals how different task types respond to pruning, and how our method maintains performance across diverse task types compared to baseline

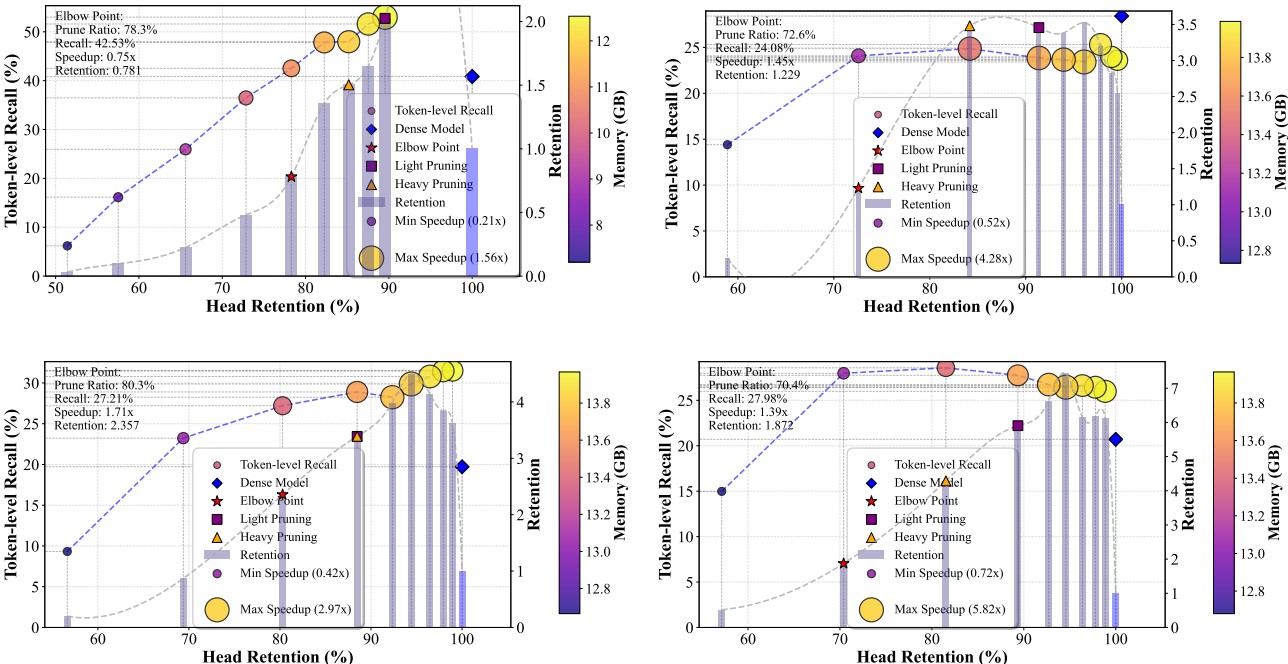

*Figure 18.* **Pruning ratio vs Recall for Qwen2.5-14B** under different pruning strengths compared to dense model. Top-left: XDomain-Bench (averaged). Top-right: CommonsenseQA. Bottom-left: Natural Questions. Bottom-right: ARC.

methods.

**Task-type breakdown by pruning ratio.** Table 10 provides a detailed breakdown of performance by task type (multiple_choice, factual, reasoning, code) across different pruning ratios for all four models. The tables show results for dense models (no pruning) and our method at different pruning ratios ($\eta$ from 0.1 to 0.9). Each row corresponds to a different pruning ratio, with the dense model results shown at the top.

**Analysis.** Table 10 reveals distinct pruning sensitivity patterns across task types. At light pruning ratios ($\eta \geq 0.6$), our method consistently outperforms dense models across most task types, with multiple_choice and factual tasks showing the most significant improvements (e.g., Qwen2.5-7B: +19.8 Recall for multiple_choice at $\eta = 0.9$). Code tasks also benefit substantially from light pruning, suggesting that removing noise heads enhances code-related reasoning. However, reasoning tasks exhibit higher sensitivity to pruning, with performance degradation occurring earlier (around $\eta = 0.5$–0.6) compared to other task types. As pruning becomes more aggressive ($\eta \leq 0.3$), all task types show rapid performance decline, with reasoning tasks being most severely affected. Notably, larger models (LLaMA-2-13B, Qwen2.5-14B) maintain better performance at moderate pruning ratios ($\eta = 0.4$–0.5) compared to smaller models, indicating better pruning resilience with increased model capacity. The optimal pruning ratio varies by task type: multiple_choice and factual tasks tolerate more aggressive pruning, while reasoning tasks require more conservative pruning to maintain performance.

**Task-type breakdown at light and heavy pruning ratios.** Table 11 provides a detailed breakdown of performance by task type at both light and heavy pruning ratios. For light pruning, we use approximately 89% head retention (11% pruning ratio). For heavy pruning, the ratios vary by split: Selected-domain uses approximately 85% head retention (15% pruning), OOD uses approximately 86% head retention (14% pruning), and Cross-domain uses approximately 73% head retention (27% pruning).

**Analysis.** Table 11 demonstrates our method's effectiveness across different task types at both light and heavy pruning ratios. Under light pruning, our method consistently outperforms dense baselines across all four models, with particularly strong gains in multiple_choice tasks (e.g., Qwen2.5-7B: +10.5 Recall, Qwen2.5-14B: +11.2 Recall) and factual tasks (e.g., LLaMA-2-13B: +13.8 Recall, LLaMA-3.1-8B: +7.4 Recall). Code tasks also show substantial improvements, indicating that domain-aware pruning effectively preserves code-related knowledge pathways. At heavy pruning, performance gains are more modest but still positive for most task types, with factual tasks maintaining the strongest relative performance (e.g.,

*Table 10.* **Task-type breakdown by pruning ratio.** Performance breakdown by task type (multiple_choice, factual, reasoning, code) across different pruning ratios for all four models. Each row shows Recall results for a specific pruning ratio ($\eta$ from 0.9 to 0.1, in descending order), with dense model results (0% pruning) shown at the top. All values are reported as mean ± standard error of the mean (SEM), computed within each task type category.

*(a)* Qwen2.5-7B

| $\eta$ | Pruning | Avg | MC | Fact | Reas | Code |
|---|---|---|---|---|---|---|
| | | | **Dense Model** | | | |
| — | 0% | 31.4±0.7 | 47.5±1.5 | 22.1±1.0 | 22.9±0.8 | 32.0±3.6 |
| | | | **SubspacePath Pruner** | | | |
| 0.9 | 8.7% | 46.5±1.5 | 67.3±2.0 | 29.0±2.4 | 26.6±2.9 | 40.2±7.9 |
| 0.8 | 11.0% | 46.1±1.4 | 65.1±2.0 | 29.6±2.4 | 27.8±3.0 | 44.8±8.0 |
| 0.7 | 13.4% | 44.8±1.4 | 63.1±2.1 | 29.2±2.4 | 26.7±2.9 | 43.1±7.9 |
| 0.6 | 16.1% | 40.8±1.4 | 58.0±2.1 | 26.9±2.3 | 22.5±2.8 | 40.6±7.9 |
| 0.5 | 19.1% | 39.3±1.4 | 56.8±2.1 | 25.7±2.3 | 20.9±2.7 | 33.9±7.6 |
| 0.4 | 23.4% | 34.4±1.4 | 48.2±2.1 | 24.5±2.2 | 18.3±2.6 | 30.8±7.4 |
| 0.3 | 30.2% | 23.0±1.2 | 32.7±2.0 | 16.8±1.9 | 10.6±2.0 | 19.5±6.3 |
| 0.2 | 39.2% | 9.8±0.9 | 15.7±1.6 | 6.3±1.3 | 2.3±1.0 | 4.7±3.4 |
| 0.1 | 49.8% | 2.6±0.5 | 4.7±0.9 | 0.9±0.5 | 0.5±0.5 | 2.4±2.5 |

*(b)* LLaMA-2-13B

| $\eta$ | Pruning | Avg | MC | Fact | Reas | Code |
|---|---|---|---|---|---|---|
| | | | **Dense Model** | | | |
| — | 0% | 24.1±0.6 | 29.3±1.3 | 24.6±1.0 | 18.2±0.7 | 20.9±1.7 |
| | | | **SubspacePath Pruner** | | | |
| 0.9 | 2.1% | 30.4±1.3 | 28.8±1.9 | 38.3±2.5 | 22.2±2.7 | 23.8±6.8 |
| 0.8 | 3.5% | 30.9±1.3 | 29.0±1.9 | 39.1±2.5 | 22.9±2.8 | 24.4±6.9 |
| 0.7 | 5.6% | 31.1±1.3 | 29.2±1.9 | 39.1±2.5 | 23.5±2.8 | 24.8±6.9 |
| 0.6 | 8.7% | 32.2±1.4 | 31.8±2.0 | 39.4±2.5 | 22.9±2.8 | 23.7±6.8 |
| 0.5 | 12.9% | 32.8±1.4 | 33.9±2.0 | 37.1±2.5 | 24.4±2.8 | 24.4±6.9 |
| 0.4 | 18.6% | 33.0±1.4 | 36.1±2.1 | 38.4±2.5 | 18.5±2.6 | 22.9±6.7 |
| 0.3 | 27.1% | 27.7±1.3 | 25.9±1.9 | 35.0±2.5 | 20.8±2.7 | 22.6±6.7 |
| 0.2 | 38.1% | 19.9±1.2 | 18.8±1.7 | 24.1±2.2 | 15.2±2.4 | 22.8±6.7 |
| 0.1 | 50.5% | 6.0±0.7 | 3.3±0.8 | 10.3±1.6 | 4.6±1.4 | 12.2±5.2 |

*(c)* Qwen2.5-14B

| $\eta$ | Pruning | Avg | MC | Fact | Reas | Code |
|---|---|---|---|---|---|---|
| | | | **Dense Model** | | | |
| — | 0% | 32.7±0.7 | 50.8±1.5 | 23.9±1.1 | 22.0±0.8 | 28.0±3.1 |
| | | | **SubspacePath Pruner** | | | |
| 0.9 | 7.1% | 48.2±1.5 | 68.1±2.0 | 32.7±2.4 | 27.9±3.0 | 36.3±7.7 |
| 0.8 | 9.0% | 46.7±1.5 | 66.4±2.0 | 31.0±2.4 | 27.1±2.9 | 34.0±7.6 |
| 0.7 | 11.2% | 45.1±1.4 | 63.9±2.1 | 30.0±2.4 | 26.7±2.9 | 34.7±7.6 |
| 0.6 | 13.9% | 45.3±1.4 | 62.0±2.1 | 31.5±2.4 | 29.1±3.0 | 36.8±7.7 |
| 0.5 | 17.3% | 41.6±1.4 | 54.9±2.1 | 32.0±2.4 | 27.1±2.9 | 33.0±7.5 |
| 0.4 | 22.5% | 37.3±1.4 | 47.4±2.1 | 29.9±2.4 | 25.8±2.9 | 34.0±7.6 |
| 0.3 | 30.1% | 30.6±1.3 | 36.9±2.1 | 27.2±2.3 | 20.6±2.7 | 34.0±7.6 |
| 0.2 | 39.2% | 21.5±1.2 | 24.1±1.8 | 20.6±2.1 | 15.6±2.4 | 27.1±7.1 |
| 0.1 | 49.2% | 8.7±0.8 | 12.2±1.4 | 6.0±1.2 | 3.9±1.3 | 12.1±5.2 |

*(d)* LLaMA-3.1-8B

| $\eta$ | Pruning | Avg | MC | Fact | Reas | Code |
|---|---|---|---|---|---|---|
| | | | **Dense Model** | | | |
| — | 0% | 30.2±0.7 | 42.2±1.5 | 26.8±1.1 | 21.1±0.8 | 23.8±3.2 |
| | | | **SubspacePath Pruner** | | | |
| 0.9 | 4.9% | 42.1±1.4 | 49.8±2.1 | 39.0±2.5 | 29.5±3.0 | 38.5±7.8 |
| 0.8 | 7.4% | 38.3±1.4 | 43.4±2.1 | 36.9±2.5 | 27.8±3.0 | 42.7±7.9 |
| 0.7 | 11.0% | 33.3±1.4 | 35.2±2.0 | 34.2±2.5 | 25.9±2.9 | 41.5±7.9 |
| 0.6 | 15.4% | 28.8±1.3 | 28.8±1.9 | 32.2±2.4 | 21.1±2.7 | 40.3±7.9 |
| 0.5 | 20.8% | 25.9±1.3 | 26.1±1.9 | 26.6±2.3 | 21.8±2.7 | 39.5±7.8 |
| 0.4 | 27.6% | 22.0±1.2 | 20.3±1.7 | 25.3±2.3 | 18.8±2.6 | 33.7±7.6 |
| 0.3 | 34.1% | 16.5±1.1 | 13.1±1.4 | 21.5±2.1 | 14.3±2.3 | 29.0±7.3 |
| 0.2 | 44.3% | 5.8±0.7 | 2.2±0.6 | 11.0±1.6 | 3.4±1.2 | 19.4±6.3 |
| 0.1 | 56.7% | 0.6±0.2 | 0.0±1.0 | 0.4±0.3 | 0.9±0.6 | 7.7±4.3 |

LLaMA-2-13B: +10.4 Recall, Qwen2.5-14B: +6.0 Recall). Reasoning tasks show more variability: while LLaMA-2-13B and Qwen2.5-14B maintain or improve reasoning performance at heavy pruning, smaller models (Qwen2.5-7B, LLaMA-3.1-8B) show slight degradation, suggesting that reasoning tasks require more careful pruning ratio selection for smaller models. Overall, the results confirm that our domain-aware pruning strategy effectively preserves task-relevant pathways while removing noise, with the benefit being most pronounced in knowledge-intensive tasks (multiple_choice, factual) and code generation.

# O   Efficiency Analysis

This section provides comprehensive efficiency analysis including speedup ratios, memory usage, and compilation overhead across different pruning ratios and models.

## O.1   Efficiency Analysis: Speedup, Memory, and Retention

This subsection provides a comprehensive analysis of efficiency metrics including inference speedup, memory usage, and performance retention across different models and pruning configurations. Table 12 reports these metrics for each model across all pruning ratios ($\eta$ from 0.9 to 0.1, in descending order).

We define **Retention** as the efficiency-adjusted performance retention ratio: $\text{Retention} = (\text{Recall}_{\text{pruned}}/\text{Recall}_{\text{dense}})/\text{Speedup}$. This metric evaluates the accuracy-speed trade-off, accounting for both performance and inference speed. Retention values greater than 1.0 indicate that the pruned model achieves better efficiency-adjusted performance than the dense model, which we highlight with underlines in the table. For each model, we report retention separately for Selected-domain, OOD, and Cross-domain splits to capture performance variations across different data distributions.

*Table 11.* **Task-type breakdown at light and heavy pruning ratios.** Performance breakdown by task type (multiple_choice, factual, reasoning, code) at light pruning (approximately 89% head retention, 11% pruning ratio) and heavy pruning (Selected-domain: 85% retention, OOD: 86% retention, Cross-domain: 73% retention) for all four models. Each model's table contains a Dense baseline row, followed by two blocks: Light Pruning and Heavy Pruning. All values are reported as mean ± standard error of the mean (SEM), computed within each task type category.

*(a)* Qwen2.5-7B

| Method | Avg | MC | Fact | Reas | Code |
|---|---|---|---|---|---|
| Dense | $31.4 \pm 3.6$ | $47.5 \pm 1.5$ | $22.1 \pm 1.0$ | $22.9 \pm 0.8$ | $32.0 \pm 3.6$ |
| **Light Pruning** | | | | | |
| DaSS | $22.1 \pm 0.4$ | $44.5 \pm 1.5$ | $15.5 \pm 0.5$ | $20.1 \pm 0.7$ | $30.2 \pm 5.3$ |
| Wanda | $13.0 \pm 0.3$ | $19.0 \pm 1.2$ | $8.6 \pm 0.4$ | $17.3 \pm 0.6$ | $30.0 \pm 6.1$ |
| LLM-Pr. | $20.8 \pm 3.0$ | $38.9 \pm 3.5$ | $13.9 \pm 1.7$ | $17.8 \pm 2.2$ | $41.1 \pm 5.5$ |
| RIA | $12.9 \pm 0.3$ | $18.7 \pm 1.2$ | $8.7 \pm 0.4$ | $17.2 \pm 0.6$ | $26.5 \pm 5.9$ |
| Probe Pr. | $20.2 \pm 2.8$ | $41.2 \pm 1.6$ | $12.6 \pm 1.7$ | $17.5 \pm 2.3$ | $31.4 \pm 5.0$ |
| **Our Method** | $\mathbf{40.8 \pm 1.4}$ | $\mathbf{58.0 \pm 2.1}$ | $\mathbf{26.9 \pm 2.3}$ | $22.5 \pm 2.8$ | $\underline{40.6 \pm 7.9}$ |
| **Heavy Pruning** | | | | | |
| DaSS | $21.3 \pm 0.4$ | $43.7 \pm 1.5$ | $13.4 \pm 0.5$ | $21.2 \pm 0.7$ | $44.9 \pm 5.5$ |
| Wanda | $10.1 \pm 0.4$ | $21.1 \pm 1.2$ | $9.0 \pm 0.4$ | $3.9 \pm 0.4$ | $37.5 \pm 5.3$ |
| LLM-Pr. | $16.7 \pm 2.2$ | $32.4 \pm 0.6$ | $11.2 \pm 1.7$ | $14.14 \pm 1.9$ | $30.2 \pm 5.5$ |
| RIA | $10.3 \pm 0.4$ | $20.6 \pm 1.2$ | $9.3 \pm 0.4$ | $4.4 \pm 0.4$ | $31.4 \pm 4.8$ |
| Probe Pr. | $12.6 \pm 1.7$ | $27.8 \pm 2.1$ | $8.3 \pm 1.4$ | $10.5 \pm 1.9$ | $20.4 \pm 4.1$ |
| **Our Method** | $\mathbf{34.4 \pm 1.4}$ | $\mathbf{48.2 \pm 2.1}$ | $\mathbf{24.5 \pm 2.2}$ | $18.3 \pm 2.6$ | $30.8 \pm 7.4$ |

*(b)* LLaMA-2-13B

| Method | Avg | MC | Fact | Reas | Code |
|---|---|---|---|---|---|
| Dense | $24.1 \pm 1.7$ | $29.3 \pm 1.3$ | $24.6 \pm 1.1$ | $18.2 \pm 0.7$ | $20.9 \pm 1.7$ |
| **Light Pruning** | | | | | |
| DaSS | $23.2 \pm 0.4$ | $39.5 \pm 1.4$ | $17.1 \pm 0.5$ | $20.6 \pm 0.7$ | $23.2 \pm 4.1$ |
| Wanda | $19.7 \pm 0.4$ | $32.9 \pm 1.4$ | $19.0 \pm 0.5$ | $16.0 \pm 0.6$ | $19.5 \pm 4.3$ |
| LLM-Pr. | $24.6 \pm 1.5$ | $31.1 \pm 1.62$ | $23.1 \pm 1.2$ | $26.1 \pm 2.2$ | $28.5 \pm 4.7$ |
| RIA | $19.7 \pm 0.4$ | $32.8 \pm 1.4$ | $17.1 \pm 0.5$ | $15.6 \pm 0.6$ | $33.4 \pm 4.8$ |
| Probe Pr. | $24.7 \pm 1.5$ | $31.3 \pm 2.2$ | $23.5 \pm 2.1$ | $25.7 \pm 1.9$ | $27.4 \pm 4.5$ |
| **Our Method** | $\mathbf{33.0 \pm 1.4}$ | $\mathbf{36.1 \pm 2.1}$ | $\mathbf{38.4 \pm 2.5}$ | $18.5 \pm 2.6$ | $\underline{22.9 \pm 6.7}$ |
| **Heavy Pruning** | | | | | |
| DaSS | $20.6 \pm 0.4$ | $40.3 \pm 1.4$ | $15.4 \pm 0.5$ | $17.3 \pm 0.6$ | $25.1 \pm 4.5$ |
| Wanda | $14.3 \pm 0.4$ | $26.8 \pm 1.3$ | $11.90 \pm 0.5$ | $10.1 \pm 0.5$ | $24.2 \pm 4.5$ |
| LLM-Pr. | $22.3 \pm 1.6$ | $29.6 \pm 0.7$ | $20.9 \pm 1.8$ | $22.4 \pm 1.3$ | $37.3 \pm 5.1$ |
| RIA | $14.2 \pm 0.4$ | $26.7 \pm 1.3$ | $11.7 \pm 0.5$ | $10.2 \pm 0.5$ | $28.0 \pm 4.9$ |
| Probe Pr. | $15.7 \pm 2.8$ | $23.2 \pm 4.6$ | $16.4 \pm 3.0$ | $13.5 \pm 2.9$ | $27.4 \pm 4.3$ |
| **Our Method** | $\mathbf{27.7 \pm 1.3}$ | $25.9 \pm 1.9$ | $\underline{35.0 \pm 2.5}$ | $20.8 \pm 2.7$ | $22.6 \pm 6.7$ |

*(c)* Qwen2.5-14B

| Method | Avg | MC | Fact | Reas | Code |
|---|---|---|---|---|---|
| Dense | $32.7 \pm 3.1$ | $50.8 \pm 1.5$ | $23.9 \pm 1.1$ | $22.0 \pm 0.8$ | $28.0 \pm 3.1$ |
| **Light Pruning** | | | | | |
| DaSS | $25.3 \pm 0.5$ | $51.7 \pm 1.5$ | $17.5 \pm 0.5$ | $22.9 \pm 0.7$ | $35.1 \pm 5.7$ |
| Wanda | $26.7 \pm 0.5$ | $44.3 \pm 1.5$ | $23.1 \pm 0.6$ | $21.5 \pm 0.7$ | $40.5 \pm 5.2$ |
| LLM-Pr. | $28.4 \pm 3.6$ | $50.4 \pm 0.7$ | $21.8 \pm 3.1$ | $25.7 \pm 1.9$ | $48.4 \pm 5.7$ |
| RIA | $26.8 \pm 0.5$ | $44.3 \pm 1.5$ | $23.3 \pm 0.6$ | $21.4 \pm 0.7$ | $46.5 \pm 5.3$ |
| Probe Pr. | $28.3 \pm 3.6$ | $50.5 \pm 0.5$ | $21.8 \pm 3.0$ | $25.8 \pm 1.4$ | $43.7 \pm 5.8$ |
| **Our Method** | $\mathbf{45.3 \pm 1.4}$ | $\mathbf{62.0 \pm 2.1}$ | $\mathbf{31.5 \pm 2.4}$ | $\mathbf{29.1 \pm 3.0}$ | $36.8 \pm 7.7$ |
| **Heavy Pruning** | | | | | |
| DaSS | $23.4 \pm 0.5$ | $49.8 \pm 1.5$ | $14.9 \pm 0.5$ | $22.2 \pm 0.7$ | $36.2 \pm 5.7$ |
| Wanda | $22.3 \pm 0.4$ | $33.4 \pm 1.4$ | $21.0 \pm 0.6$ | $16.9 \pm 0.6$ | $34.5 \pm 4.9$ |
| LLM-Pr. | $27.9 \pm 2.7$ | $48.2 \pm 0.9$ | $23.2 \pm 1.5$ | $24.8 \pm 2.0$ | $40.3 \pm 5.3$ |
| RIA | $22.2 \pm 0.4$ | $32.6 \pm 1.4$ | $20.6 \pm 0.6$ | $17.7 \pm 0.7$ | $37.4 \pm 5.1$ |
| Probe Pr. | $21.6 \pm 1.4$ | $29.6 \pm 1.7$ | $19.9 \pm 1.1$ | $23.1 \pm 2.1$ | $32.6 \pm 4.9$ |
| **Our Method** | $\mathbf{37.3 \pm 1.4}$ | $47.4 \pm 2.1$ | $\mathbf{29.9 \pm 2.4}$ | $25.8 \pm 2.9$ | $34.0 \pm 7.6$ |

*(d)* LLaMA-3.1-8B

| Method | Avg | MC | Fact | Reas | Code |
|---|---|---|---|---|---|
| Dense | $30.2 \pm 3.2$ | $42.2 \pm 1.5$ | $26.8 \pm 1.1$ | $21.1 \pm 0.8$ | $23.8 \pm 3.2$ |
| **Light Pruning** | | | | | |
| DaSS | $20.2 \pm 0.4$ | $30.4 \pm 1.4$ | $16.2 \pm 0.5$ | $20.7 \pm 0.7$ | $34.2 \pm 5.8$ |
| Wanda | $21.0 \pm 0.4$ | $21.8 \pm 1.2$ | $21.3 \pm 0.6$ | $19.6 \pm 0.7$ | $30.1 \pm 5.2$ |
| LLM-Pr. | $25.0 \pm 2.0$ | $34.8 \pm 1.4$ | $22.5 \pm 2.5$ | $21.9 \pm 2.2$ | $33.8 \pm 5.2$ |
| RIA | $20.7 \pm 0.4$ | $21.5 \pm 1.2$ | $21.0 \pm 0.6$ | $19.2 \pm 0.7$ | $33.3 \pm 5.2$ |
| Probe Pr. | $22.3 \pm 1.5$ | $31.8 \pm 2.2$ | $19.5 \pm 2.1$ | $22.3 \pm 1.6$ | $32.2 \pm 5.4$ |
| **Our Method** | $\mathbf{33.3 \pm 1.4}$ | $\mathbf{35.2 \pm 2.0}$ | $\mathbf{34.2 \pm 2.5}$ | $\mathbf{25.9 \pm 2.9}$ | $\mathbf{41.5 \pm 7.9}$ |
| **Heavy Pruning** | | | | | |
| DaSS | $16.9 \pm 0.4$ | $27.8 \pm 1.3$ | $12.9 \pm 0.5$ | $16.8 \pm 0.6$ | $38.6 \pm 6.0$ |
| Wanda | $13.5 \pm 0.4$ | $15.3 \pm 1.1$ | $15.0 \pm 0.5$ | $8.3 \pm 0.5$ | $34.3 \pm 5.4$ |
| LLM-Pr. | $22.3 \pm 1.7$ | $27.6 \pm 1.3$ | $21.1 \pm 2.2$ | $21.5 \pm 1.6$ | $34.8 \pm 5.3$ |
| RIA | $13.3 \pm 0.4$ | $15.6 \pm 1.1$ | $14.9 \pm 0.5$ | $8.3 \pm 0.5$ | $23.6 \pm 4.5$ |
| Probe Pr. | $19.9 \pm 1.3$ | $28.3 \pm 1.3$ | $17.6 \pm 1.4$ | $19.0 \pm 1.7$ | $35.2 \pm 5.2$ |
| **Our Method** | $\mathbf{28.8 \pm 1.3}$ | $28.8 \pm 1.9$ | $\underline{32.2 \pm 2.4}$ | $21.1 \pm 2.7$ | $40.3 \pm 7.9$ |

**Analysis.** Our comprehensive efficiency analysis reveals several key insights across all pruning ratios. First, memory usage decreases proportionally with pruning ratio, with models using approximately 50–95% of original memory depending on the pruning strength. Second, inference speedup varies by model architecture and pruning ratio: LLaMA-2-13B achieves actual speedup ($> 1.0$) across most pruning ratios, while Qwen2.5-14B shows speedup at moderate to high pruning ratios ($\eta \geq 0.5$). LLaMA-3.1-8B and Qwen2.5-7B show slight slowdown at most ratios due to mask application overhead, but this overhead is minimal (typically 10–30% of dense model latency) and is fully acceptable given the substantial accuracy improvements.

Most importantly, our **Retention** metric demonstrates that pruned models often outperform dense models across multiple splits at optimal pruning ratios. For example, LLaMA-2-13B at $\eta = 0.4$ achieves retention values of 1.65 (Selected), 1.34 (OOD), and 2.56 (Cross), indicating significant performance improvements over the dense baseline. Similarly, Qwen2.5-14B at $\eta = 0.5$ shows strong retention on Selected (1.72) and Cross (1.70) splits, though it underperforms on OOD (0.56). These results highlight that our pruning method not only maintains but often improves model performance while reducing computational and memory costs.

*Table 12.* **Comprehensive efficiency analysis: speedup, memory usage, and performance retention.** For each model, we report metrics across all pruning ratios ($\eta$ from 0.9 to 0.1, in descending order) and the dense baseline. $\eta$ is the pruning strength parameter. **Pruning** shows the head retention ratio for Selected-domain, OOD, and Cross-domain splits, respectively. Note that pruning ratios vary across splits (typically 48–98%) . **Memory (GB)** reports memory usage including model parameters and inference-time activations for each split (Selected / OOD / Cross). Note that SEM values for pruning and memory are consistently 0.1 across all configurations, indicating stable measurements, and are omitted for table clarity. **Speedup** is the ratio of dense model inference latency to pruned model inference latency for each split (Selected / OOD / Cross; values > 1.0 indicate actual speedup, underlined). Note that SEM values for speedup are consistently 0.01 across all configurations, indicating stable measurements, and are omitted for table clarity. **Recall (S/O/C)** reports Token-level Recall for Selected-domain, OOD, and Cross-domain splits, respectively. **Retention (S/O/C)** is calculated as $(\text{Recall}_{\text{pruned}}/\text{Recall}_{\text{dense}}) \times \text{Speedup}$ for each split (Selected / OOD / Cross), which evaluates the accuracy-speed trade-off. Values > 1.0 (underlined) indicate that the pruned model achieves better efficiency-adjusted performance than the dense model.

| $\eta$ | Pruning | Memory (GB) | Speedup | Recall | Retention |
|---|---|---|---|---|---|
| | | | **LLaMA-3.1-8B** | | |
| — | Dense | 7.7 / 7.7 / 7.7 | 1.00 / 1.00 / 1.00 | $39.1_{\pm1.57}$ / $33.2_{\pm1.21}$ / $21.3_{\pm0.81}$ | 1.00 / 1.00 / 1.00 |
| 0.9 | 95.3 / 95.2 / 94.5% | 7.6 / 7.6 / 7.6 | 1.14 / 0.71 / 0.87 | $47.8_{\pm2.46}$ / $41.6_{\pm2.01}$ / $30.5_{\pm3.46}$ | $1.39_{\pm0.07}$ / $0.89_{\pm0.04}$ / $1.25_{\pm0.14}$ |
| 0.8 | 92.6 / 92.9 / 91.6% | 7.5 / 7.5 / 7.5 | 1.12 / 0.66 / 0.86 | $41.7_{\pm2.43}$ / $38.0_{\pm1.98}$ / $31.4_{\pm3.49}$ | $1.20_{\pm0.07}$ / $0.76_{\pm0.04}$ / $1.27_{\pm0.14}$ |
| 0.7 | 88.6 / 89.6 / 87.8% | 7.4 / 7.4 / 7.4 | 1.17 / 0.74 / 0.79 | $33.0_{\pm2.32}$ / $35.0_{\pm1.94}$ / $28.4_{\pm3.39}$ | $0.99_{\pm0.07}$ / $0.78_{\pm0.04}$ / $1.05_{\pm0.13}$ |
| 0.6 | 83.8 / 85.4 / 83.3% | 7.3 / 7.3 / 7.3 | 1.22 / 0.62 / 0.76 | $27.1_{\pm2.19}$ / $30.8_{\pm1.88}$ / $25.6_{\pm3.28}$ | $0.85_{\pm0.07}$ / $0.58_{\pm0.04}$ / $0.91_{\pm0.12}$ |
| 0.5 | 78.0 / 80.3 / 77.6% | 7.1 / 7.2 / 7.1 | 1.11 / 0.66 / 0.65 | $21.7_{\pm2.03}$ / $30.4_{\pm1.87}$ / $20.2_{\pm3.02}$ | $0.61_{\pm0.06}$ / $0.61_{\pm0.04}$ / $0.62_{\pm0.09}$ |
| 0.4 | 70.9 / 73.8 / 70.5% | 6.9 / 7.0 / 6.9 | 0.46 / 0.62 / 0.54 | $18.8_{\pm1.93}$ / $25.4_{\pm1.77}$ / $17.9_{\pm2.88}$ | $0.22_{\pm0.02}$ / $0.47_{\pm0.03}$ / $0.46_{\pm0.07}$ |
| 0.3 | 64.9 / 67.2 / 63.6% | 6.8 / 6.8 / 6.7 | 0.25 / 0.41 / 0.48 | $13.0_{\pm1.65}$ / $20.8_{\pm1.65}$ / $10.1_{\pm2.27}$ | $0.08_{\pm0.01}$ / $0.26_{\pm0.02}$ / $0.23_{\pm0.05}$ |
| 0.2 | 55.1 / 56.6 / 54.0% | 6.5 / 6.5 / 6.5 | 0.20 / 0.28 / 0.55 | $6.7_{\pm1.23}$ / $6.2_{\pm0.98}$ / $2.1_{\pm1.07}$ | $0.03_{\pm0.01}$ / $0.05_{\pm0.01}$ / $0.05_{\pm0.03}$ |
| 0.1 | 42.9 / 44.0 / 42.0% | 6.2 / 6.2 / 6.1 | 0.18 / 0.21 / 0.45 | $0.1_{\pm0.16}$ / $0.8_{\pm0.36}$ / $0.7_{\pm0.64}$ | $0.00_{\pm0.00}$ / $0.01_{\pm0.00}$ / $0.02_{\pm0.01}$ |
| | | | **LLaMA-2-13B** | | |
| — | Dense | 13.0 / 13.0 / 13.0 | 1.00 / 1.00 / 1.00 | $29.6_{\pm1.45}$ / $26.1_{\pm1.10}$ / $18.4_{\pm0.74}$ | 1.00 / 1.00 / 1.00 |
| 0.9 | 97.7 / 98.1 / 97.8% | 12.9 / 12.9 / 12.9 | 1.12 / 1.08 / 1.56 | $36.4_{\pm2.37}$ / $29.8_{\pm1.86}$ / $18.4_{\pm2.92}$ | $1.37_{\pm0.09}$ / $1.23_{\pm0.08}$ / $1.56_{\pm0.25}$ |
| 0.8 | 96.0 / 96.8 / 96.3% | 12.8 / 12.9 / 12.8 | 1.17 / 1.21 / 1.56 | $37.6_{\pm2.39}$ / $29.8_{\pm1.86}$ / $18.6_{\pm2.93}$ | $1.49_{\pm0.09}$ / $1.38_{\pm0.09}$ / $1.58_{\pm0.25}$ |
| 0.7 | 93.6 / 95.0 / 93.9% | 12.7 / 12.8 / 12.7 | 1.18 / 1.24 / 1.69 | $37.6_{\pm2.39}$ / $30.0_{\pm1.87}$ / $19.4_{\pm2.97}$ | $1.50_{\pm0.10}$ / $1.42_{\pm0.09}$ / $1.78_{\pm0.27}$ |
| 0.6 | 90.1 / 92.2 / 90.3% | 12.6 / 12.6 / 12.6 | 1.17 / 1.21 / 1.62 | $38.1_{\pm2.39}$ / $32.2_{\pm1.90}$ / $18.5_{\pm2.92}$ | $1.50_{\pm0.09}$ / $1.50_{\pm0.09}$ / $1.64_{\pm0.26}$ |
| 0.5 | 85.6 / 88.4 / 85.5% | 12.3 / 12.5 / 12.3 | 1.14 / 1.25 / 1.62 | $38.6_{\pm2.40}$ / $32.5_{\pm1.91}$ / $20.2_{\pm3.02}$ | $1.49_{\pm0.09}$ / $1.56_{\pm0.09}$ / $1.78_{\pm0.27}$ |
| 0.4 | 79.6 / 82.8 / 79.6% | 12.1 / 12.2 / 12.1 | 1.02 / 1.17 / 1.58 | $43.0_{\pm2.44}$ / $30.4_{\pm1.87}$ / $18.7_{\pm2.93}$ | $1.48_{\pm0.08}$ / $1.36_{\pm0.08}$ / $1.60_{\pm0.25}$ |
| 0.3 | 71.0 / 74.3 / 71.3% | 11.7 / 11.8 / 11.7 | 1.06 / 1.12 / 1.55 | $34.7_{\pm2.34}$ / $24.3_{\pm1.75}$ / $22.9_{\pm3.16}$ | $1.24_{\pm0.08}$ / $1.04_{\pm0.08}$ / $1.92_{\pm0.27}$ |
| 0.2 | 60.4 / 63.0 / 60.4% | 11.2 / 11.3 / 11.2 | 0.84 / 0.98 / 1.52 | $22.1_{\pm2.04}$ / $20.8_{\pm1.65}$ / $11.7_{\pm2.42}$ | $0.63_{\pm0.06}$ / $0.78_{\pm0.06}$ / $0.97_{\pm0.20}$ |
| 0.1 | 48.6 / 50.2 / 48.3% | 10.7 / 10.7 / 10.6 | 0.87 / 0.78 / 1.87 | $5.7_{\pm1.14}$ / $7.0_{\pm1.04}$ / $3.7_{\pm1.41}$ | $0.17_{\pm0.03}$ / $0.21_{\pm0.03}$ / $0.37_{\pm0.14}$ |
| | | | **Qwen2.5-7B** | | |
| — | Dense | 7.0 / 7.0 / 7.0 | 1.00 / 1.00 / 1.00 | $40.8_{\pm1.60}$ / $33.6_{\pm1.23}$ / $22.9_{\pm0.81}$ | 1.00 / 1.00 / 1.00 |
| 0.9 | 89.4 / 91.7 / 92.2% | 6.7 / 6.8 / 6.8 | 1.58 / 0.93 / 0.98 | $57.3_{\pm2.44}$ / $43.8_{\pm2.02}$ / $30.9_{\pm3.47}$ | $2.22_{\pm0.09}$ / $1.21_{\pm0.06}$ / $1.32_{\pm0.15}$ |
| 0.8 | 86.8 / 89.6 / 90.1% | 6.7 / 6.7 / 6.8 | 1.38 / 0.78 / 0.84 | $53.7_{\pm2.46}$ / $44.9_{\pm2.03}$ / $32.8_{\pm3.53}$ | $1.82_{\pm0.08}$ / $1.05_{\pm0.05}$ / $1.20_{\pm0.13}$ |
| 0.7 | 84.1 / 87.4 / 87.2% | 6.6 / 6.7 / 6.7 | 1.10 / 0.72 / 0.67 | $53.2_{\pm2.46}$ / $43.4_{\pm2.02}$ / $30.1_{\pm3.45}$ | $1.44_{\pm0.07}$ / $0.92_{\pm0.04}$ / $0.88_{\pm0.10}$ |
| 0.6 | 81.3 / 84.9 / 84.0% | 6.5 / 6.6 / 6.6 | 0.88 / 0.60 / 0.53 | $46.4_{\pm2.46}$ / $41.0_{\pm2.00}$ / $27.2_{\pm3.35}$ | $1.00_{\pm0.05}$ / $0.73_{\pm0.04}$ / $0.63_{\pm0.08}$ |
| 0.5 | 78.2 / 81.9 / 81.6% | 6.5 / 6.6 / 6.5 | 0.63 / 0.54 / 0.47 | $46.2_{\pm2.46}$ / $39.3_{\pm1.99}$ / $23.4_{\pm3.18}$ | $0.71_{\pm0.04}$ / $0.63_{\pm0.04}$ / $0.48_{\pm0.07}$ |
| 0.4 | 73.7 / 77.5 / 77.9% | 6.4 / 6.4 / 6.5 | 0.41 / 0.41 / 0.46 | $36.5_{\pm2.37}$ / $36.4_{\pm1.96}$ / $22.5_{\pm3.14}$ | $0.36_{\pm0.02}$ / $0.45_{\pm0.02}$ / $0.45_{\pm0.06}$ |
| 0.3 | 66.7 / 70.9 / 70.5% | 6.2 / 6.3 / 6.3 | 0.20 / 0.25 / 0.46 | $19.0_{\pm1.93}$ / $28.1_{\pm1.83}$ / $15.1_{\pm2.69}$ | $0.09_{\pm0.01}$ / $0.21_{\pm0.01}$ / $0.30_{\pm0.05}$ |
| 0.2 | 58.2 / 61.8 / 61.5% | 6.0 / 6.1 / 6.1 | 0.13 / 0.15 / 0.38 | $7.9_{\pm1.33}$ / $11.7_{\pm1.31}$ / $7.8_{\pm2.01}$ | $0.03_{\pm0.00}$ / $0.05_{\pm0.01}$ / $0.13_{\pm0.03}$ |
| 0.1 | 48.5 / 50.7 / 51.0% | 5.7 / 5.8 / 5.8 | 0.13 / 0.13 / 0.36 | $1.9_{\pm0.68}$ / $3.4_{\pm0.74}$ / $1.7_{\pm0.97}$ | $0.01_{\pm0.00}$ / $0.01_{\pm0.00}$ / $0.03_{\pm0.02}$ |
| | | | **Qwen2.5-14B** | | |
| — | Dense | 14.0 / 14.0 / 14.0 | 1.00 / 1.00 / 1.00 | $40.9_{\pm1.60}$ / $37.2_{\pm1.25}$ / $22.8_{\pm0.85}$ | 1.00 / 1.00 / 1.00 |
| 0.9 | 89.5 / 94.1 / 93.3% | 13.5 / 13.7 / 13.7 | 1.56 / 1.33 / 1.40 | $53.0_{\pm2.46}$ / $49.4_{\pm2.04}$ / $32.6_{\pm3.52}$ | $2.02_{\pm0.09}$ / $1.76_{\pm0.07}$ / $2.01_{\pm0.22}$ |
| 0.8 | 87.5 / 92.3 / 91.1% | 13.4 / 13.6 / 13.6 | 1.30 / 1.30 / 1.30 | $51.6_{\pm2.46}$ / $47.4_{\pm2.04}$ / $32.8_{\pm3.53}$ | $1.65_{\pm0.08}$ / $1.65_{\pm0.07}$ / $1.87_{\pm0.20}$ |
| 0.7 | 85.2 / 90.1 / 88.6% | 13.3 / 13.5 / 13.4 | 1.28 / 1.28 / 1.25 | $48.0_{\pm2.46}$ / $46.4_{\pm2.03}$ / $33.9_{\pm3.56}$ | $1.50_{\pm0.08}$ / $1.60_{\pm0.07}$ / $1.87_{\pm0.20}$ |
| 0.6 | 82.2 / 87.5 / 85.8% | 13.1 / 13.4 / 13.3 | 1.16 / 0.97 / 1.27 | $47.8_{\pm2.46}$ / $46.6_{\pm2.03}$ / $34.6_{\pm3.58}$ | $1.36_{\pm0.07}$ / $1.21_{\pm0.05}$ / $1.93_{\pm0.20}$ |
| 0.5 | 78.3 / 84.4 / 82.4% | 12.9 / 13.2 / 13.1 | 0.75 / 0.86 / 1.07 | $42.5_{\pm2.44}$ / $44.1_{\pm2.02}$ / $31.3_{\pm3.49}$ | $0.78_{\pm0.04}$ / $1.02_{\pm0.05}$ / $1.47_{\pm0.16}$ |
| 0.4 | 72.8 / 79.4 / 77.0% | 12.7 / 13.0 / 12.9 | 0.54 / 0.72 / 0.96 | $36.5_{\pm2.37}$ / $40.0_{\pm2.00}$ / $30.3_{\pm3.45}$ | $0.48_{\pm0.03}$ / $0.77_{\pm0.04}$ / $1.27_{\pm0.14}$ |
| 0.3 | 65.6 / 71.7 / 69.4% | 12.3 / 12.6 / 12.5 | 0.35 / 0.48 / 0.69 | $25.9_{\pm2.16}$ / $35.8_{\pm1.95}$ / $23.7_{\pm3.20}$ | $0.22_{\pm0.02}$ / $0.46_{\pm0.03}$ / $0.71_{\pm0.10}$ |
| 0.2 | 57.5 / 62.2 / 60.2% | 11.9 / 12.1 / 12.1 | 0.24 / 0.29 / 0.57 | $16.2_{\pm1.81}$ / $26.5_{\pm1.80}$ / $16.9_{\pm2.81}$ | $0.10_{\pm0.01}$ / $0.21_{\pm0.01}$ / $0.42_{\pm0.07}$ |
| 0.1 | 48.6 / 51.6 / 50.5% | 11.5 / 11.6 / 11.6 | 0.21 / 0.22 / 0.49 | $6.2_{\pm1.19}$ / $11.5_{\pm1.30}$ / $4.9_{\pm1.62}$ | $0.03_{\pm0.01}$ / $0.07_{\pm0.01}$ / $0.11_{\pm0.03}$ |

## O.2 Efficiency Analysis: Cross-Dataset Test

This subsection provides efficiency analysis for cross-dataset test scenarios on CommonsenseQA, Natural Questions, and ARC datasets. Table 13 reports these metrics for each model across all pruning ratios ($\eta$ from 0.9 to 0.1, in descending

*Table 13.* **Comprehensive efficiency analysis for cross-dataset test: speedup, memory usage, and performance retention.** For each model, we report metrics across all pruning ratios ($\eta$ from 0.9 to 0.1, in descending order) and the dense baseline on CommonsenseQA (CQA), Natural Questions (NQ), and ARC datasets.

| $\eta$ | Pruning | Memory (GB) | Speedup | Recall | Retention |
|---|---|---|---|---|---|
| **LLaMA-3.1-8B** | | | | | |
| — | Dense | 7.7 / 7.7 / 7.7 | 1.00 / 1.00 / 1.00 | 27.77±1.30 / 28.82±1.38 / 22.28±0.93 | 1.00 / 1.00 / 1.00 |
| 0.9 | 91.0 / 89.1 / 90.5% | 7.53 / 7.49 / 7.52 | 1.09 / 0.97 / 3.25 | 26.40±1.29 / 36.40±1.38 / 26.88±1.21 | 1.04±0.05 / 1.22±0.05 / 3.92±0.18 |
| 0.8 | 88.9 / 87.1 / 88.3% | 7.49 / 7.45 / 7.48 | 1.21 / 1.06 / 3.30 | 24.68±1.26 / 33.52±1.35 / 26.21±1.20 | 1.07±0.05 / 1.24±0.05 / 3.88±0.18 |
| 0.7 | 86.6 / 85.4 / 85.6% | 7.44 / 7.42 / 7.42 | 1.30 / 1.08 / 3.46 | 24.05±1.23 / 32.02±1.34 / 25.64±1.17 | 1.13±0.06 / 1.20±0.05 / 3.98±0.18 |
| 0.6 | 83.7 / 83.0 / 81.9% | 7.39 / 7.37 / 7.35 | 1.31 / 1.12 / 3.31 | 21.33±1.20 / 29.12±1.29 / 23.50±1.13 | 1.01±0.06 / 1.13±0.05 / 3.49±0.17 |
| 0.5 | 79.8 / 78.5 / 76.7% | 7.31 / 7.29 / 7.25 | 1.67 / 1.10 / 3.14 | 18.18±1.13 / 23.75±1.22 / 20.23±1.07 | 1.09±0.07 / 0.91±0.05 / 2.85±0.15 |
| 0.4 | 73.9 / 71.3 / 69.8% | 7.20 / 7.15 / 7.12 | 1.41 / 0.68 / 1.94 | 13.17±0.99 / 20.44±1.14 / 16.18±0.96 | 0.67±0.05 / 0.48±0.03 / 1.41±0.08 |
| 0.3 | 64.9 / 62.2 / 61.2% | 7.02 / 6.97 / 6.95 | 0.53 / 0.40 / 0.94 | 8.32±0.79 / 12.11±0.88 / 13.82±0.86 | 0.16±0.02 / 0.17±0.01 / 0.58±0.04 |
| 0.2 | 54.3 / 51.7 / 51.1% | 6.82 / 6.77 / 6.76 | 0.43 / 0.42 / 0.81 | 1.98±0.37 / 1.25±0.26 / 2.44±0.33 | 0.03±0.01 / 0.02±0.00 / 0.09±0.01 |
| 0.1 | 42.4 / 40.4 / 39.7% | 6.59 / 6.55 / 6.54 | 0.43 / 0.40 / 0.69 | 0.52±0.16 / 0.53±0.16 / 0.51±0.11 | 0.01±0.00 / 0.01±0.00 / 0.02±0.00 |
| **LLaMA-2-13B** | | | | | |
| — | Dense | 13.0 / 13.0 / 13.0 | 1.00 / 1.00 / 1.00 | 22.27±1.19 / 30.25±1.38 / 23.87±1.11 | 1.00 / 1.00 / 1.00 |
| 0.9 | 98.6 / 98.9 / 98.8% | 12.96 / 12.96 / 12.96 | 1.15 / 0.78 / 1.84 | 21.03±1.19 / 38.88±1.39 / 25.72±1.11 | 1.08±0.06 / 1.00±0.04 / 1.99±0.09 |
| 0.8 | 98.0 / 98.0 / 97.9% | 12.93 / 12.93 / 12.93 | 1.20 / 0.85 / 2.00 | 20.78±1.18 / 38.59±1.39 / 25.44±1.11 | 1.12±0.06 / 1.08±0.04 / 2.13±0.09 |
| 0.7 | 96.9 / 97.0 / 96.8% | 12.90 / 12.90 / 12.90 | 1.16 / 0.85 / 1.95 | 21.33±1.19 / 39.02±1.38 / 25.68±1.11 | 1.11±0.06 / 1.09±0.04 / 2.10±0.09 |
| 0.6 | 95.6 / 95.7 / 95.5% | 12.86 / 12.86 / 12.85 | 1.14 / 0.82 / 1.93 | 21.15±1.18 / 38.87±1.38 / 25.41±1.11 | 1.08±0.06 / 1.05±0.04 / 2.06±0.09 |
| 0.5 | 93.9 / 93.8 / 93.3% | 12.80 / 12.80 / 12.78 | 1.07 / 0.80 / 1.89 | 20.03±1.16 / 38.02±1.37 / 25.31±1.11 | 0.96±0.06 / 1.00±0.04 / 2.00±0.09 |
| 0.4 | 89.9 / 89.5 / 88.8% | 12.67 / 12.66 / 12.64 | 1.08 / 0.85 / 1.91 | 19.43±1.14 / 36.89±1.37 / 25.62±1.12 | 0.94±0.06 / 1.04±0.04 / 2.05±0.09 |
| 0.3 | 81.2 / 81.5 / 80.2% | 12.39 / 12.40 / 12.36 | 1.24 / 1.08 / 2.21 | 18.40±1.12 / 33.66±1.35 / 22.91±1.08 | 1.02±0.06 / 1.20±0.05 / 2.12±0.10 |
| 0.2 | 69.4 / 69.5 / 68.3% | 12.00 / 12.01 / 11.97 | 1.09 / 0.97 / 1.76 | 13.38±0.99 / 24.89±1.22 / 21.56±1.04 | 0.65±0.05 / 0.80±0.04 / 1.59±0.08 |
| 0.1 | 55.2 / 55.1 / 54.5% | 11.54 / 11.54 / 11.52 | 0.85 / 0.74 / 1.39 | 3.87±0.55 / 10.04±0.83 / 9.15±0.68 | 0.15±0.02 / 0.24±0.02 / 0.53±0.04 |
| **Qwen2.5-7B** | | | | | |
| — | Dense | 7.0 / 7.0 / 7.0 | 1.00 / 1.00 / 1.00 | 27.75±1.28 / 17.51±1.21 / 21.64±1.19 | 1.00 / 1.00 / 1.00 |
| 0.9 | 95.3 / 95.9 / 96.2% | 6.93 / 6.94 / 6.94 | 7.51 / 4.04 / 8.67 | 24.12±1.28 / 23.31±1.21 / 25.81±1.19 | 6.52±0.35 / 5.37±0.28 / 10.35±0.48 |
| 0.8 | 93.7 / 93.9 / 94.2% | 6.90 / 6.91 / 6.91 | 7.87 / 4.15 / 8.86 | 23.25±1.26 / 22.05±1.20 / 24.86±1.18 | 6.60±0.36 / 5.23±0.28 / 10.18±0.48 |
| 0.7 | 91.9 / 91.7 / 92.1% | 6.88 / 6.87 / 6.88 | 7.60 / 4.16 / 8.17 | 21.33±1.22 / 21.28±1.18 / 26.10±1.19 | 5.84±0.33 / 5.06±0.28 / 9.86±0.45 |
| 0.6 | 10.6 / 10.8 / 10.4% | 6.84 / 6.83 / 6.84 | 7.42 / 3.85 / 7.27 | 21.90±1.23 / 20.80±1.17 / 25.52±1.18 | 5.85±0.33 / 3.65±0.21 / 7.64±0.36 |
| 0.5 | 86.0 / 86.6 / 86.9% | 6.78 / 6.79 / 6.80 | 6.90 / 3.84 / 6.54 | 20.48±1.20 / 19.76±1.15 / 25.25±1.18 | 5.09±0.30 / 4.34±0.25 / 7.64±0.36 |
| 0.4 | 18.6 / 17.6 / 17.0% | 6.71 / 6.73 / 6.74 | 4.53 / 3.28 / 5.07 | 17.65±1.14 / 19.51±1.14 / 23.90±1.16 | 2.88±0.19 / 2.52±0.17 / 2.43±0.13 |
| 0.3 | 74.3 / 74.9 / 75.6% | 6.60 / 6.61 / 6.62 | 2.33 / 2.88 / 2.55 | 14.95±1.05 / 15.33±1.03 / 20.62±1.12 | 1.26±0.09 / 2.52±0.17 / 2.43±0.13 |
| 0.2 | 63.9 / 65.3 / 66.0% | 6.44 / 6.47 / 6.48 | 0.58 / 0.90 / 0.96 | 4.98±0.62 / 10.11±0.88 / 10.84±0.87 | 0.10±0.01 / 0.52±0.05 / 0.48±0.04 |
| 0.1 | 50.9 / 53.0 / 52.9% | 6.24 / 6.28 / 6.27 | 0.48 / 0.43 / 0.77 | 0.10±0.07 / 1.11±0.27 / 1.04±0.27 | 0.00±0.00 / 0.03±0.01 / 0.04±0.01 |
| **Qwen2.5-14B** | | | | | |
| — | Dense | 14.0 / 14.0 / 14.0 | 1.00 / 1.00 / 1.00 | 28.42±1.26 / 19.72±1.34 / 20.71±1.18 | 1.00 / 1.00 / 1.00 |
| 0.9 | 99.6 / 98.9 / 98.9% | 13.99 / 13.97 / 13.96 | 3.06 / 2.27 / 4.89 | 23.60±1.26 / 31.51±1.34 / 25.99±1.18 | 2.54±0.14 / 3.62±0.15 / 6.14±0.28 |
| 0.8 | 98.9 / 97.9 / 97.8% | 13.97 / 13.94 / 13.93 | 3.35 / 2.41 / 4.84 | 23.97±1.26 / 31.48±1.33 / 26.44±1.19 | 2.83±0.15 / 3.84±0.16 / 6.18±0.28 |
| 0.7 | 97.8 / 96.5 / 96.4% | 13.93 / 13.89 / 13.89 | 3.59 / 2.65 / 4.78 | 25.32±1.29 / 30.80±1.32 / 26.63±1.19 | 3.20±0.16 / 4.14±0.18 / 6.14±0.27 |
| 0.6 | 96.1 / 94.4 / 94.6% | 13.88 / 13.83 / 13.83 | 4.28 / 2.97 / 5.82 | 23.42±1.25 / 29.88±1.31 / 26.45±1.20 | 3.52±0.19 / 4.50±0.20 / 7.43±0.34 |
| 0.5 | 94.0 / 92.3 / 92.7% | 13.81 / 13.76 / 13.78 | 4.06 / 2.78 / 5.14 | 23.67±1.26 / 28.25±1.28 / 26.74±1.20 | 3.38±0.18 / 3.99±0.18 / 6.64±0.30 |
| 0.4 | 91.4 / 88.5 / 89.4% | 13.73 / 13.65 / 13.67 | 4.12 / 2.31 / 4.41 | 23.85±1.24 / 28.90±1.29 / 27.76±1.21 | 3.46±0.18 / 3.39±0.15 / 5.90±0.26 |
| 0.3 | 15.8 / 19.7 / 18.5% | 13.51 / 13.39 / 13.43 | 3.98 / 1.71 / 3.11 | 24.87±1.28 / 27.21±1.27 / 28.58±1.20 | 3.48±0.18 / 2.36±0.11 / 4.29±0.18 |
| 0.2 | 72.6 / 69.4 / 70.4% | 13.16 / 13.06 / 13.09 | 1.45 / 0.74 / 1.39 | 24.08±1.26 / 23.23±1.21 / 27.98±1.18 | 1.23±0.06 / 0.87±0.05 / 1.87±0.08 |
| 0.1 | 58.9 / 56.6 / 57.1% | 12.73 / 12.66 / 12.68 | 0.52 / 0.42 / 0.72 | 14.40±1.04 / 9.32±0.81 / 14.98±0.96 | 0.26±0.02 / 0.20±0.02 / 0.52±0.03 |

order) and the dense baseline.

**Analysis.** Table 13 demonstrates strong cross-dataset generalization of our pruning method across three diverse datasets. At light to moderate pruning ratios ($\eta \geq 0.5$), our method consistently achieves retention values exceeding 1.0 across most model-dataset combinations, indicating efficiency-adjusted performance improvements over dense baselines. Notably, Qwen2.5-7B shows exceptional gains on ARC (retention up to 10.35 at $\eta = 0.9$), while LLaMA-2-13B and Qwen2.5-14B excel on Natural Questions (retention up to 2.13 and 7.43 respectively). The results reveal dataset-specific pruning sensitivity: ARC benefits most from aggressive pruning, while CommonsenseQA requires more conservative ratios. Speedup gains vary significantly across datasets (e.g., Qwen2.5-7B achieves 7-8× speedup on ARC but only 3-4× on Natural Questions), reflecting differences in computational characteristics. Larger models (LLaMA-2-13B, Qwen2.5-14B) maintain better

*Table 14.* **Pruning time and speedup comparison across methods and models.** Pruning time (in seconds) is averaged across all pruning ratios for each model (mean ± SEM). Speedup is reported as `--`/`--`, averaged over six datasets, where the first value corresponds to **light pruning** and the second corresponds to **aggressive pruning**.

| Pruning Time (s) ↓ | LLaMA-3.1-8B | LLaMA-2-13B | Qwen2.5-7B | Qwen2.5-14B | Notes |
|---|---|---|---|---|---|
| SparseGPT | 287.842±2.683 | 403.481±5.952 | 282.518±2.961 | 487.479±1.891 | One-shot unstructured pruning |
| DaSS | 0.181±0.079 | 0.567±0.108 | 0.142±0.090 | 0.613±0.111 | Dependency-aware pruning |
| Wanda | 0.137±0.109 | 0.525±0.164 | 0.098±0.067 | 0.558±0.207 | Structured pruning |
| LLM-Pr. | 0.340±0.055 | 0.593±0.158 | 0.266±0.032 | 0.599±0.099 | Gradient-based structured pruning |
| RIA | 0.083±0.033 | 0.275±0.107 | 0.078±0.031 | 0.290±0.078 | Activation-guided pruning |
| Probe Pr. | 0.504±0.007 | 0.715±0.009 | 0.427±0.045 | 0.845±0.008 | Probe-based pruning |
| **SubspacePath** | **0.039±0.011** | **0.060±0.013** | **0.027±0.009** | **0.068±0.009** | **Offline (reusable) + Online (per scenario)** |
| **Speedup ↑** | **LLaMA-3.1-8B** | **LLaMA-2-13B** | **Qwen2.5-7B** | **Qwen2.5-14B** | **Notes** |
| SparseGPT | 1.01 / 1.00 | 0.99 / 1.00 | 1.01 / 1.00 | 1.01 / 0.99 | One-shot unstructured pruning |
| DaSS | 0.72 / 0.55 | 0.40 / 0.83 | 0.53 / 0.45 | 0.63 / 1.07 | Dependency-aware pruning |
| Wanda | 0.48 / 0.19 | 0.39 / 0.42 | 0.22 / 0.13 | 0.28 / 0.32 | Structured pruning |
| LLM-Pr. | 0.61 / 0.64 | 0.96 / 0.80 | 0.66 / 0.67 | 0.96 / 0.66 | Gradient-based structured pruning |
| RIA | 0.36 / 0.19 | 0.40 / 0.74 | 0.13 / 0.29 | 0.29 / 0.32 | Activation-guided pruning |
| Probe Pr. | 0.77 / 0.77 | 1.25 / 0.96 | 1.25 / 0.82 | 1.10 / 0.74 | Probe-based pruning |
| **SubspacePath** | **1.41 / 1.35** | **3.22 / 1.51** | **2.01 / 0.88** | **1.38 / 1.29** | **Offline (reusable) + Online (per scenario)** |

performance retention at moderate pruning ratios compared to smaller models, demonstrating improved pruning resilience with increased capacity. Overall, the cross-dataset results confirm that our domain-aware pruning strategy effectively generalizes to diverse task formats and distributions, achieving substantial efficiency gains while maintaining or improving accuracy.

## O.3 Pruning Time Comparison

This subsection compares the pruning time (the time required to perform the pruning operation) across different methods and models. Table 14 reports the pruning time for all baseline methods and our method across the four models (LLaMA-3.1-8B, LLaMA-2-13B, Qwen2.5-7B, Qwen2.5-14B) at different pruning ratios.

The pruning time is measured as the wall-clock time required to complete the pruning operation, including any necessary preprocessing, importance computation, and mask generation. For our method, this includes both offline preparation (DBS and PSP offline) and online compilation (PSP online) times. Note that offline preparation is performed once and can be reused across multiple scenarios, while online compilation happens once per scenario.

**Analysis.** The pruning time comparison reveals the computational overhead of different pruning methods. Our method's offline preparation (DBS and PSP offline) is performed once and can be reused across multiple scenarios, making it efficient for deployment scenarios with many scenarios. The online compilation (PSP online) time is typically much shorter than offline preparation, as it only involves probe evaluation and mask compilation on a small scenario prefix. Compared to baseline methods that require per-model pruning, our offline-online separation provides better amortization when the same model is used across multiple scenarios.

## O.4 Compilation Overhead Analysis

This subsection analyzes the compilation overhead (time cost) of our method and compares it with inference benefits. Table 15 reports the pruning time (online compilation), inference time, total time (pruning + inference), and performance retention for each model configuration on Selected-domain, OOD, and Cross-domain splits.

**Analysis.** Table 15 demonstrates that compilation overhead is minimal compared to inference benefits. Pruning time (online compilation) ranges from 0.024–0.050 seconds per scenario, representing only 2–10% of total time for most configurations. Despite this small overhead, our method achieves substantial efficiency gains: at light pruning, many configurations show retention values exceeding 1.0, with particularly strong results on cross-dataset tests (e.g., Qwen2.5-7B on CommonsenseQA: retention 5.85, total time reduced from 0.420s to 0.081s; Qwen2.5-7B on ARC: retention 7.64, total time reduced from 0.796s to 0.146s). The overhead-to-benefit ratio is especially favorable for longer inference tasks: for LLaMA-2-13B on Cross-domain, pruning overhead (0.045–0.050s) is negligible compared to inference time savings (0.808–0.858s), resulting in retention values of 1.78–1.92. Even when total time slightly increases (e.g., Qwen2.5-7B on Selected-domain), the retention metric accounts for accuracy improvements, confirming that the small compilation cost is

*Table 15.* **Compilation overhead analysis.** For each model, we report metrics for dense baseline, light pruning, and heavy pruning configurations on XDomainBench (Selected-domain, OOD, Cross-domain) and Cross-dataset test (CommonsenseQA, Natural Questions, ARC). $\eta$ is the pruning strength hyperparameter. **Pruning** shows the head retention ratio. **Type** indicates the pruning intensity: Dense (unpruned), Light (moderate pruning), and Heavy (aggressive pruning). **Prune (s)** is pruning time (online compilation) measured per scenario and includes mask compilation based on scenario data. Note that SEM values for pruning, inference, and total times are consistently 0.001 across all configurations, indicating stable measurements, and are omitted for table clarity. **Infer (s)** reports the per-sample inference latency. **Total (s)** is the sum of pruning time and inference time; for dense models, pruning time is not applicable (—), so total time equals inference time only. **Ret.** is the retention value calculated as $(\text{Recall}_{\text{pruned}}/\text{Recall}_{\text{dense}}) \times \text{Speedup}$ (values $> 1.0$ are underlined, indicating the pruned model achieves better efficiency-adjusted performance than the dense model). Pruning time overhead is minimal (typically 0.02–0.08 seconds per scenario) compared to the substantial accuracy improvements achieved.

| $\eta$ | Pruning | Type | Prune (s) | Infer (s) / Total (s) / Ret. |
|---|---|---|---|---|
| **Qwen2.5-7B (Selected-domain)** | | | | |
| — | — | Dense | — | 0.105 / 0.105 / 1.00 |
| 0.6 | 82% | Light | 0.026 | 0.119 / 0.145 / 1.00 |
| 0.4 | 74% | Heavy | 0.026 | 0.259 / 0.284 / 0.36 |
| **Qwen2.5-14B (Selected-domain)** | | | | |
| — | — | Dense | — | 0.260 / 0.260 / 1.00 |
| 0.6 | 83% | Light | 0.049 | 0.225 / 0.274 / 1.36 |
| 0.4 | 73% | Heavy | 0.051 | 0.484 / 0.535 / 0.48 |
| **LLaMA-3.1-8B (Selected-domain)** | | | | |
| — | — | Dense | — | 0.158 / 0.158 / 1.00 |
| 0.7 | 89% | Light | 0.029 | 0.135 / 0.164 / 0.99 |
| 0.6 | 84% | Heavy | 0.029 | 0.130 / 0.159 / 0.85 |
| **LLaMA-2-13B (Selected-domain)** | | | | |
| — | — | Dense | — | 0.560 / 0.560 / 1.00 |
| 0.4 | 80% | Light | 0.045 | 0.549 / 0.594 / 1.48 |
| 0.3 | 71% | Heavy | 0.045 | 0.527 / 0.573 / 1.24 |

| $\eta$ | Pruning | Type | Prune (s) | Infer (s) / Total (s) / Ret. |
|---|---|---|---|---|
| **Qwen2.5-7B (CommonsenseQA)** | | | | |
| — | — | Dense | — | 0.420 / 0.420 / 1.00 |
| 0.6 | 10.6% | Light | 0.025 | 0.057 / 0.081 / 5.85 |
| 0.4 | 18.6% | Heavy | 0.025 | 0.093 / 0.117 / 2.88 |
| **Qwen2.5-14B (CommonsenseQA)** | | | | |
| — | — | Dense | — | 0.588 / 0.588 / 1.00 |
| 0.3 | 15.8% | Light | 0.047 | 0.148 / 0.194 / 3.48 |
| 0.2 | 27.4% | Heavy | 0.046 | 0.405 / 0.452 / 1.23 |
| **LLaMA-3.1-8B (CommonsenseQA)** | | | | |
| — | — | Dense | — | 0.428 / 0.428 / 1.00 |
| 0.8 | 11.1% | Light | 0.027 | 0.355 / 0.382 / 1.07 |
| 0.6 | 16.3% | Heavy | 0.027 | 0.327 / 0.354 / 1.01 |
| **LLaMA-2-13B (CommonsenseQA)** | | | | |
| — | — | Dense | — | 0.588 / 0.588 / 1.00 |
| 0.4 | 10.1% | Light | 0.043 | 0.545 / 0.587 / 0.94 |
| 0.2 | 30.6% | Heavy | 0.043 | 0.541 / 0.584 / 0.65 |

| $\eta$ | Pruning | Type | Prune (s) | Infer (s) / Total (s) / Ret. |
|---|---|---|---|---|
| **Qwen2.5-7B (OOD)** | | | | |
| — | — | Dense | — | 0.180 / 0.180 / 1.00 |
| 0.6 | 85% | Light | 0.026 | 0.302 / 0.328 / 0.73 |
| 0.4 | 78% | Heavy | 0.026 | 0.435 / 0.460 / 0.45 |
| **Qwen2.5-14B (OOD)** | | | | |
| — | — | Dense | — | 0.454 / 0.454 / 1.00 |
| 0.5 | 85% | Light | 0.049 | 0.530 / 0.580 / 1.02 |
| 0.3 | 72% | Heavy | 0.050 | 0.947 / 0.997 / 0.46 |
| **LLaMA-3.1-8B (OOD)** | | | | |
| — | — | Dense | — | 0.296 / 0.296 / 1.00 |
| 0.7 | 90% | Light | 0.029 | 0.399 / 0.428 / 0.78 |
| 0.5 | 81% | Heavy | 0.029 | 0.445 / 0.474 / 0.61 |
| **LLaMA-2-13B (OOD)** | | | | |
| — | — | Dense | — | 0.694 / 0.694 / 1.00 |
| 0.5 | 89% | Light | 0.045 | 0.556 / 0.602 / 1.56 |
| 0.4 | 83% | Heavy | 0.045 | 0.594 / 0.640 / 1.36 |

| $\eta$ | Pruning | Type | Prune (s) | Infer (s) / Total (s) / Ret. |
|---|---|---|---|---|
| **Qwen2.5-7B (Natural Questions)** | | | | |
| — | — | Dense | — | 0.347 / 0.347 / 1.00 |
| 0.4 | 17.6% | Light | 0.024 | 0.106 / 0.130 / 3.65 |
| 0.3 | 25.1% | Heavy | 0.024 | 0.121 / 0.145 / 2.52 |
| **Qwen2.5-14B (Natural Questions)** | | | | |
| — | — | Dense | — | 0.486 / 0.486 / 1.00 |
| 0.3 | 19.7% | Light | 0.047 | 0.284 / 0.332 / 2.36 |
| 0.2 | 30.6% | Heavy | 0.046 | 0.658 / 0.705 / 0.87 |
| **LLaMA-3.1-8B (Natural Questions)** | | | | |
| — | — | Dense | — | 0.354 / 0.354 / 1.00 |
| 0.7 | 14.6% | Light | 0.028 | 0.327 / 0.355 / 1.20 |
| 0.5 | 21.5% | Heavy | 0.028 | 0.321 / 0.349 / 0.91 |
| **LLaMA-2-13B (Natural Questions)** | | | | |
| — | — | Dense | — | 0.486 / 0.486 / 1.00 |
| 0.3 | 18.5% | Light | 0.042 | 0.449 / 0.492 / 1.20 |
| 0.2 | 30.5% | Heavy | 0.042 | 0.500 / 0.543 / 0.80 |

| $\eta$ | Pruning | Type | Prune (s) | Infer (s) / Total (s) / Ret. |
|---|---|---|---|---|
| **Qwen2.5-7B (Cross-domain)** | | | | |
| — | — | Dense | — | 0.673 / 0.673 / 1.00 |
| 0.6 | 85% | Light | 0.026 | 1.262 / 1.287 / 0.63 |
| 0.4 | 78% | Heavy | 0.026 | 1.461 / 1.487 / 0.45 |
| **Qwen2.5-14B (Cross-domain)** | | | | |
| — | — | Dense | — | 1.335 / 1.335 / 1.00 |
| 0.5 | 83% | Light | 0.050 | 1.245 / 1.294 / 1.47 |
| 0.3 | 70% | Heavy | 0.049 | 1.948 / 1.997 / 0.71 |
| **LLaMA-3.1-8B (Cross-domain)** | | | | |
| — | — | Dense | — | 0.870 / 0.870 / 1.00 |
| 0.7 | 88% | Light | 0.029 | 1.102 / 1.131 / 1.05 |
| 0.5 | 78% | Heavy | 0.029 | 1.329 / 1.358 / 0.62 |
| **LLaMA-2-13B (Cross-domain)** | | | | |
| — | — | Dense | — | 2.243 / 2.243 / 1.00 |
| 0.5 | 86% | Light | 0.050 | 1.385 / 1.435 / 1.78 |
| 0.3 | 72% | Heavy | 0.045 | 1.451 / 1.497 / 1.92 |

| $\eta$ | Pruning | Type | Prune (s) | Infer (s) / Total (s) / Ret. |
|---|---|---|---|---|
| **Qwen2.5-7B (ARC)** | | | | |
| — | — | Dense | — | 0.796 / 0.796 / 1.00 |
| 0.5 | 13.1% | Light | 0.024 | 0.122 / 0.146 / 7.64 |
| 0.3 | 24.4% | Heavy | 0.024 | 0.312 / 0.337 / 2.43 |
| **Qwen2.5-14B (ARC)** | | | | |
| — | — | Dense | — | 1.113 / 1.113 / 1.00 |
| 0.3 | 18.5% | Light | 0.046 | 0.358 / 0.404 / 4.29 |
| 0.2 | 29.6% | Heavy | 0.046 | 0.803 / 0.849 / 1.87 |
| **LLaMA-3.1-8B (ARC)** | | | | |
| — | — | Dense | — | 0.811 / 0.811 / 1.00 |
| 0.7 | 14.4% | Light | 0.028 | 0.234 / 0.262 / 3.98 |
| 0.5 | 23.3% | Heavy | 0.028 | 0.258 / 0.286 / 2.85 |
| **LLaMA-2-13B (ARC)** | | | | |
| — | — | Dense | — | 1.114 / 1.114 / 1.00 |
| 0.3 | 19.8% | Light | 0.043 | 0.503 / 0.545 / 2.12 |
| 0.2 | 31.7% | Heavy | 0.042 | 0.634 / 0.677 / 1.59 |

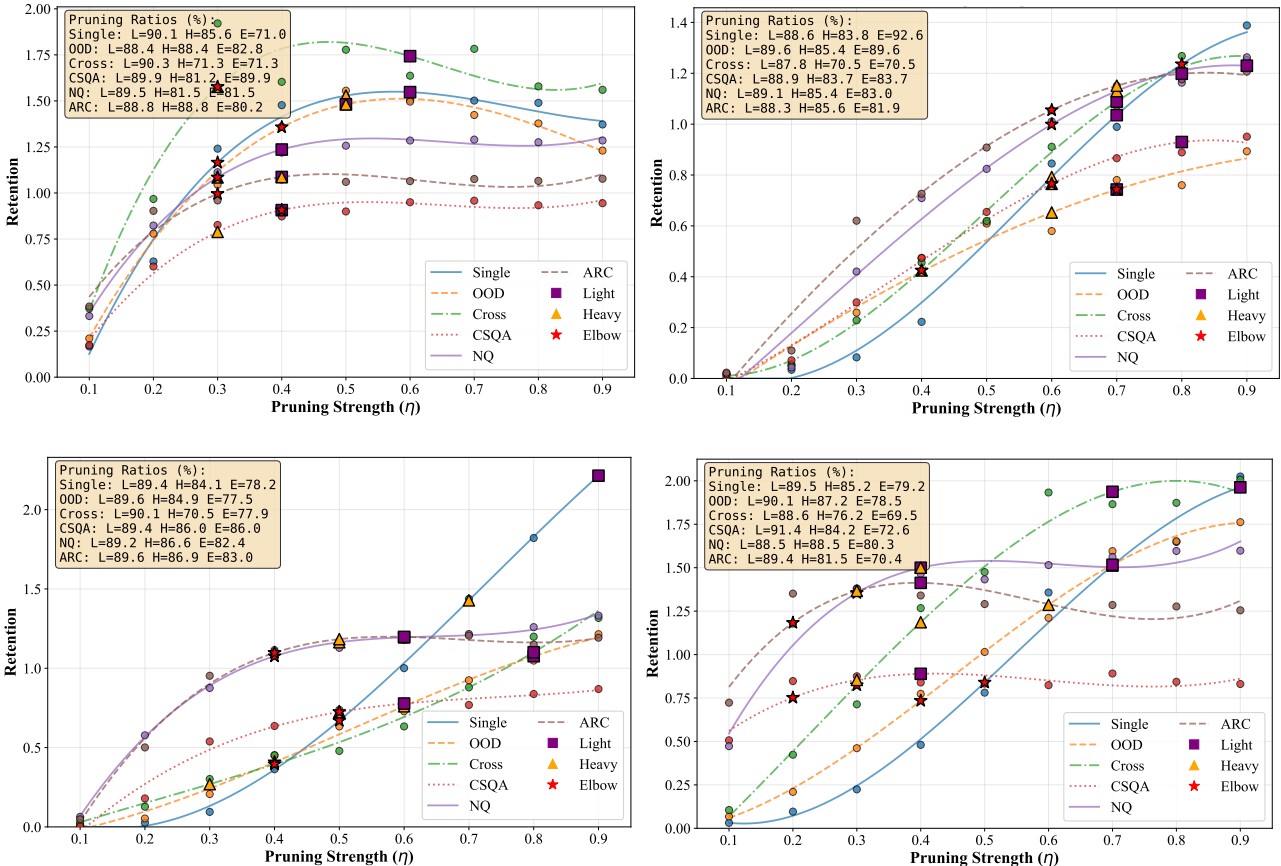

*Figure 19.* **Retention trend analysis for eta parameter selection.** Fitted retention curves (polynomial interpolation) for each model across six datasets: XDomainBench (Selected, OOD, Cross) and Cross-dataset (CSQA, NQ, ARC). Top-left: LLaMA-2-13B. Top-right: LLaMA-3.1-8B. Bottom-left: Qwen2.5-7B. Bottom-right: Qwen2.5-14B. Actual data points from Tables 12 and 13 are overlaid as markers. Vertical dashed lines indicate recommended pruning points: purple (light pruning, $\eta \approx 0.6$), orange (heavy pruning, $\eta \approx 0.3$), and red (elbow point, model- and dataset-specific). The curves reveal retention trends and validate our pruning point recommendations.

justified by performance gains. The results validate that our online compilation approach is practical for deployment, with overhead amortized across multiple samples within each scenario.

## O.5 Eta Parameter Selection Guidelines

This subsection provides guidelines for selecting the pruning strength parameter $\eta$ based on comprehensive efficiency analysis across all models and datasets. We analyze retention trends using fitted curves derived from Tables 12 and 13, and recommend three pruning strategies: light pruning, heavy pruning, and elbow point selection.

**Three Pruning Strategies.** We identify three key pruning configurations based on retention behavior:

- **Light Pruning** ($\eta \approx 0.5$–$0.7$): At moderate pruning ratios (typically 80–90% head retention), performance often improves over dense baseline due to noise reduction effects, while achieving moderate speedup gains (typically 1.08–1.26×). This configuration is recommended when performance improvement is prioritized with acceptable speedup.
- **Heavy Pruning** ($\eta \approx 0.2$–$0.4$): At aggressive pruning ratios (typically 68–72% head retention), performance approaches or slightly degrades from dense baseline, while speedup varies significantly by dataset and hardware (0.97–1.76×). Speedup variation arises from two factors: (1) hardware-dependent mask application overhead (sparse matrix operations may not benefit from optimized dense kernels on some hardware), and (2) matrix sparsity patterns (irregular sparsity can reduce cache efficiency). This configuration requires careful selection and is recommended only when aggressive compression is necessary.
- **Elbow Point** ($\eta$ varies by model and dataset): The elbow point represents the optimal trade-off between accuracy and speed, where retention exhibits a sharp transition from high to low values. This point provides the best balance, typically

*Table 16.* **Ablations for subspace–pathway coupling on Qwen2.5-7B, LLaMA-3.1-8B, and Qwen2.5-14B.** with pruning ratio at the elbow point for each dataset type.

*(a)* LLaMA-3.1-8B

| Ablation | XDB (Selected) | XDB (OOD) | XDB (Cross) | CSQA | NQ | ARC |
|---|---|---|---|---|---|---|
| | $\eta=0.8$ | $\eta=0.7$ | $\eta=0.4$ | $\eta=0.6$ | $\eta=0.6$ | $\eta=0.6$ |
| Dense | $39.1\pm1.57$ | $33.2\pm1.21$ | $21.3\pm0.81$ | $27.77\pm1.30$ | $28.82\pm1.38$ | $22.28$ |
| Full: DBS + PSP | $41.7\pm2.43$ | $35.0\pm1.94$ | $17.9\pm2.88$ | $21.33\pm1.20$ | $29.12\pm1.29$ | $23.50\pm1.13$ |
| w/o DBS selection (random axes) | $1.8\pm0.6$ | $1.4\pm0.4$ | $0.9\pm0.3$ | $1.33\pm0.27$ | $0.33\pm0.10$ | $1.22\pm0.18$ |
| w/o whitelist ($\mathcal{W}=\emptyset$) | $22.4\pm1.9$ | $18.4\pm1.4$ | $24.4\pm2.5$ | $18.37\pm1.14$ | $6.55\pm0.70$ | $16.34\pm0.99$ |
| w/o multi-domain mixing | $6.0\pm1.1$ | $1.4\pm0.5$ | $3.7\pm1.1$ | $1.23\pm0.28$ | $0.54\pm0.13$ | $4.59\pm0.54$ |

*(b)* Qwen2.5-7B

| Ablation | XDB (Selected) | XDB (OOD) | XDB (Cross) | CSQA | NQ | ARC |
|---|---|---|---|---|---|---|
| | $\eta=0.5$ | $\eta=0.4$ | $\eta=0.4$ | $\eta=0.5$ | $\eta=0.4$ | $\eta=0.4$ |
| Dense | $40.8\pm1.6$ | $33.6\pm1.2$ | $22.9\pm0.8$ | $27.75\pm1.28$ | $17.51\pm1.21$ | $21.64\pm1.19$ |
| Full: DBS + PSP | $46.2\pm2.46$ | $36.4\pm1.96$ | $22.5\pm3.14$ | $20.48\pm1.20$ | $19.51\pm1.14$ | $23.90\pm1.16$ |
| w/o DBS selection (random axes) | $3.4\pm0.9$ | $0.8\pm0.3$ | $0.0\pm0.0$ | $0.00\pm0.00$ | $0.12\pm0.10$ | $0.16\pm0.07$ |
| w/o whitelist ($\mathcal{W}=\emptyset$) | $12.6\pm1.6$ | $5.2\pm0.9$ | $22.5\pm2.3$ | $7.73\pm0.80$ | $0.63\pm0.21$ | $0.19\pm0.06$ |
| w/o multi-domain mixing | $13.9\pm1.5$ | $10.6\pm1.1$ | $13.4\pm1.8$ | $6.95\pm0.73$ | $11.31\pm0.87$ | $14.09\pm0.88$ |

*(c)* Qwen2.5-14B

| Ablation | XDB (Selected) | XDB (OOD) | XDB (Cross) | CSQA | NQ | ARC |
|---|---|---|---|---|---|---|
| | $\eta=0.5$ | $\eta=0.4$ | $\eta=0.3$ | $\eta=0.2$ | $\eta=0.3$ | $\eta=0.2$ |
| Dense | $40.8\pm1.60$ | $37.2\pm1.25$ | $22.8\pm0.85$ | $28.42\pm1.26$ | $19.72\pm1.34$ | $20.71\pm1.18$ |
| Full: DBS + PSP | $42.5\pm2.44$ | $40.0\pm2.00$ | $23.7\pm3.20$ | $24.08\pm1.26$ | $27.21\pm1.27$ | $27.98\pm1.18$ |
| w/o DBS selection (random axes) | $4.8\pm1.0$ | $2.7\pm0.6$ | $1.3\pm0.8$ | $0.10\pm0.07$ | $0.25\pm0.11$ | $0.53\pm0.11$ |
| w/o whitelist ($\mathcal{W}=\emptyset$) | $7.5\pm1.2$ | $2.7\pm0.6$ | $19.8\pm2.2$ | $0.00\pm0.00$ | $0.21\pm0.12$ | $0.03\pm0.03$ |
| w/o multi-domain mixing | $18.1\pm1.7$ | $18.4\pm1.4$ | $13.8\pm1.8$ | $3.03\pm0.46$ | $13.83\pm0.93$ | $6.04\pm0.60$ |

achieving retention values above 1.0 while maintaining substantial speedup gains.

**Retention Trend Analysis.** Figure 19 shows fitted retention curves for all four models across six datasets (XDomainBench: Selected, OOD, Cross; Cross-dataset: CSQA, NQ, ARC). Each subplot displays six curves (one per dataset) with actual data points overlaid, enabling visual inspection of retention trends as pruning strength increases.

The fitted curves reveal consistent patterns across models:

- Retention generally decreases as $\eta$ decreases (pruning increases), but the rate of decrease varies by dataset.
- Cross-domain and cross-dataset scenarios often show higher retention at moderate pruning ratios, benefiting from subspace union structure (as established in Section F).
- The elbow point location varies by model architecture and dataset, reflecting model-specific sensitivity to pruning.

Our recommended pruning points (light, heavy, elbow) align with the fitted curve trends:

- **Light pruning** points ($\eta \approx 0.5$–$0.7$) typically lie on the plateau region where retention remains high ($> 1.0$), confirming that moderate pruning preserves performance while providing speedup.
- **Heavy pruning** points ($\eta \approx 0.2$–$0.4$) lie on the steep decline region, where retention drops below 1.0 but remains above catastrophic failure thresholds, validating that aggressive pruning requires careful calibration.
- **Elbow points** (model- and dataset-specific) are identified at the inflection point where retention transitions from plateau to decline, providing optimal efficiency-adjusted performance.

These recommendations are consistent with the empirical trends observed in the comprehensive efficiency tables and provide practical guidance for inference-time pruning configuration.

## P    Ablation Studies for Other Models

This section provides ablation studies for Qwen2.5-7B, LLaMA-3.1-8B, and Qwen2.5-14B, following the same format as Table 4 in the main text (which reports results for LLaMA-2-13B). Table 16 reports the ablation studies for all three models.

**Analysis.** Table 16 confirms the consistent effectiveness of our method's key components across different model architectures. Removing DBS selection (using random axes) causes catastrophic performance drops across all three models, with Recall scores dropping to near-zero on most datasets (e.g., Qwen2.5-7B: from 46.2 to 3.4 on Selected, from 20.48 to 0.00 on CSQA), demonstrating that domain-aware axis selection is fundamental to the method's success. The whitelist mechanism shows dataset-dependent importance: while removing it severely degrades performance on Selected-domain and OOD splits (e.g., LLaMA-3.1-8B: from 41.7 to 22.4 on Selected), it sometimes maintains or even improves performance on Cross-domain tasks (e.g., LLaMA-3.1-8B: 24.4 vs 17.9, Qwen2.5-7B: 22.5 vs 22.5), suggesting that critical heads identified by the whitelist are more crucial for in-domain scenarios. Multi-domain mixing proves essential across all models and datasets, with its removal causing severe degradation (e.g., Qwen2.5-14B: from 42.5 to 18.1 on Selected, from 27.98 to 6.04 on ARC), highlighting the importance of adaptive domain combination for handling diverse knowledge requirements. Notably, larger models (Qwen2.5-14B) show better resilience to component removal compared to smaller models, particularly on cross-dataset tests, indicating improved robustness with increased model capacity. Overall, the ablation results validate that DBS, whitelist, and multi-domain mixing are all necessary components that work synergistically to achieve effective domain-aware pruning.

## Q    Case Studies

This section presents detailed case studies showing the complete inference process for representative scenarios: cross-dataset (Natural Questions single-turn), out-of-domain multi-turn (Biology), and cross-domain multi-turn (Philosophy+Sociology). Each case study includes visualizations of the inference process, domain identification, head scoring, mask compilation, and analysis of how the method adapts to different distribution shifts and knowledge requirements.

### Q.1    Case 1: Cross-Dataset Sample (Natural Questions)

This case study examines a single-turn question from Natural Questions to understand how our method handles cross-dataset scenarios with different question formats and styles. Figure 20 shows the complete inference process for the query "What was guantanamo bay before it was a prison?", including domain identification, head scoring, and mask compilation.

**Analysis.** The Natural Questions case demonstrates how our method adapts to real-world open-domain queries that differ from training data. The query "What was guantanamo bay before it was a prison?" requires factual knowledge retrieval about a specific location's historical context. The system identifies domain relevance (shown as a domain label and similarity score), leveraging the Head Importance Cache and Whitelist Cache to prioritize critical attention heads. The Head Score per Selected Domain visualization shows how different domains contribute to the head selection process, with domain-specific scores aggregated into the Head Score Sum. The final Pruned Subnetwork/pathway (green cells for retained heads, red for pruned) reflects a sparse but effective configuration tailored to this factual query. The method successfully generates the concise answer "Guantanamo Bay Naval Base" by preserving heads essential for geographical and historical knowledge retrieval, demonstrating effective cross-dataset generalization despite the distribution shift from training data.

### Q.2    Case 2: Out-of-Domain Multi-Turn Session (Biology)

This case study examines a multi-turn session in the Biology domain (out-of-domain scenario) to understand how our method handles distribution shifts and reuses pruned subnetworks across multiple interactions. Figure 21 shows a complete multi-turn session focused on protein synthesis, demonstrating inference-time pruning at session start and subsequent reuse of the pruned subnetwork.

**Analysis.** This Biology out-of-domain case demonstrates how our method adapts to distribution shifts and efficiently handles multi-turn interactions. In Turn 1, the system performs inference-time pruning based on the initial prompt about protein synthesis. The domain similarity scores (e.g., [0.9999224, 0.5434691, 0.3046012, ...]) identify relevant domains (Chemistry, finance, philosophy), with the system filtering to selected domains. The session breadth calculation (0.419387) quantifies the domain diversity, which modulates the pruning budget ($\eta = 0.6$). The pruning process results in a subnetwork with 616 heads kept and 308 pruned (39.8% pruning ratio) from a total of 1024 heads. Critically, this pruned subnetwork is

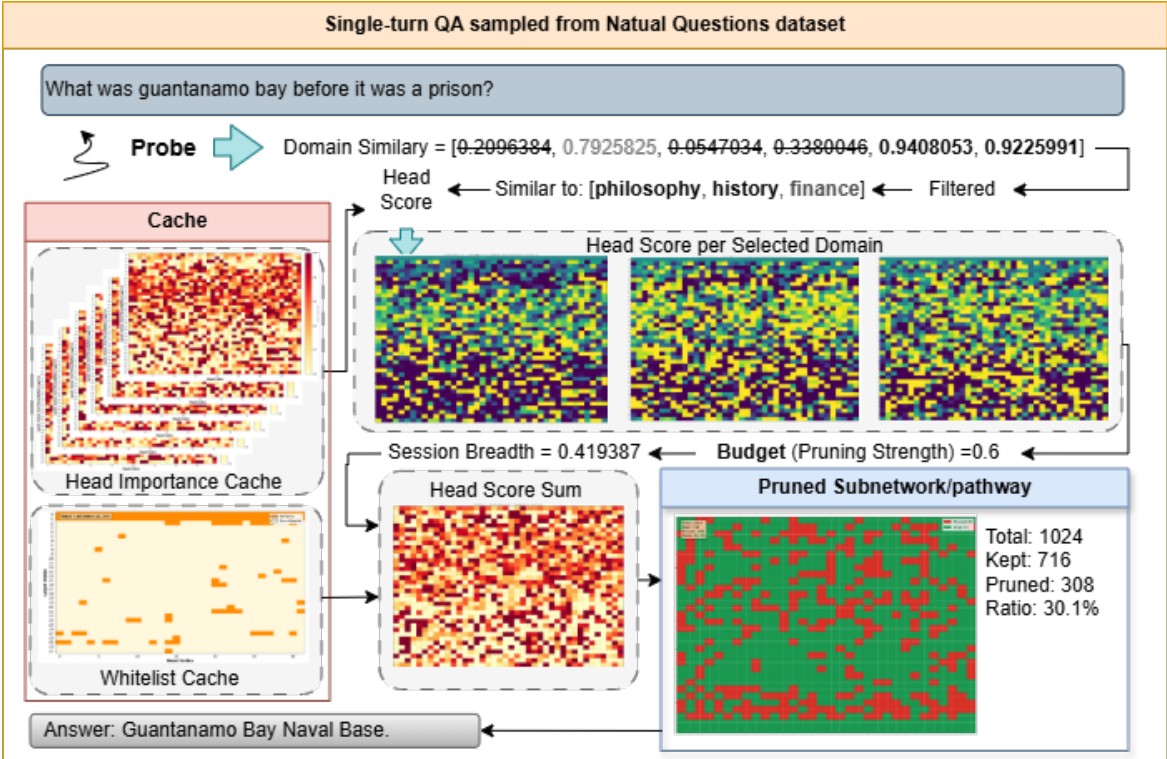

*Figure 20.* **Natural Questions single-turn QA case study.** This figure shows the complete inference process for a Natural Questions sample, including domain similarity scoring, head importance caching, domain-specific head scoring, and the resulting pruned subnetwork, demonstrating how our method identifies relevant knowledge domains and compiles an appropriate mask for factual knowledge retrieval.

marked for reuse across the entire session, as shown by the "Reuse for the entire session" instruction. In Turn 2 and Turn 3, the same pruned configuration is reused without recompilation, demonstrating efficiency gains. The method successfully handles questions about sea anemone toxicity (Turn 2) and protein synthesis (Turn 3) using the same specialized subnetwork, showing that the initial domain-aware pruning captures relevant biological knowledge pathways that generalize across related questions within the session. This case highlights the method's ability to adapt to out-of-domain scenarios while maintaining efficiency through session-level subnetwork reuse.

### Q.3    Case 3: Cross-Domain Multi-Turn Session (Philosophy+Sociology)

This case study examines a multi-turn session spanning Philosophy and Sociology domains (cross-domain scenario) to understand how our method handles multi-domain knowledge requirements and reuses pruned subnetworks. Figure 22 shows a complete multi-turn session on the philosophical and sociological implications of Zoroastrian dualism, demonstrating how the method adapts to cross-domain scenarios.

**Analysis.**    This Philosophy+Sociology cross-domain case demonstrates how our method handles multi-domain knowledge requirements through adaptive pruning and session-level reuse. In Turn 1, the system processes a question about how Zoroastrianism's dualistic philosophy influences societal structures. The domain similarity analysis identifies relevant domains (with scores like 0.6040141, 0.4993707, 0.7383702), and the system filters to selected domains (chemistry, finance, technology are shown for comparison, though the primary domains are philosophy and sociology). The Head Importance Cache and Whitelist Cache provide prior knowledge about critical heads, while the Head Score per Selected Domain visualization shows how different domains contribute to head selection. The Head Score Sum aggregates these domain-specific scores, and the final Pruned Subnetwork/pathway (green for retained, red for pruned) creates a specialized configuration for this cross-domain topic. The system generates the answer "Promotes moral and ethical order," demonstrating effective handling of abstract philosophical and sociological concepts. In Turn 2 and Turn 3, the same pruned subnetwork is reused, with the system providing conversational history as context. The method successfully answers questions about hierarchies ("Cosmic order versus chaos") and justice ("Promotes ethical and moral justice") using the same specialized subnetwork, showing that the initial cross-domain pruning captures relevant pathways for both philosophical reasoning and sociological

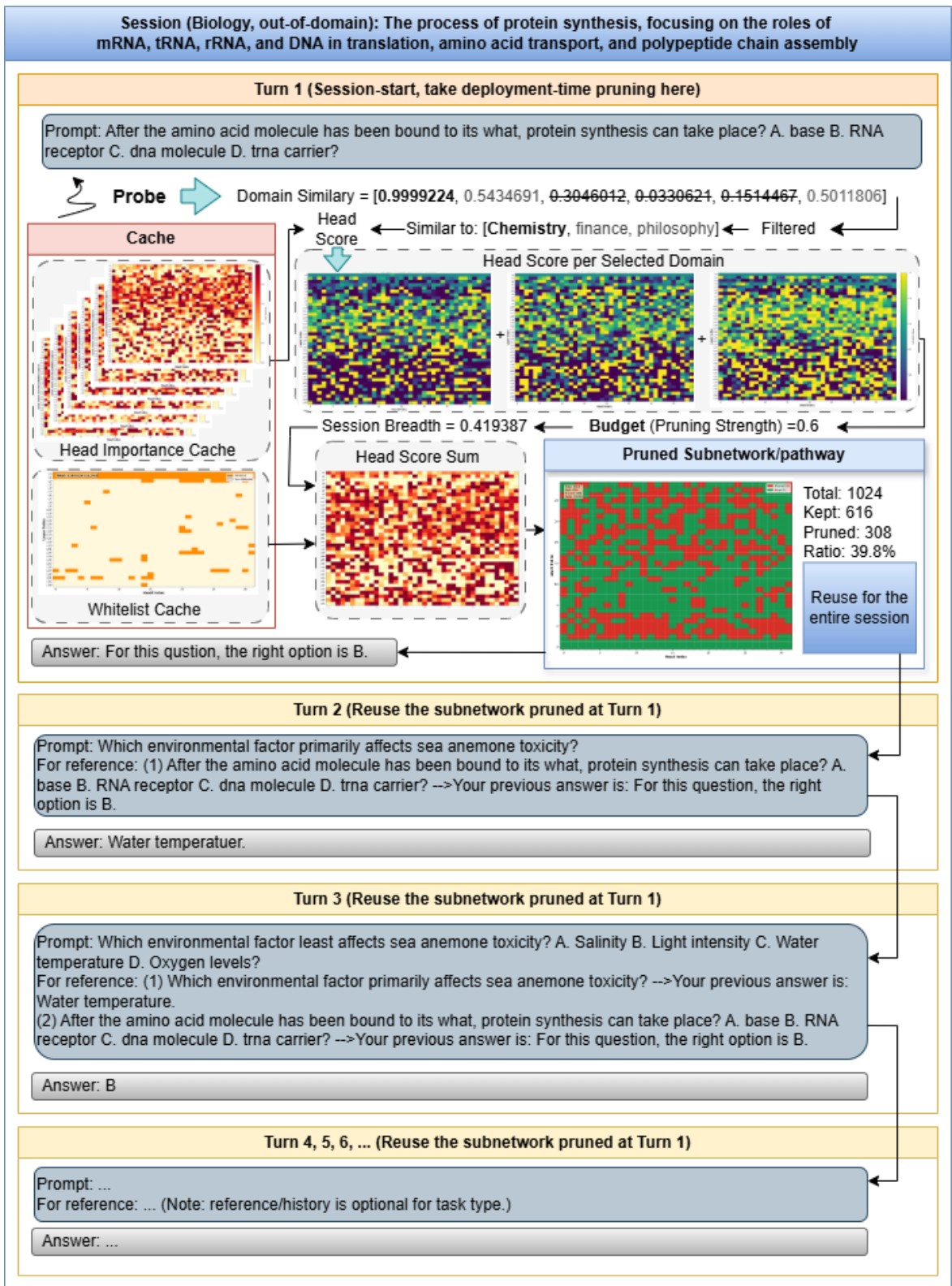

*Figure 21.* **Out-of-domain multi-turn session case study (Biology).** This figure shows a multi-turn session on protein synthesis, including the initial inference-time pruning process in Turn 1 and the reuse of the pruned subnetwork in subsequent turns. The visualization demonstrates domain similarity scoring, session breadth calculation, budget-based pruning, and the efficiency gains from reusing a single pruned configuration across the entire session.

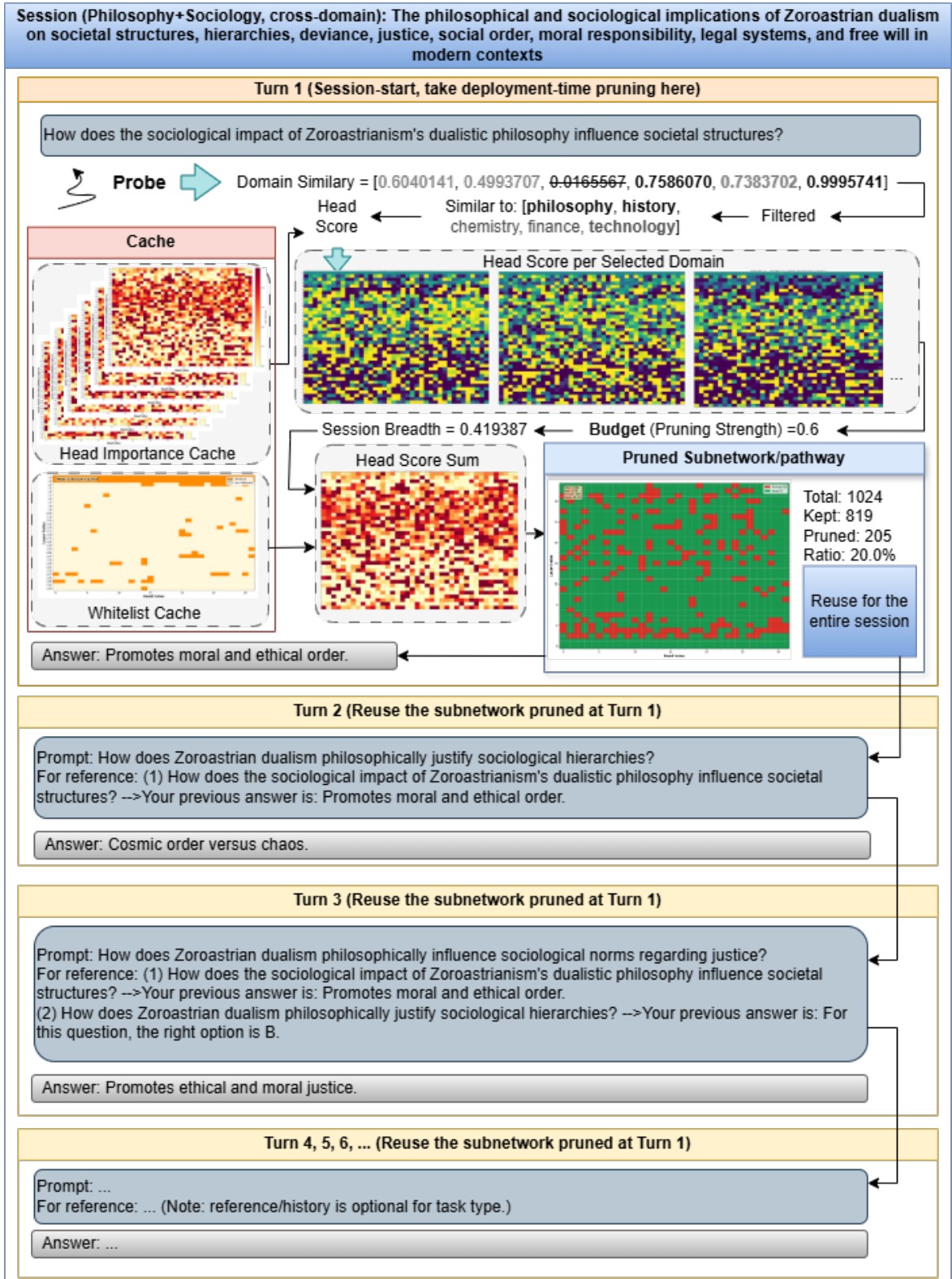

*Figure 22.* **Cross-domain multi-turn session case study (Philosophy+Sociology).** This figure shows a multi-turn session on Zoroastrian dualism's implications for societal structures, hierarchies, justice, and social order. The visualization demonstrates the initial inference-time pruning in Turn 1, which creates a specialized subnetwork for this cross-domain topic, and the reuse of this subnetwork in subsequent turns with accumulated conversational history.

analysis. This case highlights the method's ability to handle complex multi-domain scenarios while maintaining efficiency through session-level subnetwork reuse, with the pruned configuration effectively supporting both abstract reasoning and domain-specific knowledge retrieval.

**Cross-domain generalization insights.** The Philosophy+Sociology case study reveals several key insights about our method's cross-domain handling:

- **Multi-domain integration**: The method successfully integrates knowledge from multiple domains (philosophy and sociology) by identifying relevant domain axes and combining their contributions in head scoring.
- **Session-level efficiency**: Reusing the pruned subnetwork across multiple turns within a session eliminates recompilation overhead while maintaining performance on related questions.
- **Abstract reasoning preservation**: The pruned subnetwork preserves heads essential for abstract reasoning and conceptual understanding, enabling effective handling of philosophical and sociological queries.
- **Conversational context**: The method leverages accumulated conversational history to provide context for subsequent turns, enhancing coherence across the multi-turn session.

# Part E: Limitations and Discussion.

## R  Limitations

This section discusses the limitations of our method based on the experimental results and inference considerations.

**Method limitations.**

- **Domain pool dependency**: Our method requires input-only pools $\{\mathcal{P}_k\}$ for offline DBS preparation. The quality and representativeness of these pools directly affect the constructed axis basis and subsequent pruning performance. As shown in Table 4, removing DBS selection (using random axes) causes severe performance degradation, particularly on OOD and cross-domain scenarios.
- **Pruning ratio sensitivity**: While our method achieves good performance at moderate pruning ratios (elbow points), aggressive pruning beyond the elbow point leads to significant performance degradation, as illustrated in Figure 5. The method's effectiveness depends on selecting appropriate pruning ratios for each dataset type.
- **Task-type variation**: Performance varies across task types (multiple_choice, factual, reasoning, code) as shown in Table 11. While our method generally maintains or improves performance on multiple_choice and factual tasks, reasoning tasks show more sensitivity to pruning, particularly under heavy pruning configurations.

**Experimental limitations.**

- **Model and dataset scope**: Our evaluation covers four models (7B–14B parameters) and four datasets (XDomainBench, CommonsenseQA, Natural Questions, ARC). The method's effectiveness on larger models (e.g., 70B+) and other task types (e.g., generation, summarization) requires further validation.
- **Baseline comparison**: While we compare against representative pruning methods, the comparison is limited to training-free methods under the same pruning ratios. Methods that allow fine-tuning or additional training may achieve different trade-offs.

**inference considerations.**

- **Compilation overhead**: Mask compilation occurs once per scenario at scenario start, with overhead typically 0.02–0.08 seconds per scenario (Table 15). For very short scenarios or high-frequency scenario switching, this overhead may be non-negligible.
- **Scenario boundary assumption**: Our method assumes clear scenario boundaries where domain mixture can be estimated from a scenario prefix. Handling dynamic or ambiguous scenario boundaries, or scenarios with rapidly changing domain distributions, may require additional mechanisms.

## S  Future Work

Based on our experimental findings and limitations, we identify several directions for future research:

- **Adaptive pruning ratio selection**: Developing automatic methods to select optimal pruning ratios based on scenario characteristics, rather than relying on fixed elbow points.
- **Task-aware pathway selection**: Extending the method to explicitly consider task type (multiple_choice, factual, reasoning, code) in pathway compilation, potentially improving performance on reasoning tasks.
- **Integration with quantization**: Combining our head-level pruning with quantization techniques for further memory and inference efficiency gains, while maintaining the training-free inference paradigm.
- **Extension to generation tasks**: Adapting the method for generation tasks (e.g., summarization, translation) where evaluation metrics and pruning criteria may differ from question-answering tasks.

