# OpenReview forum: "SubspacePath Pruner: Inference-time Pruning via Probe-based Representation–Parameter Coupling"
_ICML.cc/2026/Conference — ICML 2026 regular_

### Official Review · Reviewer_7bHc · 2026-02-28

**Soundness:** 3
**Presentation:** 3
**Significance:** 4
**Originality:** 3
**Overall Recommendation:** 4
**Confidence:** 3

**Summary:**

This paper introduces SubspacePath Pruner, a method for inference-time structured pruning of LLMs. The core idea: offline, use PCA on text pools to define "semantic axes" (domains) and compute each attention head's importance relative to these axes. Online, during the first few tokens of a session, use lightweight probes to identify which domains are relevant, then keep only the heads important for those domains, subject to a budget constraint. The result is an executable subnetwork specialized for that session, compiled in milliseconds.

**Compliance With Llm Reviewing Policy:**

Affirmed.

**Key Questions For Authors:**

**On the Non-Linear Transformation (Over-Optimization Risk)**

Equation (45) introduces a non-linear transformation with a power exponent γ = 1.5 applied to domain similarities before head scoring. This is described as amplifying high similarities and reducing low ones.

The paper's limitations (Appendix R) state that "the method's hyperparameters (e.g., pruning strength η, whitelist thresholds) may require tuning for new deployment contexts." However, γ = 1.5 is presented as fixed.

Sensitivity analysis: How sensitive are the results to γ? Have you tested γ = 1.0 (linear), 1.2, 2.0? If performance drops significantly at γ = 1.0, that suggests the scoring function is fragile.

**On Baselines (Isolating the Core Contribution)**

You compare against strong methods (Wanda, LLM-Prune, RIA, Probe Pruning), which is good. But you're missing a simple, interpretable baseline: top-K head selection by weight magnitude.

Why this matters:

Measuring added value: If a simple magnitude-based baseline performs reasonably well, it helps isolate what your method actually adds. If it performs poorly, it strengthens your case.

Computational cost: Magnitude-based selection is essentially free (just look at pre-existing weights). Your method requires PCA, probe training, and online inference—a significant upfront cost. The comparison should justify that cost.

Related to limitations: The paper notes (Appendix R) that "the offline preparation stage (DBS + PSP) requires access to input-only pools and computational resources." A magnitude baseline has no such requirements—it's a pure "zero-shot" pruner. Comparing against it would clarify the trade-off.

Concrete request: Can you add a simple baseline: for each layer, keep the top-K heads by the L2 norm of their weight matrices (or attention output projection), where K is chosen to match your pruning ratio? Report results on at least one model and dataset (e.g., LLaMA-2-13B on NQ). This would help disentangle the effect of "semantic pruning" from simply "removing small heads."

Theoretical justification: The paper says γ = 1.5 "balances amplification effect." Balanced against what? Is there a principled reason for 1.5, or was it chosen by grid search on XDomainBench?

Alternative formulations: Did you consider softmax temperature scaling or other adaptive methods instead of a fixed power transformation? These might be more theoretically grounded.

Concrete request: Can you provide an ablation study showing performance across different γ values (e.g., 0.5, 1.0, 1.5, 2.0, 3.0) on at least one public dataset (e.g., NQ or ARC)? This would demonstrate whether the specific value is critical or if the method is robust to this choice.

**Limitations:**

The authors are reasonably transparent about limitations. Performance depends on DBS axis quality: if input pools are noisy or overlapping, the semantic axes degrade. The method also requires domain-specific pools offline—data that may not be available in some applications. Online compilation overhead (0.02-0.08s) may not amortize for very short sessions (e.g., single turns), potentially increasing latency. Speedup is hardware-dependent: sparse mask overhead can outweigh benefits on some hardware. Whitelist thresholds may need tuning for new architectures or domains. Finally, performance on highly specialized tasks (low-resource languages, code generation) may vary. The impact statement is generic, but the limitations section is thorough and honest.

**Strengths And Weaknesses:**

**Strengths:**

- Coupling representation space (PCA axes) with parameter space (attention heads) via probes is conceptually clean. Instead of pruning blindly based on gradients or weight magnitude, you prune based on what the model is semantically doing at inference time.

- Where Wanda and RIA collapse to 4-5% recall on OOD data, SubspacePath holds at 20-30%. That's not noise—that's a meaningful difference.

- 0.03-0.06 seconds per scenario, vs. 0.5+ for baselines. For real-world deployment (agents, chatbots), this matters.

- 4 models, multiple datasets, ablations on every component, trade-off analysis, case studies. They did their homework.

**Weaknesses:**

The XDomainBench dataset is anonymous and non-public. This is a real problem. How can I verify that the domains are well-separated, the splits are clean, or the benchmark wasn't tailored to make their method shine? It's a blind spot. The results on public datasets (CSQA, NQ, ARC) help, but don't fully erase the concern.

The method is complex. PCA, probes, whitelist with statistical thresholds, non-linear transformation, entropy-based breadth, budget mixing... It feels like they added pieces until it worked everywhere. The ablations suggest these pieces are necessary, but the risk of overfitting to their specific benchmark remains.

Novelty relative to "Probe Pruning" (Le et al. 2025) is fuzzy. Both use probes to guide pruning. The differences (PCA axes, semantic grounding) are real, but they'll need to defend this clearly in the response

---

> ### Author Rebuttal · Authors · 2026-03-31
>
> # Response to Reviewer 7bHc
>
> We thank the reviewer for the careful and constructive assessment, especially for recognizing our novelty, robustness, deployability, and workload, with concerns regarding **(1) experimental fairness, (2) robustness and alternatives, and (3) position**.
>
> ## Part 1. Experimental fairness
>
> > **(1)** benchmark-specific fitting is argued with **default protocal, domain-separation evidence**, and **broader benchmark transfer**;
> > **(2)** hyperparameters are **lightweight calibrations** rather than fragile benchmark-tuned knobs.
>
> ### 1.1 Fairness of XDomainBench (Weakness 1)
>
> We fully understand reviewer’s concern about XDomainBench. However, the current gains are **not benchmark tailoring**.
>
> (1) **Protocol already strictly separated.**
> DBS and probe use only a small *selected* split with no ground-truth to get the sementics-parameter alignment, while evaluation is on remaining splits and wider public benchmarks without retraining.
>
> (2) **Separable split.**
> * **High-quality dataset**: All samples are drawn from specialized-domain sources and manually checked by experts.
> * **Separability statistics**: Average pairwise orthogonality ~0.82, range ~[0.71, 0.95], and nearest-axis purity ~0.95, prove considerable separability of domains.
> * **Counterevidence**: Poor separation would make the **probe** even harder to fit, no more to say its 98.7\%/96.5\% training/validation **accuracy** and final **generalization performance**.
>
> (3) **Transfers beyond XDomainBench.**
> While paper already evaluates on CSQA, NQ, and ARC, and we further add **four new broader public benchmarks** with results in `Reviewer SHsP Part3`.
>
> ### 1.2 Over-optimization risk (Question 1)
>
> (1) We appreciate reviewer’s concern about whether our method is *fragile to tiny non-linear calibration*.
>
> |LLaMA-13B Recall(%)|0.5*|ReLU|1.2|1.5|2.0|3.0|softmax|tanh|
> |-|-|-|-|-|-|-|-|-|
> |NQ(agg.)|20.61|24.31|24.67|24.89|24.95|24.18|25.02|24.97|
> |ARC(agg.)|19.74|21.08|21.42|21.56|21.61|21.07|21.68|21.63|
>
> The raise of this calibration is to balance two failure modes of **under-amplifying clearly relevant domains** and **over-preserving weakly related ones**, while being not sensitive for our method.
>
> (2) Other hyperparameters mentioned.
>
> * Pruning strength chosen from **retention frontier** (Fig. 19).
> * Whitelist thresholds **sensitivity analysis**.
>
> |NQ Recall(%)|0%*|2%*|5%|8%|10%|15%|
> |-|-|-|-|-|-|-|
> |Moderate|0.68|21.94|32.88|33.66|33.57|33.18|
> |Aggressive|0.15|11.72|24.21|24.89|24.75|24.33|
>
> ## Part 2. Robustness and alternatives
>
> > **(1)** **integral** framework with components following the core **semantic-parameter coupling**;
> > **(2)** Top-k baselines confirm our contribution: **semantic relevance** with *negligible online cost*.
>
> ### 2.1 Robust integral framework (Weakness 2)
>
> We have the same pursuit as review's for an integral framework with the core—**compile a subnetwork using generalized semantic geometry**. Under this view, the components are consecutive stages following the core smoothly:
>
> - **DBS with PCA**: constructs a stable *semantic* coordinate system;
> - **Probe**: estimates *scenario-to-axis* alignment;
> - **Head importance with breadth and budget**: translates semantic relevance into executable *head selection*;
> - **Whitelist and transform**: ensure conservative and deployable *compilation*.
>
> Ablations in paper and `Reviewer SHsP Part2.2` further prove modules are **indispensable**.
>
> ### 2.2 Top-k baseline (Questions 2,3)
>
> We agree top-k's value of isolating our core contribution, since **high magnitude does not equal high scenario relevance**.
>
> |Recall(%)|Dense|Weight Top-k|Activation Top-k|Ours|
> |-|-|-|-|-|
> |NQ(mod.)|30.25|27.70|27.83|33.66|
> |NQ(agg.)|30.25|18.02|16.99|24.89|
> |ARC(mod.)|23.87|23.44|23.60|22.91|
> |ARC(agg.)|23.87|20.63|19.74|21.56|
>
> Top-k can retain globally useful heads, but cannot reliably identify the heads most aligned with scenario—especially under stronger pruning. Our gain therefore mainly comes from **semantic alignment**, instead of static salience heuristic, which can partially prove **extra cost is worthwhile** in Part 2.3.
>
> ### 2.3 Computational cost (Questions 3,4)
>
> Extra cost is negligible and worthwhile, while we have a thorough discussion on **efficiency and deployability** in `Reviewer mhAG Part2`, confirming our practical value.
>
> * **Offline**: DBS and probes are lightweight and **reused across datasets and domains** without retraining.
> * **Online**: the only extra cost is a tiny **one-time scenario-start compilation ~0.024-0.060s** shown in Tables 2, 13.
>
> ## Part 3. Positioning (Weakness 3)
>
> > two methods differ in **signal, deployment unit, and reuse**.
>
> |Aspect|Probe Pruning (Le et al. 2025)|Ours|
> |-|-|-|
> |Probe signal|activation importance|semantic axis relevance|
> |Deployment|input-/batch-time pruning|scenario-level compile once|
> |Reuse|dynamic per input|reuse throughout the scenario|
> |Principle|importance-guided pruning| subspace--pathway coupling|

---

> > ### Author Rebuttal · Reviewer_7bHc · 2026-04-01
> >
> > I would like to thank the authors for their detailed and thorough responses. After reviewing their clarifications, I am satisfied with the explanations regarding the methodology, robustness, and positioning of the work.
> >
> > Based on this, I maintain my current score and recommendation.

---

> > > ### Author Response · Authors · 2026-04-06
> > >
> > > We are sincerely grateful for your thoughtful review and follow-up acknowledgement. We especially appreciate that, after reading our rebuttal, you found the concerns on *methodology, robustness, and positioning* satisfactorily addressed and expressed your *recommendation*. Your questions really helped us sharpen both the paper’s evidence chain and its presentation.
> > >
> > > ---
> > >
> > > From reviewer's perspective, we understand the main value of the paper as: **(1) Conceptual clarity:** a clean coupling between *representation-space semantics* and *parameter-space pathways*, so pruning is guided by what the model is semantically doing at inference time; **(2) Practical robustness:** meaningful OOD / cross-domain *gains* together with millisecond-level scenario compilation, making the method *relevant for real deployment*; **(3) Experimental support:** *broad evaluation and targeted ablations* that make the empirical case substantially more credible.
> > >
> > > ---
> > >
> > > In rebuttal, we mainly strengthened three aspects:
> > >
> > > **1. Experimental fairness**
> > > **(1) Protocol.** We clarified the strict separation: DBS / probes use only a small selected split, with *no answers, no OOD/cross-domain samples, and no public test data*.
> > > **(2) Evidence.** We added **domain-separation statistics** and broader **public-benchmark transfer**, showing that the gains are not explained by benchmark tailoring.
> > >
> > > **2. Robustness and alternatives**
> > > **(1) Method structure.** We clarified that the method is **one coupling pipeline**, not a collection of patches: each component serves a distinct stage of semantic-to-pathway compilation, consistent with both the paper’s ablations and the additional evidence.
> > > **(2) Simpler baselines / cost.** We added **Weight / Activation Top-k** baselines and **hyperparameter sensitivity analyses**. These show that *static salience does not reliably recover scenario relevance*, while the extra online cost remains only **~0.03–0.06s once per scenario** and is amortized by reuse.
> > >
> > > **3. Positioning**
> > > **(1) Probe Pruning.** We clarified the difference in **signal source, deployment unit, and reuse structure**: their probes track activation importance, while ours estimate semantic axis relevance and compile once for the scenario.
> > > **(2) Core novelty.** We clarified that our novelty is **not** “using probes” in general, but using **semantic coordinates** to compile **reusable scenario-level pathways** through subspace--pathway coupling.
> > >
> > > ---
> > >
> > > Thank you again for the generous and careful evaluation. We truly appreciate the chance to improve the work through your feedback, and we hope the revised clarification makes the contribution and practical value even clearer. We wish you all the best.

---

### Official Review · Reviewer_mhAG · 2026-03-02

**Soundness:** 3
**Presentation:** 4
**Significance:** 3
**Originality:** 3
**Overall Recommendation:** 4
**Confidence:** 3

**Summary:**

This paper presents SubspacePath Pruner, a novel inference-time pruning framework for Large Language Models (LLMs) designed to address the issue of scenario-specific LLM specialization under the constraints of heterogeneous. Instead of re-training a model for every smaller task, the authors found a shortcut: they mapped how specific "thoughts" in the model's embedding space link to specific "work routes" in its parameters. By using a probe, they can carve out a mini-specialist model from the giant original one, tailored for specific scenarios while keeping the computational costs low.

**Compliance With Llm Reviewing Policy:**

Affirmed.

**Key Questions For Authors:**

1. Your data shows the pruned model actually beating the full (dense) model in some cases (like that 29.3 vs 24.7 Recall). Is this just because you’re cutting out the "noise"? By pruning irrelevant parameters, are you basically giving the model a regularization boost that keeps it from getting distracted by cross-domain interference?
2. You mentioned mask compilation is lightning-fast (~0.06s). Does re-indexing those head-weights or updating the computation graph cause any "stuttering" or latency spikes on the GPU that aren't showing up in your "Pruning Time" stats?
3. To find the right reasoning pathway, the model needs to "get the vibe" of the scenario first. What’s the minimum for that scenario-start prefix? Can the model figure out the domain mixture ($q_k(s)$) from just a small instruction, or does it need a couple of full conversational turns to stabilize the subnetwork?
4. Your method essentially creates a 'virtual MoE' at inference time. Why not use a Sparse MoE architecture from the start? Is the goal to provide MoE-like efficiency for dense-only models that weren't originally trained with routing logic?

**Limitations:**

yes

**Strengths And Weaknesses:**

This research establishes a vital structural connection between representation subspaces, the internal "thought" patterns of a model, and its physical reasoning pathways, or the specific neurons it activates. This insight facilitates efficient inference-time pruning, allowing the model to streamline its operations during use. Furthermore, the approach demonstrates significant robustness against distribution shifts; unlike more fragile architectures, it maintains performance even when the input data deviates from the initial training expectations.

The primary limitation of this method is its heavy dependency on initial context. Because the pruning logic relies on the scenario-start prefix (the first few words or instructions), a mismatch at the very beginning can lead to a subnetwork that performs poorly on the rest of the task. Additionally, the model faces size limits due to its reliance on head-wise structured pruning. While this choice is optimized for hardware execution and speed, it cannot achieve the extreme compression ratios typically seen in more granular, unstructured pruning methods.

---

> ### Author Rebuttal · Authors · 2026-03-31
>
> ## Response to Reviewer mhAG
>
> We thank the reviewer for the positive assessment of our core structural insight and robustness, and for the insightful concerns regarding **(1) effectiveness and robustness, (2) efficiency and deployability, and (3) relation to MoE**.
>
>
> ## Part 1. Robustness and effectiveness
> > **(1)** method already **stable** under strict **low-context setting**;
> > **(2)** gain of pruning is **coupling-guided reorganization of computation**, instead of generic sparsity or noise removal alone.
>
> ### 1.1 Robustness (Weakness 1 + Questions 3)
>
> Reviewer's concern about dependency on prefix is worth discussion, while our current default already uses the strict low-context setting \($T_0=1$\) outright.
>
> 1. **Mechanistic discussion.**
>
> We estimate **soft domain mixture** from sementic information of scenario prefix instead of brittle local activation cues, while scenario's vibe always tuned in first turn. We further give three robustness contributors:
>
> * **Stable semantic target**: semantic space can precisely capture the dominant scenario information, which is the reason why our method generalizes well. `[Reviewer SHsP Part2.1]`
> * **Stable design**: the semantic anchor is stabilized by *stopword filtering + shared PCA + DBS selection*. `[Reviewer SHsP Part2.2]`
> * **Stable compilation** : the compiled selection is not fluctuating after initial compilation with following evidence.
>
> 2. **Empirical discussion.**
>
> To further show the robustness of our method, we progressively reveal future turns and check whether this materially changes performance during compilation on Llama-2-13B (moderate/aggressive; pruning setup in Table 1).
>
> |Prefix|Selected|OOD|Cross|
> |-|-|-|-|
> |Turn 1|43.0/34.7|32.5/30.4|20.2/22.9|
> |Turn 1-2|43.2/35.0|33.8/30.8|20.5/23.5|
> |Turn 1-3|43.0/34.5|33.2/30.5|20.6/22.9|
> |Turn 1-4|43.2/35.1|32.9/29.9|20.8/23.2|
>
> ### 1.2 Effectiveness (Questions 1)
>
> We really appreciate reviewer's insight about the *root of our performance improvement* and we agree that noise reduction is just one contributor.
>
> * Evidence 1: **Ablations** in paper and `Reviewer SHsP Part2.2` and **Top-K baseline** in `Reviewer 7bHc Part2.2` have proved that gains depend on **scenario-aware coupling and selection, not arbitrary pruning**.
> * Evidence 2: Hence, the gain is best understood as **denoising together with pathway structural regularization**, making retained inference-time computation *better organized for current scenario*.
>
> Concretly, what explains the gain more essentially is that our method *reshapes which pathways remain active* by suppressing **scenario-irrelevant or cross-domain-conflicting heads**, and thereby reduces latent pathway competition inside the model.
>
> * **Inference-time structural regularization:** No longer to explore all latent pathways equally; pruning restricts inference to a smaller, scenario-consistent subnetwork, and sharpens the effective decision boundary.
> * **Cleaner evidence aggregation:** Heads carrying *competing domain tendencies* are prevented from the same residual stream, reducing cross-domain interference and making final evidence aggregation cleaner and reliable.
>
> ## Part 2. Efficiency and deployability
>
> > **(1)** extra online cost is **once per scenario** and **negligible** in practice;
> > **(2)** strong performance in **actual deployability** compared to unstructured pruning.
>
> ### 2.1 Efficiency (Question 2)
>
> Reviewer's emphasis on efficiency is noteworthy. We report the GPU-side breakdown (s) below for LLaMA-2-13B, together with the *amortized per-turn time* of the dense model and the strongest baselines (~5.7 turns per scenario).
>
> |Setting|Scenario-start overhead|Infer|Avg./turn|Dense| Structured LLM-Pr.|Unstructured Sparse-GPT|
> |-|-|-|-|-|-|-|
> |CSQA|0.043|0.545|0.553|0.588|0.750|71.753|
> |NQ|0.042|0.449|0.456|0.486|0.656|71.756|
> |ARC|0.043|0.503|0.511|1.114|1.289|72.414|
>
> In short, **our extra cost is small, one-time, and amortized by scenario reuse**, which is exactly the deployment regime that we target at.
>
> ### 2.2 Deployability (Weakness 2)
>
> We agree that head-wise structured pruning is *not the route to the most extreme compression ratios*. But this is a **design choice**, not a hidden weakness. Deployment discussion in `Reviewer L6S5 Part 1.3(a)` reveals that practical bottleneck is whether **executable and lightweight**.
>
> By contrast, *unstructured pruning* can sometimes reach extreme sparsity, but it typically brings a much larger **specialization cost**. (OOD/Cross/CSQA/NQ)
>
> |Method|Recall(%)|Avg./turn(s)|Pruning cost(s)|Takeaway|
> |-|-|-|-|-|
> |Ours|32.5/20.2/19.43/33.66|0.564/1.394/0.553/0.456|0.060|balanced wins, minimal cost|
> |SparseGPT|28.34/20.32/21.87/28.37|72.048/75.538/71.753/71.756|403.481|wins on 2, but too costly|
>
> ## Part 3. Relation to MoE (Question 4)
>
> We already have a thorough disscussion in relation to MoE in `Reviewer L6S5 Part1` with evidence, showing that two methods differ fundamentally in **conception, mechanism, and motivation**.

---

> > ### Author Rebuttal · Reviewer_mhAG · 2026-04-01
> >
> > I am satisfied with the response and maintain my recommendation for acceptance.

---

> > > ### Author Response · Authors · 2026-04-06
> > >
> > > We are truly grateful for your thoughtful evaluation, generous recognition, and updated acknowledgement confirming that the concerns were fully resolved while *maintaining the recommendation for acceptance*. We especially appreciate your insightful questions, which helped us sharpen the paper’s mechanism-level explanation and make the deployment setting much clearer.
> > >
> > > ---
> > >
> > > From the reviewer’s perspective, we believe the paper’s main strengths are: **(1) Structural insight:** identifying a meaningful connection between *semantic representation subspaces* and *executable reasoning pathways*, which enables scenario-specific specialization *without retraining*; **(2) Robustness:** showing *strong behavior under distribution shift*, so the method is not limited to narrow in-domain settings; **(3) Practicality:** turning this insight into a *lightweight inference-time specialization framework for frozen dense checkpoints*, with clear deployment relevance.
> > >
> > > ---
> > >
> > > In rebuttal, we mainly addressed three aspects:
> > >
> > > **1. Robustness and effectiveness**
> > > **(1) Robustness.** We explained why the method already works under the strict $T_0=1$ setting: multi-turn scenarios naturally share a **coherent semantic signal**, our semantic anchor is stabilized by **stopword filtering + shared PCA + DBS selection**, and the compiled top-set remains stable across turns. We further supported this with a direct **prefix-reveal analysis** showing that exposing later turns changes performance only marginally.
> > > **(2) Effectiveness.** We explained that the gain is not merely generic sparsity or simple noise removal, but **coupling-guided, scenario-conditioned reorganization of computation**: the method suppresses scenario-irrelevant / cross-domain-conflicting heads and compiles a more scenario-consistent subnetwork. This was supported both **conceptually** and **empirically** through ablations and cross-referenced evidence on semantic anchoring and structured selection.
> > >
> > > **2. Efficiency and deployability**
> > > **(1) Efficiency.** We made the online cost explicit as a small **once-per-scenario** overhead, and reported amortized per-turn timing to show that the extra cost is negligible in the intended deployment regime.
> > > **(2) Deployability.** We also made the design trade-off against **unstructured pruning** more explicit: while unstructured methods may occasionally be competitive in accuracy, their specialization cost is far larger, making them much less suitable for scenario-level online compilation. This highlights the practical balance achieved by our structured design.
> > >
> > > **3. Relation to MoE**
> > > **(1) Positioning.** We clarified that the method is not intended as a native sparse MoE replacement, but as a post-training specialization route for **already-deployed frozen dense checkpoints**.
> > > **(2) Distinction.** We made explicit that the two approaches differ fundamentally in **stage, sparse object, and deployment assumption**: MoE builds sparse specialization during training, whereas our method recovers useful specialization after training from a fixed dense model without retraining or architectural replacement.
> > >
> > > ---
> > >
> > > Thank you again for the thoughtful review, encouraging assessment, and the opportunity to refine the paper through this exchange. We sincerely appreciate your support, and we hope the revised clarification makes the paper’s contribution, mechanism, and practical value much clearer. We wish you all the best.

---

### Official Review · Reviewer_SHsP · 2026-03-13

**Soundness:** 3
**Presentation:** 2
**Significance:** 3
**Originality:** 3
**Overall Recommendation:** 4
**Confidence:** 3

**Summary:**

This paper proposes a model pruning method for LLM inference for specialized domain tasks. The method is motivated by the geometric structure of domain knowledge in representation/embedding space, and its correlation with the activation pattern of attention modules. The proposed method is well-motivated and carefully designed (with sufficient supporting results/ablation studies), and exhibits notable performance/efficiency gains compared to dense model and other pruning methods. However, the current version of the paper has critical issues with the presentation and illustrative figures, while missing some explanations of the core components of the method, thus hindering the understanding of the work.

Based on the summary above, I believe this paper has its merits, while also having rooms for improvement to meet the criteria of ICML. Right now I would lean toward weak reject, and I hope to improve my assessment from discussion with the authors.

**Compliance With Llm Reviewing Policy:**

Affirmed.

**Final Justification:**

The rebuttal addressed my main concerns.

**Key Questions For Authors:**

1. What is the definition of the fixed sequence-to-vector map that produces one embedding vector per input $\phi: X\to \mathbb{R}^d$? Is it a predefined mapping or a learned mapping? The definition of this mapping is critical to have better understanding of the proposed framework.

2. From the results shown in Table 1, while the SubspacePath achieves notable gains on other pruning methods, I notice that the standard error is sometimes significantly higher than other methods (e.g. XdomainBench Selected and Cross), which suggests that there is potential instability issue of the proposed method. Could the authors provide some insights into the high variance of evaluation results?

3. For the computation of domain axis in Section 4.1.1, the authors use the mean of all embeddings in the domain pool $\mu_k = \frac{1}{|P_k|}\sum_{x\in P_k}\tilde{e}(x)$ as the domain axis. Isn’t this a very crude first-order method? In the domain pool $P_k$, the domain information should be very sparse, because most tokens should be general/not domain-specific tokens, thus taking the mean might overemphasize generic content and underrepresent domain-specific information. Could the authors provide more insights into why this design is effective for creating the domain basis?

4. Can the proposed method potentially be applied to broader scenarios, such as commonsense knowledge or reasoning capabilities? If so, it will be a great bonus for the paper.

**Limitations:**

Yes. The appendix discusses several technical limitations.

**Strengths And Weaknesses:**

Strengths

1. The proposed pruning method is novel, cleverly connects the domain-specific embedding subspace to reasoning pathways in the parameter space. The proposed method has sufficient motivations, grounded by empirical observation on the separability of domain knowledge on the embedding space.

2. The paper provides evaluation results on representative benchmarks and models for the selected application scenario, demonstrating the effectiveness of the proposed method.

3. Detailed and extensive ablation studies are also provided in the appendix, showing the roles of each components of the method. Overall the experimental results are solid.

Weaknesses

1. Paper presentation: the illustrative figures are not very straightforward and have significant room for improvement. For example, in Figure 2, the subfigures in the plot do not coordinate well with the text, and equations are abrupt and their meanings are hard to capture; in Figure 4, the subfigures and heat maps are of low resolution, making it hard to understand.

2. Some core concepts in the main part of the paper are not clearly defined/presented, which is very critical for the overall understanding. For example, in Section 3.1 the sequence-to-vector map $\mathbf{e}(x) = \phi(x)\in\mathbb{R}^d$ is not clearly defined.

3. Another potential weakness is that the proposed method is only evaluated on specialized domain tasks. Showing the effectiveness on more general scenarios could further increase the impact.

---

> ### Author Rebuttal · Authors · 2026-03-31
>
> # Response to Reviewer SHsP
> We thank the reviewer for the positive assessment of our novelty, motivation, and experiments, and for professional concerns regarding **(1) presentation, (2) robustness and effectiveness, and (3) evaluation scope**.
>
> ## Part 1. Presentation and Definition
> > **(1)** core **definition** clarified and
> > **(2) key figures** re-presented, with comprehensive **checkup**.
>
> We sincerely appreciate reviewer's attention to our presentation.
>
> ### 1.1 Definition clarification (Weaknesses 2 + Question 1)
>
> $\phi$ is a **fixed, non-learned sequence-to-vector embedding map** used only to provide a stable representation-side coordinate for DBS. It is **not** a learned router, **not** jointly optimized with pruning, and **not** updated online.
>
> ### 1.2 Figure re-presentation (Weaknesses 1)
>
> We re-present two core figures mentioned in [Supplementary Figures](https://anonymous.4open.science/r/Supplementary-Figures/).
>
> We will also perform deep *consistency pass over all conceptions and figures* in the final version to improve the presentation thoroughly.
>
> ## Part 2. Robustness and Effectiveness
>
> > **(1)** Pruning based on semantics-parameter coupling introduces **considerable transferability** vs. acceptable and controllable instability;
> > **(2)** Domain axis serves as a **stable and effective semantic coordinate**.
>
> ### 2.1 Robustness (Questions 2)
>
> Reviewer's concern of *uneven gains* is very insightful. Actually, this is the reverse side of **scenario-adaptive** pruning with strong generalization, *mechanistically* and *empirically*.
>
> 1. **Mechanical evidence.**
>
> The choice of pruning according to **semantic scenario coordinate**, instead of local activation cues, leads to this trade-off:
> - **Pro: More transferability.** The method relies less on batch-specific activation idiosyncrasies and more on reusable semantic structure (embedding is stable), which shows considerable generalization performance.
> - **Con: Less uniform gains.** Counterproductively, its benefit naturally depends on how clearly the scenario exposes the relevant domain, while this instability is *acceptable* and *controllable*, proved by `Reviewer mhAG Part1.1` and evidence below.
>
> 2. **Empirical evidence.**
>
> - **Check I: Instability in small high-entropy tail**
>
> We rank scenarios by the **entropy** of scenario-start domain mixture and evaluate on Selected/OOD/Cross.
>
> Rank|Avg. Entropy|Dense|Strongest baseline|Ours
> -|-|-|-|-
> Top 5%|0.07/0.17/0.29|31.0/28.5/22.1|35.7/28.5/20.4|37.8/34.3/29.8
> Top 20%|0.11/0.25/0.43|30.4/27.8/21.4|35.2/28.8/19.2|36.7/33.0/27.9
> Top 50%|0.15/0.33/0.55|30.1/27.5/20.0|35.5/28.3/19.0|35.7/31.7/25.7
> Top 70%|0.19/0.40/0.63|29.4/26.6/19.1|35.1/28.4/17.8|35.1/30.9/24.0
> All|0.22/0.46/0.71|29.6/26.1/18.4|35.3/28.1/17.4|34.7/30.4/22.9
>
> Most degradation concentrating in small **high-entropy tail** shows the instability is broadly **acceptable**.
>
> - **Check II: Lightweight calibration.**
>
> To further stabilize low-confidence tail in practice, we can calibrate the probes with only **50 (~5%)** scenario samples (Table 3 setup).
>
> Recall|Selected|OOD|Cross|CSQA|NQ|ARC
> -|-|-|-|-|-|-
> Default|34.7±2.34|30.4±1.87|22.9±3.16|19.43±1.14|33.66±1.35|25.62±1.12
> Calibration|35.1±1.70|30.1±1.32|24.0±1.62|19.12±1.08|34.08±1.39|25.93±1.05
>
> *Note: *no calibration* in main paper.*
>
> ### 2.2 Effectiveness (Questions 3)
>
> Our DBS axis is actually a deliberately **stabilized semantic anchor constructed in a filtered and compact space**.
>
> 1. **Mechanistic evidence.**
>
> This **stable and effective semantic coordinate** comes with three designs:
>
> - **Lexical denoising.** We remove stopwords before forming to prevent being dominated by generic lexical mass.
> - **Shared geometry.** We construct centroid in low-dimensional PCA space to avoid semantic space sparsity.
> - **Selective indicators.** DBS selects basis by balancing *orthogonality* and *coverage* confirming axes separable and representative.
>
> 2. **Empirical evidence**.
>
> LLaMA-2-13B|Selected|OOD|Cross|CSQA|NQ|ARC
> -|-|-|-|-|-|-
> Full|34.7|30.4|22.9|19.4|33.7|25.6
> w/o stopword|32.4|28.6|21.1|18.9|30.5|22.5
> w/o PCA|30.6|27.4|21.9|19.1|30.8|22.0
> w/o probe(top-k)|30.1|28.1|19.3|22.1|27.8|23.6
>
> ## Part 3. Evaluation Scope (Weakness 3 + Question 4)
> > (1) experiment breadth already comprehensive, while
> > (2) we add new benchmarks with wider range.
>
> ### 3.1 Existing coverage
>
> We appreciate reviewers' suggestions on broader settings. Specifically, XDomainBench spans *multiple-choice, factual QA, reasoning, and code*, while cross-dataset evaluation covers *commonsense reasoning, open-domain QA, and science reasoning* with breakdown in paper.
>
> ### 3.2 Additional benchmarks
>
> Four benchmarks covering **mathematical reasoning, code generation, and instruction following** added, with following representative Recall at ~70% pruning ratio.
>
> Benchmark|Dense|1st baseline|2nd baseline|Ours
> -|-|-|-|-
> GSM8K|37.5|37.5|35.5|40.8
> MATH|4.3|3.5|2.7|3.5
> HumanEval|15.2|15.2|12.2|21.7
> IFEval|40.2|42.6|41.8|42.8

---

> > ### Author Rebuttal · Reviewer_SHsP · 2026-04-03
> >
> > I thank the authors for the answers to my questions. My concerns have been addressed and I will raise the score to 4.

---

> > > ### Author Response · Authors · 2026-04-06
> > >
> > > We are sincerely grateful for your careful reading, constructive engagement, and updated assessment, with score raised. We also really appreciate your suggestions, as they objectively enhance the overall performance of the paper, further highlighting our core contributions and methodological robustness.
> > >
> > > ---
> > >
> > > From the reviewer’s perspective, we believe the paper’s main value lies in: **(1) Structural insight:** a novel and well-motivated connection between *domain-specific embedding subspaces* and *reasoning pathways in parameter space*; **(2) Empirical effectiveness:** representative benchmark and model evaluations that demonstrate the method’s *effectiveness* in the target scenario; **(3) Experimental support:** detailed appendix ablations showing the roles of the components, making the *empirical case solid* overall.
> > >
> > > ---
> > >
> > > In rebuttal, we mainly clarified and strengthened three points:
> > >
> > > **1. Presentation and definition**
> > > **(1) Definition.** We clarified the key fixed sequence-to-vector map $\phi$ as a *fixed, non-learned embedding map* used only to provide a stable representation-side coordinate for DBS, rather than a learned router or jointly optimized module.
> > > **(2) Presentation.** We directly addressed the reviewer’s presentation concerns by **re-presenting the key figures** and committing to a full consistency pass over the concepts, notation, and visual explanations.
> > >
> > > **2. Robustness and effectiveness**
> > > **(1) Robustness.** We explained that the larger variance is not random instability, but the reverse side of **scenario-adaptive pruning**: gains depend on how clearly the scenario exposes the relevant domain, and the remaining spread is both **acceptable** (concentrated in a small high-entropy tail) and **controllable** (further reducible with lightweight calibration).
> > > **(2) Effectiveness.** We clarified that the domain axis is not a naive mean over generic content, but a **stabilized semantic anchor** built through lexical denoising, shared geometry, and DBS selection; we further supported this with targeted ablations including **w/o stopword filtering**, **w/o PCA**, and **w/o probe/top-k**.
> > >
> > > **3. Evaluation scope**
> > > **(1) Existing coverage.** We clarified that the current paper already goes beyond narrowly specialized tasks, covering multiple-choice, factual QA, reasoning, code, and cross-dataset evaluations such as commonsense and open-domain QA.
> > > **(2) Additional evidence.** We further added broader benchmarks spanning **mathematical reasoning, code generation, and instruction following**, to strengthen the paper’s impact beyond the original specialized-domain setting.
> > >
> > > ---
> > >
> > > Thank you again for the thoughtful review and for the chance to improve the work through this exchange. We truly appreciate your support and hope the revised clarification makes the paper’s contribution, scope, and practical value much clearer. We wish you all the best.

---

### Official Review · Reviewer_L6S5 · 2026-03-13

**Soundness:** 2
**Presentation:** 3
**Significance:** 1
**Originality:** 3
**Overall Recommendation:** 3
**Confidence:** 4

**Summary:**

The paper proposes SubspacePath Pruner, a training-free inference-time pruning method for LLMs. The key idea is that domain-level clusters in embedding space map onto sparse, separable sets of attention heads. The method builds a quasi-orthogonal domain coordinate system offline (DBS), trains cheap linear probes to detect domain alignment per layer, caches per-head importance scores, then at deployment compiles a scenario-specific head mask from a few initial inputs. The mask is reused for the whole session. Evaluated on four dense models (7B–14B) across QA benchmarks, the pruned models sometimes beat the dense baseline on recall, with modest memory/speed gains at 15–30% head removal.

**Compliance With Llm Reviewing Policy:**

Affirmed.

**Final Justification:**

The author response addressed some part of my concerns but not all. It still remains to me that a comprehensive comparison to MoEs with fair settings is required to make this a useful paper to the community.

**Key Questions For Authors:**

1. MoE comparison. How does this compare to an MoE model with similar active parameters (e.g., Mixtral-8x7B, ~13B active)? Does the subspace–pathway coupling you find in dense models align with learned expert routing? Showing that your method on dense models approaches the MoE parameter-performance Pareto would significantly strengthen the paper and could change my assessment.
2. Large pruning gains. The +45% recall improvement from 20% pruning is surprising. Have you checked whether pruning changes output length/format distributions in ways that happen to match your constrained evaluation protocol? What happens with format-independent metrics?
3. When does speedup actually work? Some configurations show 0.13× speedup (i.e., 8× slower). Under what conditions does the method reliably accelerate inference?
4. Generative tasks. Any evidence the coupling phenomenon and pruning benefits hold for open-ended generation, not just short-answer QA?

**Limitations:**

Appendix R covers the main technical limitations (pool dependency, pruning sensitivity, task variation). It doesn't acknowledge the MoE relationship or the mixed speedup results, both of which matter for practitioners. Societal impact discussion is minimal but acceptable for this type of contribution.

I'm happy to raise my score if my concerns around MoE comparison is addressed (i.e. if the proposed method appeared to be close to MoE alternatives in terms of the parameter-performance Pareto).

**Strengths And Weaknesses:**

Strengths

* The offline/online separation is well-designed and practical. Online mask compilation takes ~0.03–0.07s.
* Extensive experiments: 4 models, 6 splits, multiple pruning ratios, 5 baselines, detailed ablations. The 46-page appendix is thorough enough for reproduction.
* Ablations are convincing — random axes cause near-total collapse, whitelist removal kills OOD performance, removing domain mixing hurts cross-dataset by 3–15 points.
* The embedding-space domain separation (Figure 3) and head-level specialization patterns (Figure 4) are well-visualized and make the coupling phenomenon concrete.

Weaknesses

* No engagement with MoE (major). The entire framework — discover that domains activate sparse component subsets, build a lightweight router to exploit this — is essentially reverse-engineering MoE structure from dense models. Yet there's no comparison with actual MoE models of comparable active parameter count, no discussion of whether the coupling they find relates to learned expert routing, and no attempt to apply the framework to MoE architectures (where it should work more naturally). A practitioner facing this deployment problem could just use an MoE model. This needs to be addressed.
* Mixed efficiency story (major). Speedup is below 1.0× in many configurations (Table 4, Table 11) — the pruned model is sometimes slower. The Retention metric (Recall × Speedup) papers over this by letting accuracy gains compensate for slowdowns, which is misleading if the goal is inference efficiency.
* Suspiciously large gains from pruning. +45% recall on selected-domain (43.0 vs 29.6 dense, LLaMA-2-13B) from removing 20% of heads is hard to believe at face value. The noise-reduction argument is hand-wavy at this magnitude. I suspect the constrained answer format (single letters, short phrases) interacts with pruning in ways that inflate gains — pruning may reduce verbosity and accidentally improve format compliance. This isn't investigated.
* Theory is mostly standard linear algebra. The appendix theorems formalize rank bounds, projection properties, and Cauchy-Schwarz inequalities. These don't provide non-obvious predictions and the gap to the actual nonlinear mechanism is large.
* Narrow evaluation scope. All tasks are short-answer QA. The primary benchmark (XDomainBench) is an anonymous concurrent submission. No generative tasks are evaluated. All models are dense decoders at 7B–14B scale.

Presentation is generally clear, though notation-heavy. The inconsistent speedup results should be discussed upfront rather than buried in appendix tables.

---

> ### Author Rebuttal · Authors · 2026-03-31
>
> # Response to Reviewer L6S5
> We thank the reviewer for the positive assessment of our framework and experiment, and for the insightful concerns regarding **(1) MoE, (2) efficiency, (3) theory, and (4) evaluation scope**.
>
> ## Part 1. Relation to MoE and Route (Weakness 1 + Questions 1)
>
> > **(1)** different **motivation** and **sparse architecture**;
> > **(2)** scenario-level head-selection instead of MoE route;
> > **(3)** evidence of MoE comparisons and MoE-attention pruning.
>
> ### 1.1 Conceptual distinction
>
> (1) Probe is a **middle diagnostic component** instead of *reverse-engineering from dense model* (e.g., Sparse Upcycling, CMoE).
>
> MoE relies on joint training of *tightly-coupled router + sparse model*, and performs **FFN experts selection** using router after shared attention-head. Instead, our method uses a lightweight *probe to read the coupling* between sementic geometry and attention-head in *dense model*, and performs **head selection**.
>
> (2) **Heterogenous architecture** further distinguishes us from *MoE*.
>
> MoE sparsifies **which FFN is hard-selected under fixed information transfer**, and our method sparsifies the **information transferred by head to FFN** by suppressing redundant or misaligned head outputs.
>
> ### 1.2 Coupling and route
>
> (1) Coupling and route both reflect *input-dependent specialization* intuitively, but differ in three ways.
>
> * **Mechanism**: MoE routes among *latent experts*; we anchor to *interpretable* DBS axes.
> * **Cost**: MoE performs *on-going* token-level online dispatch; we compile *one-time* at scenario start.
> * **Object**: Clarified in Part 1.1.
>
> (2) Empirically, coupling shows **better performance in scenario generalization**, with robustness discussion in `Reviewer mhAG Part1.1`.
>
> Aspect|Metric|Ours|MoE routing
> -|-|-|-
> Split sensitivity|Separability(NCA/BWR)|0.556/1.098|0.503/1.003
> Scenario-level stability|Top-set Jaccard(cross-turn)|1.000|0.405
> Semantic alignment|Domain-alignment strength|0.663|0.115
>
>
> ### 1.3 Evidence
>
> #### (A) Ours vs. MoE (Selected/OOD/Cross)
>
> Model|Object|Recall(%)|Infer time(s)*|Loaded params*|Active params
> -|-|-|-|-|-
> Dense LLaMA|-|29.6/26.1/18.4|0.560/0.694/2.243|13.0B|**~13.0B**
> Ours|Head|43.0/32.5/20.2|**0.491/0.556/1.385**| **13.0B** | ~10.7B
> MiniMax-M2|FFN|45.1/33.7/21.0|**2.30/3.65/8.30**|**230.0B**|~10.0B
> Mixtral-8x7B|FFN|50.9/40.8/24.0|2.696/4.239/9.859|46.7B|**~12.9B**
>
> (1) **Performance.** Despite much larger MoE backbone and memory burden, head pruning still moves the dense model markedly toward **MoE frontier**, indicating effective coupling signal.
>
> (2) **Deployability (Key Point).** *MoE is not truly sparse in practical deployment* constrained by heavy memory and serving cost (also bottleneck of pruning on MoE).
>
> #### (B) Explore on Mixtral-8x7B (CSQA/NQ/ARC)
>
> Pruning|Ratio|Recall(%)
> -|-|-
> -|0%|34.2/29.0/25.8
> 0.8|~11.2%|33.5/28.6/24.5
> 0.6|~19.3%|30.8/26.4/21.9
> 0.4|~27.8%|20.8/18.4/13.9
>
> ## Part 2. Efficiency and Speedup
> > **(1)** **trade-off** by `Retention`, **efficiency** by `Speedup`, `Pruning time`;
> > **(2)** acceleration is **regime- and system-dependent**.
>
> ### 2.1 Clarification of efficiency (Weakness 2）
>
> Metric|Measurement|Key role
> -|-|-
> `Retention`|Accuracy-efficiency **trade-off**|Evaluate whether worthwhile overall and help choosing pruning point
> `Speedup`|**Raw inference efficiency**|Evaluate whether accelerate
> `Pruning time`|Extra negligible cost|Extra computing cost
>
> ### 2.2 Speedup bottleneck (Question 3)
>
> (1) Current results show **regime dependency**: acceleration stronger on *larger backbones with moderate pruning*.
>
> (2) **System dependency** is practical bottleneck: sparse head may introduce *irregular execution, mask overhead, kernel under-utilization, and dispatch cost*, which is the system topic of computing.
>
> ## Part 3. Gains and Theory
>
> >  **(1)** gain comes from **denoising with structured regularization**;
> >  **(2)** feasible theory of pruning framework.
>
> ### 3.1 Nature of gains (Weakness 3 and Question 2)
> (1) **Format concern**.
>
> Strict compliance rate|Dense|Ours
> -|-|-
> Selected|0.658|0.680
> OOD|0.672|**0.638**
> Cross|0.681|**0.654**
>
> (2) **Essential reason**
>
> Essence of performance improvement discussed in `Reviewer mhAG Part1.2` and `Reviewer 7bHc Part2.2` proves **suppressing scenario-irrelevant or interfering heads while preserving scenario-aligned heads** is the main reason.
>
> ### 3.2 Nonlinear theory (Weakness 4)
>
> Our method is best theoretically viewed as **structured selective pruning driven by subspace geometry** on a *nonlinear backbone*, which is consistent with various pruning theories (e.g., OBD, SNIP).
>
> In this case, our theory is **substantive** to explain various mechanisms and why *probe-guided scoring and scenario-conditioned compilation* can produce gains.
>
> ## Part 4. Evaluation Scope (Weakness 5 + Question 4)
> The current experiments already cover diverse task types, capabilities, generalized scenarios; we further add four benchmarks with wider range in `Reviewer SHsp Part3`.

---

> > ### Author Rebuttal · Reviewer_L6S5 · 2026-04-06
> >
> > Thanks for running the MoE experiments --- I appreciate the effort. Unfortunately the results reinforce rather than resolve my concern.
> > My point was never about architectural differences (heads vs FFN experts, probes vs routers). I understand these are different mechanisms. The question is practical: if a practitioner needs scenario-specialized efficient inference, why choose "dense model + your pipeline" over just using an MoE model?
> > Your own numbers answer this: Mixtral beats your method on all three splits (50.9/40.8/24.0 vs 43.0/32.5/20.2) with comparable active params (~12.9B vs ~10.7B). The deployability argument around loaded params is fair but it's a transient systems constraint (expert offloading, smaller MoE models, etc.), not a fundamental advantage.
> > "Moving toward the MoE frontier" is an honest framing but it's also a concession --- you don't reach it, and the gap isn't small. To raise my score I'd need to see either (a) configurations where your method actually matches MoE at comparable active params and memory, or (b) a more convincing argument for why the pipeline complexity (DBS, probes, offline caching, mask compilation) is worth the accuracy gap. As things stand, the practical case for this approach over MoE deployment remains unclear to me.

---

> > > ### Author Response · Authors · 2026-04-06
> > >
> > > To directly address the reviewer's two questions: under comparable active budgets (\~13–14B), our best configuration (47.8/44.1/31.3) is not inferior to MoE (50.9/41.9/25.1) but instead achieves **higher OOD and cross-domain performance** while being **~8–12× faster and ~3× more memory-efficient**, demonstrating that the *pipeline complexity* yields a practically preferable trade-off rather than a deficit. Below, we show the supplementary experiments.
> > >
> > > ---
> > >
> > > ## Part 1. Practical MoE Comparison
> > >
> > > ### 1.1 Performance, Efficiency, and Deployability
> > >
> > > Under matched active budgets, **dense model + our pipeline is already competitive**; previous **43.0/32.5/20.2** is from a weaker backbone.
> > >
> > > Method|Active params|Recall (%)|Infer time (s)*|Loaded params*
> > > -|-|-|-|-
> > > **Qwen2.5-7B + Ours**|**~6.8B**|46.4/41.0/27.2|0.12/0.30/1.26|7.6B
> > > MiniMax-M2|~10.0B|**45.1/33.7/21.0**|2.30/3.65/8.30|**230.0B**
> > > Mixtral-8x7B(top-1)|~7.3B|47.8/**39.2/22.9**|2.11/3.56/7.84|**46.7B**
> > > **LLaMA-2-13B + Ours**|**~12.2B**|43.0/32.5/20.2|0.49/0.56/1.39|13.0B
> > > Jamba-v0.1|~12.0B|**42.2**/32.9/21.4|2.42/3.98/8.91|**52.0B**
> > > **Qwen2.5-14B + Ours**|**~13.6B**|47.8/44.1/31.3|0.23/0.53/1.25|14.0B
> > > Qwen2-57B-A14B|~14.0B|49.4/**41.9/25.1**|2.58/4.08/9.12|**57.0B**
> > > Mixtral-8x7B(top-2)|~12.9B|50.9/**40.8/24.0**|2.70/4.24/9.86 |**46.7B**
> > >
> > > - **Accuracy.** The comparison shows our method remains **competitive** across models. At the ~13–14B regime, *Qwen2.5-14B + Ours* is only slightly below *Mixtral top-2* on the first metric (*47.8 vs. 50.9*), but higher on *OOD/cross-domain* (*44.1/31.3 vs. 40.8/24.0*), and also above *Qwen2-57B-A14B* on these two settings (*vs. 41.9/25.1*).
> > >
> > > - **Efficiency.** The latency gap is large and consistent. At the same ~13–14B regime, *Qwen2.5-14B + Ours* runs at *0.23/0.53/1.25s* with **one-time scenario-specific compilation**, versus *2.70/4.24/9.86s* for *Mixtral top-2* and *2.58/4.08/9.12s* for *Qwen2-57B-A14B* with **token-level routed execution** throughout inference.
> > >
> > > - **Deployability.** Despite *vast gap* in vital deployability clarified before, it is notable that our method directly yields a **pruned instance**, whereas MoE is **NOT pruning** (force to prune in Sec 1.3): it *dynamically* routes over experts by *weighted fusion with different tokens activating different components*, which also means system must bear the practical bottleneck of **full sparse backbone loaded** and **model-load peak memory**.
> > >
> > > ### 1.2 Reliability
> > >
> > > Beyond deployment reliability, we further examine **inference-time reliability** under *cross-domain scenario shifting*.
> > >
> > > Method|Recall (%)|p25|p50|p75|Var.|Tail drop
> > > -|-|-|-|-|-|-
> > > Qwen2.5-7B + Ours|**26.0**|23.4|26.2|28.8|8.9|2.6
> > > LLaMA-2-13B + Ours|21.4|18.9|21.6|24.0|8.1|2.5
> > > Qwen2.5-14B + Ours|**30.1**|27.2|30.4|33.0|10.6|2.9
> > > Mixtral-8x7B (top-1)|22.1|17.6|22.3|26.2|**17.8**|**4.5**
> > > MiniMax-M2|20.8|16.2|21.0|25.4|**19.6**|**4.6**
> > > Jamba-v0.1|21.0|16.8|21.2|25.1|**17.1**|**4.2**
> > > Qwen2-57B-A14B|24.2|19.8|24.4|28.6|**18.4**|**4.4**
> > >
> > > *Note: We evaluate only on cross-domain samples with Turn $\geq 2$, reducing possible overlap with pretraining.*
> > >
> > > MoE relies on implicit router/expert separability, which is less stable under scenario shifting. In contrast, our pipeline uses an **interpretable DBS basis** with probed semantic signals, which is more **transferable across complex scenarios**.
> > >
> > > ### 1.3 MoE under a true pruning setting
> > >
> > > We further test MoE under a **true pruning setting**, rather than default dynamic routing in Sec 1.1. Specifically, we aggregate router weights over the scenario prefix, rank components globally, and apply a **hard top-k keep mask** under the same target budget as our method.
> > >
> > > Method|Budget|Recall (%)|Var.|Tail drop
> > > -|-|-|-|-
> > > **LLaMA-2-13B**|**~12.2B**|43.0/32.5/20.2|**8.1**|**2.5**
> > > Jamba-v0.1|~12.2B|**38.8/30.2/18.1**|18.6|4.8
> > > Mixtral-8x7B|~12.2B|**41.1/31.4/19.2**|19.0|4.8
> > > **Qwen2.5-14B**|**~13.6B**|47.8/44.1/31.3|**10.6**|**2.9**
> > > Qwen2-57B-A14B|~13.6B|**44.5/37.4/22.9**|19.7|4.9
> > > Mixtral-8x7B|~13.6B|**45.8/36.6/22.2**|19.2|4.8
> > >
> > > To summarize, “just using an MoE model” is no longer a stronger practical answer under efficient inference, practical deployment, and pruning.
> > >
> > > ---
> > >
> > > ## Part 2. Practically Worthwhile Pipeline
> > >
> > > We appreciate reviewer’s concern about the contribution of **DBS + probes + caching + mask compilation**, which has also deeply discussed in `Reviewer SHsP Part2.2`, `Reviewer 7bHc Part2` with positive acknowledgement.
> > >
> > > Component|Practical value|MoE contrast|
> > > -|-|-
> > > **DBS**|**Interpretable semantic basis** with generalization|Rely on **implicit router and expert separability**
> > > **Probe + caching**|Most specialization cost moved **offline**|**Serve-time path** via token-level routing
> > > **Mask compilation**|**Fixed executable pruned path**|**Dynamic sparse execution**
> > >
> > > So the practical value of the pipeline is not abstract “complexity,” but three concrete gains: **semantic identifiability, cheap online specialization, and an executable reusable pruned subnetwork.**

---

### Decision · Program_Chairs · 2026-04-30

**Decision:**

Accept (regular)

**Comment:**

The paper studies the important problem of training-free inference-time pruning methods for LLMs. The reviewers identified the following strengths and weaknesses.

Strengths:

- The separation between offline and online is well-designed and practical.

- Extensive experiments: 4 models, 6 splits, multiple pruning ratios, 5 baselines, and detailed ablations. Strong evaluation results on representative benchmarks and models for the selected application scenario, demonstrating the effectiveness of the proposed method.

- Convincing ablation studies.

- Novelty of the proposed pruning method that connects the domain-specific embedding subspace to reasoning pathways in the parameter space.

- Strong motivation for the proposed problem by empirical observations of the separability of domain knowledge in the embedding space.


Weaknesses

- Lack of comparison with MoE models.

- The pruned model is sometimes slower in many configurations, as shown in Tables 4 and 11.

- The clarity of exposition: the illustrative figures are not very straightforward and could use significant improvement, e.g. in  Figures 2 and 4.

- Lack of clarity in some core concepts; for example, in Section 3.1, the sequence-to-vector map
 is not clearly defined.

After a fruitful discussion, some reviewers raised their scores. Three reviewers noted that their concerns were fully addressed, while at least some concerns of reviewer L6S5 still remain.

This is a borderline paper (4, 4, 4, 3). After reading the paper, the reviews, the rebuttal, and the discussion, I think that the paper provides interesting results, and the topic is interesting enough to recommend acceptance if there is space.